# DeFoG: Defogging Discrete Flow Matching for Graph Generation

## Abstract

Graph generation is fundamental in diverse scientific applications, due to its ability to reveal the underlying distribution of complex data, and eventually generate new, realistic data points. Despite the success of diffusion models in this domain, those face limitations in sampling efficiency and flexibility, stemming from the tight coupling between the training and sampling stages. To address this, we propose DeFoG, a novel framework using discrete flow matching for graph generation. DeFoG employs a flow-based approach that features an efficient linear interpolation noising process and a flexible denoising process based on a continuous-time Markov chain formulation. We leverage an expressive graph transformer and ensure desirable node permutation properties to respect graph symmetry. Crucially, our framework enables a disentangled design of the training and sampling stages, enabling more effective and efficient optimization of model performance. We navigate this design space by introducing several algorithmic improvements that boost the model performance, consistently surpassing existing diffusion models. We also theoretically demonstrate that, for general discrete data, discrete flow models can faithfully replicate the ground truth distribution - a result that naturally extends to graph data and reinforces DeFoG's foundations. Extensive experiments show that DeFoG achieves state-of-the-art results on synthetic and molecular datasets, improving both training and sampling efficiency over diffusion models, and excels in conditional generation on a digital pathology dataset.

## 1 Introduction

Graph generation has become a fundamental task across diverse fields, from molecular chemistry to social network analysis, due to graphs' capacity to represent complex relationships and generate novel, realistic structured data. Diffusion-based graph generative models (Niu et al., 2020; Jo et al., 2022), particularly those tailored for discrete data to capture the inherent discreteness of graphs (Vignac et al., 2022; Siraudin et al., 2024), have emerged as a compelling approach, demonstrating state-of-the-art performance in applications such as molecular generation (Irwin et al., 2024), reaction pathway design (Igashov et al., 2024), neural architecture search (Asthana et al., 2024), and combinatorial optimization (Sun & Yang, 2024). However, the sampling processes of these methods remain highly constrained due to a strong interdependence with training: design choices made during training mostly determine the options available during sampling. Therefore, optimizing parameters such as noise schedules requires re-training, which incurs significant computational costs.

Recently, Campbell et al. (2024); Gat et al. (2024) introduced a discrete flow matching (DFM) framework, which maps noisy to clean discrete data with iterations similar to diffusion models. However, DFM has demonstrated superior performance compared to discrete diffusion models by leveraging a straightforward and efficient noising process, and a flexible formulation based on continuous-time Markov chain (CTMC) for denoising. Specifically, its noising process involves a linear interpolation in the probability space between noisy and clean data, displaying well-established advantages for continuous state spaces in terms of performance (Esser et al., 2024; Lipman et al.) and theoretical properties (Liu et al., 2023). Most notably, the denoising process in DFM uses an adaptable sampling step scheme (both in size and number) and CTMC rate matrices that are independent of the training setup. Such a decoupling allows for the design of sampling algorithms that are disentangled from training, enabling efficient performance optimization at the sampling stage without extensive retraining. However, despite its promising results, the applica-

bility of DFM to graph generation remains unclear, and the proper exploitation of its additional training-sampling flexibility is still an open question in graph settings.

In this work, we propose DeFoG, a novel and versatile DFM framework for graph generation that operates under theoretical guarantees, featuring a linear interpolation noising process and CTMC-based denoising process inherited from the DFM model. Our contribution begins by proposing the first discrete flow matching model tailored to graphs, ensuring node permutation equivariance to respect graph symmetry and addressing the model expressivity limitations inherent to this data modality (Morris et al., 2019). Additionally, to "defog" the design space of this decoupled training-sampling framework, we extend and propose novel training and sampling algorithms, including alternative initial distributions, modifications of the CTMC rate matrices that govern the denoising trajectory, and time-adaptive strategies designed to align with dataset-specific characteristics. We explicitly demonstrate that these algorithmic improvements, which can be seamlessly integrated into training and sampling, accelerate convergence and enhance generation performance by a large margin. To assert DeFoG's strong theoretical foundation, we extend existing guarantees of DFM models for general discrete data. First, we reinforce the pertinence of the loss function by explicitly relating it to the model estimation error. Then, we demonstrate that the distribution generated by the denoising process in DFM models remains faithful to the ground truth distribution. Both theorems extend naturally to graph data, providing strong support for the design choices of our new graph generative framework.

Our experiments show that DeFoG achieves state-of-the-art performance on both synthetic graph datasets with diverse topologies and complex molecular datasets. Notably, while the vanilla DeFoG framework performs on par with diffusion models, our tailored training-sampling optimization pipeline permits DeFoG to consistently outperform these models. Furthermore, DeFoG achieves performance comparable to certain diffusion models with only 5 to 10% of the sampling steps, highlighting significant efficiency gains. We also underscore DeFoG's versatility through a conditional generation task in digital pathology, where it notably outperforms existing unconstrained models. In a nutshell, DeFoG opens the door to more flexible graph generative models that can efficiently handle a wide range of tasks in diverse domains with improved performance and reduced computational costs.

## 2 RELATED WORK

**Graph Generative Models** Graph generation has applications across various domains, including molecular generation (Mercado et al., 2021), combinatorial optimization (Sun & Yang, 2024), and inverse protein folding (Yi et al., 2024). Existing methods for this task generally fall into two main categories. *Autoregressive* models progressively grow the graph by inserting nodes and edges (You et al., 2018; Liao et al., 2019). Although these methods offer high flexibility in sampling and facilitate the integration of domain-specific knowledge (e.g., for molecule generation, Liu et al. (2018) perform valency checks at each iteration), they suffer from a fundamental drawback: the need to learn a node ordering (Kong et al., 2023), or use a predefined node ordering (You et al., 2018) to avoid the overly large learning space. In contrast, *one-shot* models circumvent such limitation by predicting the entire graph in a single step, enabling the straightforward incorporation of node permutation equivariance/invariance properties. Examples of these approaches include graph-adapted versions of VAEs (Kipf & Welling, 2016), GANs (De Cao & Kipf, 2018), or normalizing flows (Liu et al., 2019). Among these, diffusion models have gained prominence for their state-of-the-art performance, attributed to their iterative mapping between noise and data distributions.

**Graph Diffusion** One of the initial research directions in graph diffusion sought to adapt continuous diffusion frameworks (Sohl-Dickstein et al., 2015; Ho et al., 2020; Song et al., 2020) for graph-structured data (Niu et al., 2020; Jo et al., 2022; 2024), which introduced challenges in preserving the inherent discreteness of graphs. In response, discrete diffusion models (Austin et al., 2021) were effectively extended to the graph domain (Vignac et al., 2022; Haefeli et al.), utilizing Discrete-Time Markov Chains to model the stochastic diffusion process. However, this method restricts sampling to the discrete time points used during training. To address this limitation, continuous-time discrete diffusion models incorporating CTMCs have emerged (Campbell et al., 2022), and have been recently applied to graph generation (Siraudin et al., 2024; Xu et al., 2024). Despite employing a continuous-time framework, their sampling optimization space remains limited by training-dependent design choices, such as fixed rate matrices, which hinders further performance gains.

Figure 1: Overview of DeFoG. One node $x$, with 3 potential states {green, yellow, red}, is selected to illustrate both *noising* and *denoising* processes. For *noising*, DeFoG follows a straight path from the one-hot encoding $p_1$ of the clean node to the initial distribution $p_0$. For *denoising*, a network parameterized by $\theta$ predicts the marginal distributions of the clean graph, $p_{1|t}^\theta(\cdot|G_t)$. The predicted distribution for the node $x$ is then used to compute its rate matrix $R_t^\theta$ and, subsequently, its probability at the next time point. This is applied to all nodes and edges to denoise the entire graph.

**Discrete Flow Matching** Flow matching (FM) models emerged as a compelling alternative to diffusion models in the realm of iterative refinement generative approaches for continuous state spaces (Lipman et al.; Liu et al., 2023). This framework has been empirically shown to enhance performance and efficiency in image generation (Esser et al., 2024; Ma et al., 2024). To address discrete state spaces, such as those found in protein design or text generation, a DFM formulation has been introduced (Campbell et al., 2024; Gat et al., 2024). This DFM approach theoretically streamlines its diffusion counterpart by employing linear interpolation for the mapping from data to the prior distribution. Moreover, its CTMC-based denoising process accommodates a broader range of rate matrices, which need not be fixed during training. In practice, DFM consistently outperforms traditional discrete diffusion models. In this paper, we build on the foundations of DFM with the aim of further advancing graph generative models.

## 3 DISCRETE FLOW MATCHING

In this section, we introduce the DFM framework as originally proposed by Campbell et al. (2024). We adopt their notation for clarity and illustrate it under the graph setting in Figure 1.

### 3.1 CONTINUOUS-TIME MARKOV CHAINS AND DISCRETE FLOW MATCHING

In generative modeling, the primary goal is to generate new data samples from the underlying distribution that produced the original data, $p_{\text{data}}$. An effective approach is to learn a mapping between a simple distribution $p_\epsilon$, which is easy to sample from, and $p_{\text{data}}$. DFM models achieve this mapping via a stochastic process over the time interval $t \in [0, 1]$ for variables in discrete state spaces. The initial distribution, $p_0 = p_\epsilon$, is a predefined noise distribution, while $p_1 = p_{\text{data}}$ represents the target data distribution.

**Univariate Case** At any time $t$, we consider a discrete variable with $Z$ possible values, denoted $z_t \in \mathcal{Z} = \{1, \ldots, Z\}$. The *marginal distribution* of $z_t$ is represented by the vector $p_t \in \Delta^{Z-1}$, where $\Delta^{Z-1}$ represents the corresponding probability simplex[1]. Since handling $p_t$ directly is not straightforward, DFM establishes a conditioned *noising* trajectory from the chosen datapoint $z_1$ to the initial distribution through a simple linear interpolation:

$$p_{t|1}(z_t|z_1) = t\,\delta(z_t, z_1) + (1-t)\,p_0(z_t). \tag{1}$$

where $\delta(z_t, z_1)$ is the Kronecker delta which is 1 when $z_t = z_1$. Importantly, the original marginal distribution can be backtracked via $p_t(z_t) = \mathbb{E}_{p_1(z_1)}[p_{t|1}(z_t|z_1)]$.

In the *denoising* stage, DFM reverses the previous process using a CTMC formulation. In general, a CTMC is characterized by an initial distribution, $p_0$, and a *rate matrix*, $R_t \in \mathbb{R}^{Z \times Z}$ that models its evolution across time $t \in [0, 1]$. Specifically, the rate matrix defines the instantaneous transition rates between states, such that:

---

[1] $\Delta^K = \left\{ \mathbf{x} \in \mathbb{R}^{K+1} \mid \sum_{i=1}^{K+1} x_i = 1,\ x_i \geq 0,\ \forall i \right\}$

$$p_{t+\mathrm{d}t|t}(z_{t+\mathrm{d}t}|z_t) = \delta(z_t, z_{t+\mathrm{d}t}) + R_t(z_t, z_{t+\mathrm{d}t})\mathrm{d}t. \tag{2}$$

Intuitively, $R_t(z_t, z_{t+\mathrm{d}t})\mathrm{d}t$ yields the probability that a transition from state $z_t$ to state $z_{t+\mathrm{d}t}$ will occur in the next infinitesimal time step $\mathrm{d}t$. By definition, we have $R_t(z_t, z_{t+\mathrm{d}t}) \geq 0$ for $z_t \neq z_{t+\mathrm{d}t}$. Consequently, $R_t(z_t, z_t) = -\sum_{z_{t+\mathrm{d}t} \neq z_t} R_t(z_t, z_{t+\mathrm{d}t})$ to ensure $\sum_{z_{t+\mathrm{d}t}} p_{t+\mathrm{d}t|t}(z_{t+\mathrm{d}t}|z_t) = 1$. Under this definition, the marginal distribution and the rate matrix of a CTCM are related by a conservation law, the Kolmogorov equation, given by $\partial_t p_t = R_t^T p_t$. If expanded, this expression unveils the time derivative of the marginal distribution as the net balance between the inflow and outflow of probability mass at that state. When this relation holds, we say that $R_t$ *generates* $p_t$.

Since we aim to reverse a $z_1$-conditional noising process $p_{t|1}(\cdot|z_1) \in \Delta^{Z-1}$ (recall Eq. (1)), DFM instead considers a $z_1$-conditional rate matrix, $R_t(\cdot, \cdot|z_1) \in \mathbb{R}^{Z \times Z}$ that generates it. Under mild assumptions, Campbell et al. (2024) present a closed-form for a valid conditional rate matrix (though others exist), i.e., that verifies the corresponding Kolmogorov equation, defined as:

$$R_t^*(z_t, z_{t+\mathrm{d}t}|z_1) = \frac{\mathrm{ReLU}[\partial_t p_{t|1}(z_{t+\mathrm{d}t}|z_1) - \partial_t p_{t|1}(z_t|z_1)]}{Z_t^{>0}\, p_{t|1}(z_t|z_1)} \quad \text{for } z_t \neq z_{t+\mathrm{d}t}, \tag{3}$$

and $Z_t^{>0} = |\{z_t : p_{t|1}(z_t|z_1) > 0\}|$. Again, normalization is performed for the case $z_t = z_{t+\mathrm{d}t}$. Intuitively, $R_t^*$ redistributes probability mass by applying a positive rate to states needing more mass than the current state $z_t$.

**Multivariate Case** We jointly model $D$ variables, $(z^{(1)}, \ldots, z^{(D)}) = z^{1:D} \in \mathcal{Z}^D$. The noising process defined in Eq. (1) is performed independently for each variable, i.e., $p_{t|1}(z_t^{1:D}|z_1^{1:D}) = \prod_{d=1}^D p_{t|1}(z_t^{(d)}|z_1^{(d)})$. For the denoising process, Campbell et al. (2024) propose the utilization of an efficient approximation for CTCM simulation on multivariate non-ordinal data, where an Euler step is applied independently to each dimension $d$, with a finite time step $\Delta t$:

$$\tilde{p}_{t+\Delta t|t}(z_{t+\Delta t}^{1:D}|z_t^{1:D}) = \prod_{d=1}^D \underbrace{\left( \delta(z_t^{(d)}, z_{t+\Delta t}^{(d)}) + \mathbb{E}_{p_{1|t}^{(d)}(z_1^{(d)}|z_t^{1:D})}\left[ R_t^{(d)}(z_t^{(d)}, z_{t+\Delta t}^{(d)}|z_1^{(d)}) \right] \Delta t \right)}_{\tilde{p}_{t+\Delta t|t}^{(d)}(z_{t+\Delta t}^{(d)}|z_t^{1:D})}. \tag{4}$$

### 3.2 Modeling

Next, we discuss the practical implementation of the framework above, focusing on its two main components: the training loss formulation and the iterative sampling algorithm for denoising.

**Training** Eq. (4) employs $\mathbb{E}_{p_{1|t}^{(d)}(z_1^{(d)}|z_t^{1:D})}\left[ R_t^{(d)}(z_t^{(d)}, z_{t+\Delta t}^{(d)}|z_1^{(d)}) \right]$, which requires computing the expectation under the condition $p_{1|t}^{(d)}(z_1^{(d)}|z_t^{1:D})$. This term consists of a prediction of the clean data $z_1$ from the noisy joint variable $z_t^{1:D}$. However, unlike the noising process, the denoising process does not factorize across dimensions, rendering such predictions intractable in general. Instead, we use a neural network, $f_\theta$, parameterized by $\theta$, to approximate it, i.e., $p_{1|t}^{\theta,(d)}(z_1^{(d)}|z_t^{1:D}) \approx p_{1|t}^{(d)}(z_1^{(d)}|z_t^{1:D})$. In DFM, the network is trained by minimizing the sum of the cross-entropy losses over all variables:

$$\mathcal{L}_{\mathrm{DFM}} = \mathbb{E}_{t \sim \mathcal{U}[0,1],\, p_1(z_1^{1:D}),\, p_{t|1}(z_t^{1:D}|z_1^{1:D})}\left[ -\sum_d \log(p_{1|t}^{\theta,(d)}(z_1^{(d)}|z_t^{1:D})) \right], \tag{5}$$

where $\mathcal{U}[0,1]$ denotes the uniform distribution between 0 and 1.

**Sampling** Generating new samples using DFM amounts to simulate the CTCM formulated with rate matrices for the denoising process. This is accomplished by sampling an initial datapoint, $z_0^{1:D}$ sampled from the initial distribution $p_0$ and iteratively applying Eq. (4), using $p_{1|t}^{\theta,(d)}(z_1^{(d)}|z_t^{1:D})$.

## 4 Discrete Flow Matching for Graphs

In this section, we introduce DeFoG, a novel and flexible DFM framework for graph generation. We first describe the necessary components and desired properties for extending the method to graph data. Then, we explore various algorithmic improvements designed to enhance performance.

## 4.1 ADAPTATION OF DFM TO GRAPHS

Instead of considering the general multivariate data denoted as $z^{1:D}$, we now instantiate undirected graphs with $N$ nodes as a $D$-dimensional variable. We define the node set $x^{1:n:N} = (x^{(n)})_{1 \leq n \leq N}$ and the edge set $e^{1:i<j:N} = (e^{(ij)})_{1 \leq i<j \leq N}$. A graph corresponds to $G = (x^{1:n:N}, e^{1:i<j:N})$, such that $x^{(n)} \in \mathcal{X} = \{1, \ldots, X\}$, $e^{(ij)} \in \mathcal{E} = \{1, \ldots, E\}$, and $D = N + N(N-1)/2$, thus amenable to the DFM framework. However, graphs are not merely joint variables of nodes and edges; they encode complex structures, relationships, and global properties essential for modeling real-world systems, while adhering to specific invariances. To address these graph-specific challenges, we propose *DeFoG* (Discrete Flow Matching on Graphs), a framework tailored to develop graph generation based on the DFM paradigm. Below, we outline how DeFoG achieves these desired properties for graphs.

**Denoising Neural Architecture** The denoising of DFM requires a noisy graph $G_t$ as input and predicts the clean marginal probability for each node $x^{(n)}$ via $p_{1|t}^{\theta,(n)}(\cdot|G_t) \in \Delta^{X-1}$ and for each edge $e^{(ij)}$ via $p_{1|t}^{\theta,(ij)}(\cdot|G_t) \in \Delta^{E-1}$. They are gathered in $p_{1|t}^{\theta}(\cdot|G_t) = (p_{1|t}^{\theta,1:n:N}(\cdot|G_t), p_{1|t}^{\theta,1:i<j:N}(\cdot|G_t))$. In practice, the model receives a noisy graph where node and edge features are one-hot encoded representations of their states and returns a graph of the same dimensionality, with features corresponding to the node and edge marginal distributions at $t = 1$. This formulation boils down the graph generative task to a simple graph-to-graph mapping. Therefore, it is crucial to ensure maximal expressivity while preserving node permutation equivariance, making a Graph Transformer ($f_\theta$) a suitable choice for this mapping (Vignac et al., 2022).

**Enhancing Model Expressivity** Graph neural networks suffer from inherent expressivity constraints (Xu et al., 2019). A usual approach to overcome their limited representation power consists of explicitly augmenting the inputs with features that the networks would otherwise struggle to learn. We adopt Relative Random Walk Probabilities (RRWP) encodings that are proved to be expressive for both discriminative (Ma et al., 2023) and generative settings (Siraudin et al., 2024). RRWP encodes the likelihood of traversing from one node to another in a graph through random walks of varying lengths. In particular, given a graph with an adjacency matrix $A$, we generate $K - 1$ powers of its degree-normalized adjacency matrix, $M = D^{-1}A$, i.e., $[I, M, M^2, \ldots, M^{K-1}]$. We concatenate the diagonal entries of each power to their corresponding node embedding, while combining and appending the non-diagonal to their corresponding edge embeddings. RRWP features also stand out for being efficient to compute compared to the spectral and cycle features used in Vignac et al. (2022), as demonstrated in Appendix G.3.

**Loss Function** The nodes and edges of graphs encode distinct structural information, justifying their differential treatment in a graph-specific generative model. This distinction should be reflected in the training loss function. We define $\mathcal{L}_{\text{DeFoG}} = \mathbb{E}_{t \sim \mathcal{U}[0,1], p_1(G_1), p_{t|1}(G_t|G_1)} \text{CE}_\lambda(G_1, p_{1|t}^{\theta}(\cdot|G_t))$ similarly to $\mathcal{L}_{\text{DFM}}$, in Eq. (5), with $\text{CE}_\lambda$ defined as follows:

$$\text{CE}_\lambda(G_1, p_{1|t}^{\theta}(\cdot|G_t)) = -\sum_n \log\left(p_{1|t}^{\theta,(n)}(x_1^{(n)}|G_t)\right) - \lambda \sum_{i<j} \log\left(p_{1|t}^{\theta,(ij)}(e_1^{(ij)}|G_t)\right), \quad (6)$$

where $\lambda \in \mathbb{R}^+$ adjusts the weighting of nodes and edge. Empirically, setting $\lambda > 1$ improves generative performance by emphasizing edges to better capture underlying node interactions.

The resulting training and sampling processes are detailed in Algs. 1 and 2.

| **Algorithm 1** DeFoG Training |
|---|
| 1: **Input:** Graph dataset $\mathcal{D} = \{G^1, \ldots, G^M\}$ |
| 2: **while** $f_\theta$ not converged **do** |
| 3:  Sample $G \sim \mathcal{D}$ |
| 4:  Sample $t \sim \mathcal{U}[0,1]$ |
| 5:  Sample $G_t \sim p_{t|1}(G_t|G)$  ▷ Noising |
| 6:  $h \leftarrow \text{RRWP}(G_t)$  ▷ Extra features |
| 7:  $p_{1|t}^{\theta}(G_t) \leftarrow f_\theta(G_t, h, t)$  ▷ Denoising |
| 8:  loss $\leftarrow \text{CE}_\lambda(G, p_{1|t}^{\theta}(\cdot|G_t))$ |
| 9:  optimizer.step(loss) |

| **Algorithm 2** DeFoG Sampling |
|---|
| 1: **Input:** # graphs to sample $S$ |
| 2: **for** $i = 1$ **to** $S$ **do** |
| 3:  Sample $N$ from train set  ▷ # Nodes |
| 4:  Sample $G_0 \sim p_0(G_0)$ |
| 5:  **for** $t = 0$ **to** $1 - \Delta t$ **with step** $\Delta t$ **do** |
| 6:    $h \leftarrow \text{RRWP}(G_t)$  ▷ Extra features |
| 7:    $p_{1|t}^{\theta}(G_t) \leftarrow f_\theta(G_t, h, t)$  ▷ Denoising |
| 8:    $G_{t+\Delta t} \sim \tilde{p}_{t+\Delta t|t}(G_{t+\Delta t}|G_t)$  ▷ Eq. (4) |
| 9:  Store $G_1$ |

**Node Permutation Equivariance**    We now expose DeFoG's equivariance properties in Lemma 1. (See Appendix D.2.2 for proof.)

**Lemma 1** (Node Permutation Equivariance and Invariance Properties of DeFoG). *The DeFoG model is permutation equivariant, its loss function is permutation invariant, and its sampling probability is permutation invariant.*

## 4.2 Exploring DeFoG's Design Space

While discrete diffusion models offer a broader design space compared to one-shot models, including noise schedules and diffusion trajectories (Austin et al., 2021), its exploration is costly. In contrast, flow models enable greater flexibility and efficiency due to their decoupled noising and sampling procedures. For instance, the number of sampling steps and their sizes are not fixed as in discrete-time diffusion (Vignac et al., 2022), and the rate matrix design can be adjusted independently, unlike continuous-time diffusion models (Siraudin et al., 2024; Xu et al., 2024). In this section, we exploit this rich design space and propose key algorithmic improvements to optimize both the training and sampling stages of DeFoG.

### 4.2.1 Training Stage Optimization

**Initial Distribution**    An essential aspect to consider is the choice of the initial distribution. This distribution cannot be arbitrary. To ensure that DeFoG's sampling probabilities are permutation invariant (Lemma 1), the initial distribution must generate all permutations of a graph with equal probability. A suitable distribution is $p_\epsilon = \prod_n p_\epsilon^{\mathcal{X}} \prod_{i<j} p_\epsilon^{\mathcal{E}}$, where $p_\epsilon^{\mathcal{X}}$ and $p_\epsilon^{\mathcal{E}}$ represent shared distributions across nodes and edges, respectively. Campbell et al. (2024) propose two distributions that fit such model: *uniform*, where all states have equal mass, and *masked*, where an additional virtual state, "mask", accumulates all mass at $t = 0$. We also explore additional initial distributions, including the *marginal* distribution (Vignac et al., 2022), which represents the dataset's marginal distribution, and the *absorbing* distribution, which places all mass at the most prevalent state in the dataset.

Empirically, the marginal distribution outperforms the others with a single exception, as detailed further in Appendix C.1. This is supported by two key observations: first, Vignac et al. (2022) demonstrate that, for any given distribution, the closest distribution within the aforementioned class of factorizable initial distributions is the marginal one, thus illustrating its optimality as a prior. Second, the marginal initial distribution preserves the dataset's marginal properties throughout the noising process, maintaining graph-theoretical characteristics like sparsity (Qin et al., 2023). We conjecture that this fact facilitates the denoising task for the graph transformer.

**Train Distortion**    From Alg. 1, line 4, vanilla DeFoG samples $t$ from a uniform distribution. However, in the same vein as adjusting the noise schedule in diffusion models, we can modify this distribution to allow the model to refine its predictions differently across various time regions. We implement this procedure by sampling $t$ from a uniform distribution and applying a *distortion function* $f$ such that $t' = f(t)$. The specific distortions used and their corresponding distributions for $t'$, are detailed in Appendix B.1. In contrast to prior findings in image generation, which suggest that focusing on intermediate time regions is preferable (Esser et al., 2024), we observe that for most graph generation tasks, the best performing distortion functions particularly emphasize $t$ approaching 1. Our key insight is that, as $t$ approaches 1, discrete structures undergo abrupt transitions between states — from 0 to 1 in a one-hot encoding — rather than the smooth, continuous refinements seen in continuous domains. Therefore, later time steps are critical for detecting errors, such as edges breaking planarity or atoms violating molecular valency rules.

### 4.2.2 Sampling Stage Optimization

**Sample Distortion**    In DeFoG's vanilla sampling process, the discretization is performed using equally sized time steps (Alg. 2, line 5). Instead, we propose employing variable step sizes. This adjustment is motivated by the need for more fine-grained control during certain time intervals, for instance, to ensure that cluster structures are properly formed before focusing on intra-cluster refinements, or to prevent edge alterations that could compromise global properties once the overall structure is established. By allocating smaller, more frequent steps to these critical intervals, the generated graph can better capture the true properties of the data. To achieve this, we modify

the evenly spaced step sizes using distortion functions (see Appendix B.1). Although the optimal distortion function is highly dataset-dependent, in Appendix C.2 we propose a method to guide the selection of the distortion function based on the observed training dynamics. Notably, in diffusion models, the design for diffusion steps such as noise schedule is typically identical for training and sampling. However, in flow models, the time distribution for training detailed in Sec. 4.2.1 and the time steps used for sampling can be more flexibly disentangled. In practice, applying distortion only during sampling already yields notable performance improvements.

**Target Guidance**    To ensure robust denoising performance, diffusion and flow matching models are designed to predict the clean data directly and subsequently generate the reverse trajectory based on that prediction (Ho et al., 2020; Lipman et al.; Vignac et al., 2022). Inspired by that, we propose an alternative sampling mechanism by modifying the rate matrices to $R_t(z_t, z_{t+\mathrm{d}t}|z_1) = R_t^*(z_t, z_{t+\mathrm{d}t}|z_1) + R_t^\omega(z_t, z_{t+\mathrm{d}t}|z_1)$ for $z_t \neq z_{t+\mathrm{d}t}$. , such that:

$$R_t^\omega(z_t, z_{t+\mathrm{d}t}|z_1) = \omega \frac{\delta(z_{t+\mathrm{d}t}, z_1)}{\mathcal{Z}_t^{>0} \, p_{t|1}(z_t|z_1)}, \tag{7}$$

with $\omega \in \mathbb{R}^+$. This adjustment biases the transitions toward the clean data state $z_1$. Lemma 10, in Appendix B.2, demonstrates that this modification introduces an $O(\omega)$ violation of the Kolmogorov equation. Consequently, choosing a small value of $\omega$ is experimentally shown to be highly beneficial, while a larger $\omega$ restricts the distribution to regions of high probability, increasing the distance between the generated data and the training data, as indicated in Appendix B.4, Figure 8.

**Stochasticity**    The space of valid rate matrices, i.e., those that satisfy the Kolmogorov equation, is not exhausted by the original formulation of $R_t^*(z_t, z_{t+\mathrm{d}t}|z_1)$. Campbell et al. (2024) investigate this and show that for any rate matrix $R_t^{\mathrm{DB}}$ that satisfies the detailed balance condition, $p_{t|1}(z_t|z_1)R_t^{\mathrm{DB}}(z_t, z_{t+\mathrm{d}t}|z_1) = p_{t|1}(z_{t+\mathrm{d}t}|z_1)R_t^{\mathrm{DB}}(z_{t+\mathrm{d}t}, z_t|z_1)$, the modified rate matrix $R_t^\eta = R_t^* + \eta R_t^{\mathrm{DB}}$, with $\eta \in \mathbb{R}^+$, also satisfies the Kolmogorov equation. Increasing $\eta$ introduces more stochasticity into the trajectory of the denoising process, while different designs of $R_t^{DB}$ encode different priors for preferred transitions between states. This mechanism can be interpreted as a correction mechanism, as it enables transitions back to states that would otherwise be disallowed according to the rate matrix formulation, as described in Appendix B.5. The effect of $\eta$ across different datasets is illustrated in detail in Figure 9, Appendix B.4. To further improve the sampling performance of DeFoG, we also investigate the different formulations of $R^{\mathrm{DB}}$ under the detailed balance condition. Additional details and discussions are provided in Appendix B.3.

**Exact Expectation for Rate Matrix Computation**    The execution of line 8 in Alg. 2 requires computing $\mathbb{E}_{p_{1|t}^{\theta,(d)}(z_1^{(d)}|z_t^{1:D})}\left[R_t^{(d)}(z_t^{(d)}, z_{t+\Delta t}^{(d)}|z_1^{(d)})\right]$. In practice, Campbell et al. (2024) sample $z_1^{(d)} \sim p_{1|t}^{\theta,(d)}(z_1^{(d)}|z_t^{1:D})$ to directly approximate the expectation with $R_t^{(d)}(z_t^{(d)}, z_{t+\Delta t}^{(d)}|z_1^{(d)})$. Although this procedure converges in expectation to the intended value, it introduces more stochasticity into the denoising trajectory compared to computing the exact expectation. Given that the cardinalities of each dimension are relatively small ($X$ and $E$), we explore the exact computation of the expectation. This approach is especially useful in settings where precision is prioritized over diversity, ensuring high confidence in the validity of generated samples, even at the expense of reduced variability.

## 5    THEORETICAL GUARANTEES

In this section, we present novel theoretical results on general multivariate data, which are naturally extendable to our graph-based framework, as introduced in Sec. 4.1. Their complete versions and proofs are available in Appendix D.1. We begin by presenting a theoretical result that further justifies the design choice for the loss function of DFM, and thus of DeFoG.

**Theorem 2** (Simplified - Bounded estimation error of unconditional multivariate rate matrix)**.** *Given $t \in [0,1]$, $z_t^{1:D}, z_{t+\mathrm{d}t}^{1:D} \in \mathcal{Z}^D$ and $z_1^{1:D} \sim p_1(z_1^{1:D})$, there exist constants $C_0, C_1 > 0$ such that the rate matrix estimation error can be upper bounded by:*

$$|R_t(z_t^{1:D}, z_{t+\mathrm{d}t}^{1:D}) - R_t^\theta(z_t^{1:D}, z_{t+\mathrm{d}t}^{1:D})|^2 \le C_0 + C_1 \mathbb{E}_{p_1(z_1^{1:D})}\left[p_{t|1}(z_t^{1:D}|z_1^{1:D}) \sum_{d=1}^D -\log p_{1|t}^{\theta,(d)}(z_1^d|z_t^{1:D})\right]. \tag{8}$$

By taking the expectation over $t \sim \mathcal{U}[0,1]$ and summing over the resulting $z_t^{1:D}$, minimizing the derived upper bound with respect to $\theta$ shown in the right-hand side (RHS) of Eq. (8) corresponds

Table 1: Graph generation performance on the synthetic datasets: Planar, Tree and SBM. We present the results from five sampling runs, each generating 40 graphs, reported as the mean ± standard deviation. Full version in Tab. 7.

| Model | Class | Planar V.U.N. ↑ | Planar Ratio ↓ | Tree V.U.N. ↑ | Tree Ratio ↓ | SBM V.U.N. ↑ | SBM Ratio ↓ |
|---|---|---|---|---|---|---|---|
| Train set | — | 100 | 1.0 | 100 | 1.0 | 85.9 | 1.0 |
| GraphRNN (You et al., 2018) | Autoregressive | 0.0 | 490.2 | 0.0 | 607.0 | 5.0 | 14.7 |
| GRAN (Liao et al., 2019) | Autoregressive | 0.0 | 2.0 | 0.0 | 607.0 | 25.0 | 9.7 |
| SPECTRE (Martinkus et al., 2022) | GAN | 25.0 | 3.0 | — | — | 52.5 | 2.2 |
| DiGress (Vignac et al., 2022) | Diffusion | 77.5 | 5.1 | 90.0 | **1.6** | 60.0 | 1.7 |
| EDGE (Chen et al., 2023) | Diffusion | 0.0 | 431.4 | 0.0 | 850.7 | 0.0 | 51.4 |
| BwR (EDP-GNN) (Diamant et al., 2023) | Diffusion | 0.0 | 251.9 | 0.0 | 11.4 | 7.5 | 38.6 |
| BiGG (Dai et al., 2020) | Autoregressive | 5.0 | 16.0 | 75.0 | 5.2 | 10.0 | 11.9 |
| GraphGen (Goyal et al., 2020) | Autoregressive | 7.5 | 210.3 | 95.0 | 33.2 | 5.0 | 48.8 |
| HSpectre (Bergmeister et al., 2023) | Diffusion | 95.0 | 2.1 | **100.0** | 4.0 | 75.0 | 10.5 |
| GruM (Jo et al., 2024) | Diffusion | 90.0 | 1.8 | — | — | 85.0 | **1.1** |
| CatFlow (Eijkelboom et al., 2024) | Flow | 80.0 | — | — | — | 85.0 | — |
| DisCo (Xu et al., 2024) | Diffusion | 83.6±2.1 | — | — | — | 66.2±1.4 | — |
| Cometh (Siraudin et al., 2024) | Diffusion | **99.5**±0.9 | — | — | — | 75.0±3.7 | — |
| DeFoG (# steps = 50) | Flow | 95.0±3.2 | 3.2±1.1 | 73.5±9.0 | 2.5±1.0 | 86.5±5.3 | 2.2±0.3 |
| DeFoG (# steps = 1,000) | Flow | **99.5**±1.0 | **1.6**±0.4 | 96.5±2.6 | **1.6**±0.4 | **90.0**±5.1 | 4.9±1.3 |

directly to minimizing the DFM loss in Eq. (5). This further reinforces the pertinence of using such loss function, beyond its ELBO maximization motivation as proposed by Campbell et al. (2024).

Next, we demonstrate that the distributional deviation caused by the independent-dimensional Euler step remains bounded, validating its utilization.

**Theorem 3** (Simplified - Bounded deviation of the generated distribution). *Let $p_1$ be the marginal distribution at $t = 1$ of a groundtruth CTMC, $\{z_t^{1:D}\}_{0 \leq t \leq 1}$, and $\tilde{p}_1$ be the marginal distribution at $t = 1$ of its independent-dimensional Euler sampling approximation, with a maximum step size $\Delta t$. Then, the following total variation (TV) bound holds:*

$$\|p_1 - \tilde{p}_1\|_{TV} \leq UZD \ + \ B(ZD)^2\Delta t + \ O(\Delta t), \tag{9}$$

*with $U$ and $B$ representing constant upper bounds for the right-hand side of Eq. (8) and for the denoising process relative to its noising counterpart, respectively, for any $t \in [0, 1]$.*

This result parallels Campbell et al. (2022), who established a similar bound for continuous time discrete diffusion models using $\tau$-leaping for sampling. However, Theorem 3 extends that result to the DFM framework, employing the independent-dimensional Euler method. Specifically, the first term of the upper bound results from the estimation error, i.e., using the neural network approximation $p_{1|t}^{\theta,(d)}(z_1^{(d)}|z_t^{1:D})$ instead of the true $p_{1|t}^{(d)}(z_1^{(d)}|z_t^{1:D})$. As seen from Theorem 2, this term can be bounded. The remaining terms arise from the time discretization and approximated simulation of the CTMC, respectively. Since these terms are $O(\Delta t)$, their impact can be controlled by arbitrarily reducing the step size, ensuring that the generated distribution remains faithful to the ground truth.

## 6 EXPERIMENTS

First, we present DeFoG's performance in generating graphs with diverse topological structures and molecular datasets with rich prior information. A systematic evaluation of the proposed training and sampling algorithms is provided later to demonstrate the efficiency gains. Finally, we extend its efficacy to a real-world application in digital pathology through conditional generation. In each setting, we highlight the **best result**, underline the second-best, and report DeFoG's performance with a reduced number of steps to emphasize its sampling efficiency.

### 6.1 SYNTHETIC GRAPH GENERATION

We evaluate DeFoG using the widely adopted *Planar*, *SBM* (Martinkus et al., 2022), and *Tree* datasets (Bergmeister et al., 2023), along with the associated evaluation methodology. In Tab. 1, we report the proportion of generated graphs that are valid, unique, and novel (V.U.N.), as well as the average ratio of distances between graph statistics of the generated and test sets relative to the train and test sets (Ratio) to assess sample quality.

As shown in Tab. 1, for the Planar dataset, DeFoG achieves the best performance across both metrics. On the Tree dataset, it is only surpassed by HSpectre, which leverages a local expansion

Table 2: Large molecule generation performance. Only iterative denoising-based methods are reported here. Respecive full versions in Tab. 9 (MOSES) and Tab. 10 (Guacamol), Appendix G.2.

| Model | MOSES | | | | | | | Guacamol | | | | |
|---|---|---|---|---|---|---|---|---|---|---|---|---|
| | Val. ↑ | Unique. ↑ | Novelty ↑ | Filters ↑ | FCD ↓ | SNN ↑ | Scaf ↑ | Val. ↑ | V.U. ↑ | V.U.N.↑ | KL div↑ | FCD↑ |
| Training set | 100.0 | 100.0 | 0.0 | 100.0 | 0.01 | 0.64 | 99.1 | 100.0 | 100.0 | 0.0 | 99.9 | 92.8 |
| DiGress (Vignac et al., 2022) | 85.7 | **100.0** | 95.0 | 97.1 | **1.19** | 0.52 | 14.8 | 85.2 | 85.2 | 85.1 | 92.9 | 68.0 |
| DisCo (Xu et al., 2024) | 88.3 | **100.0** | **97.7** | 95.6 | 1.44 | 0.50 | 15.1 | 86.6 | 86.6 | 86.5 | 92.6 | 59.7 |
| Cometh (Siraudin et al., 2024) | 90.5 | 99.9 | 92.6 | **99.1** | 1.27 | 0.54 | 16.0 | 98.9 | 98.9 | 97.6 | 96.7 | 72.7 |
| DeFoG (# steps = 50) | 83.9 | 99.9 | 96.9 | 96.5 | 1.87 | 0.50 | **23.5** | 91.7 | 91.7 | 91.2 | 92.3 | 57.9 |
| DeFoG (# steps = 500) | **92.8** | 99.9 | 92.1 | 98.9 | 1.95 | **0.55** | 14.4 | **99.0** | **99.0** | **97.9** | **97.7** | **73.8** |

procedure particularly well-suited to hierarchical structures like trees. On the SBM dataset, DeFoG attains the highest V.U.N. score and an average ratio close to the optimal. Notably, across all datasets, DeFoG secures second place in 4 out of 6 cases while using only 5% of the sampling steps.

## 6.2 Molecular Graph Generation

Molecular design is a prominent real-world application of graph generation. We evaluate DeFoG's performance on this task using the QM9 (Wu et al., 2018), MOSES (Polykovskiy et al., 2020), and Guacamol (Brown et al., 2019) datasets. For QM9, we follow the dataset split and evaluation metrics from Vignac et al. (2022), presenting the results in Appendix F.1.2, Tab. 8. For the larger MOSES and Guacamol datasets, we adhere to the training setup and evaluation metrics established by Polykovskiy et al. (2020) and Brown et al. (2019), respectively, with results in Tabs. 9 and 10.

As illustrated in Tab. 2, DeFoG outperforms existing diffusion models. It achieving state-of the Validity while preserving high uniqueness on MOSES. On Guacamol, it consistently ranks best, followed by Cometh, another work utilizing a continuous-time framework. Notably, DeFoG approaches the performance of existing diffusion models with only 10% of the sampling steps. This result is further investigated in the following section.

## 6.3 Efficiency Improvement

**Sampling Efficiency** As highlighted in Tabs. 1, 2 and 8, DeFoG attains similar performance, and in some cases even outperforms, several diffusion-based methods with just 5% or 10% of their sampling steps. This efficiency is a result of DeFoG's sampling stage flexibility, whose optimization is enabled due to the flexible and disentangled training-sampling procedure within DFM. To further demonstrate the advantages of sampling optimization, Figure 2a shows the cumulative effect of each optimization step discussed in Sec. 4.2.2. As illustrated, starting from a vanilla DeFoG model, which initially performs slightly worse than DiGress, we sequentially incorporate algorithmic improvements, with each stage indicated by an increasing number of + symbols. Each addition leads to a progressive improvement over the previous stage across both the Planar and QM9 datasets under various numbers of steps for sampling, with the final performance significantly surpassing the vanilla model using only 50 steps on the Planar dataset. While the benefits of each optimization component may vary across datasets, the sampling optimization pipeline remains highly efficient, enabling quick hyperparameter exploration without requiring retraining, which is a significantly more resource-intensive stage (see Appendix F.2). We provide more details on the impact of each sampling optimization strategy across datasets in Appendix B.4.

**Training Efficiency** While our primary focus is on optimizing the sampling process given its computational efficiency, we emphasize that DeFoG's training optimization strategies can further push its performance. To illustrate this, Figure 2b shows the convergence curves for the tree and MOSES datasets. We first observe that, with the same model, leveraging only sampling distortion can further enhance its performance beyond the vanilla implementation. This suggests that optimizing the sampling procedure can be particularly useful in settings where computational resources are limited and the model is undertrained, as detailed in Appendix B.6. Moreover, when an appropriate sample distortion is known (e.g., *polydec* distortion is shown to be particularly preferred across molecular datasets), applying it to both train distortion and sample distortion typically further improves performance. Although DeFoG achieves faster convergence with both distortions, we remark that DiGress also shows good convergence speed on the Tree dataset, due to the well-tuned noise schedule employed for joint training and sampling. The mutual impact of training and sampling distortion is discussed in Appendix C.2. Besides, in the same section, we discuss a heuristic

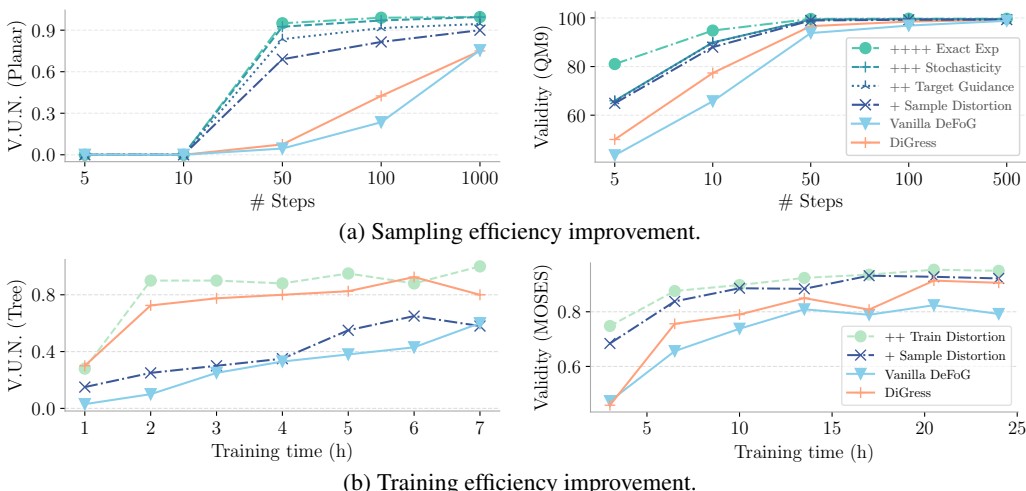

(a) Sampling efficiency improvement.

(b) Training efficiency improvement.

Figure 2: DeFoG's improvements on sampling and training efficiency.

on which distortion to use for each dataset according to DeFoG's vanilla dynamics. Further details on how the initial distribution affects training efficiency are provided in Appendix C.1.

## 6.4 CONDITIONAL GENERATION

**Setup** Tertiary Lymphoid Structure (TLS) graph datasets have been recently released with graphs built from digital pathology data (Madeira et al., 2024). These graphs are split into two subsets — low TLS and high TLS — based on their TLS content, a biologically informed metric reflecting the cell organization in the graph structure. Previously, models were trained and evaluated separately for each subset. To demonstrate the flexibility of DeFoG, we conditionally train it across both datasets simultaneously, using the low/high TLS content as a binary label for each graph. More details on the used conditional framework in Appendix E.

Table 3: TLS conditional generation results.

| | TLS Dataset | |
|---|---|---|
| Model | V.U.N. ↑ | TLS Val. ↑ |
| Train set | 0.0 | 100 |
| GraphGen (Goyal et al., 2020) | 40.2±3.8 | 25.1±1.2 |
| BiGG (Dai et al., 2020) | 0.6±0.4 | 16.7±1.6 |
| SPECTRE (Martinkus et al., 2022) | 7.9±1.3 | 25.3±0.8 |
| DiGress+ (Madeira et al., 2024) | 13.2±3.4 | 12.6±3.0 |
| ConStruct (Madeira et al., 2024) | **99.1**±1.1 | 92.1±1.3 |
| DeFoG (# steps = 50) | 44.5±4.2 | 93.0±5.6 |
| DeFoG (# steps = 1,000) | 94.5±1.8 | **95.8**±1.5 |

We evaluate two main aspects: first, how frequently the conditionally generated graphs align with the provided labels (TLS Validity); and, second, the validity, uniqueness, and novelty of the generated graphs (V.U.N.). Graphs are considered valid if they are planar and connected. For comparison, we report the average results of existing models across the two subsets, as they were not trained conditionally.

**Results** From Tab. 3, DeFoG significantly outperforms the unconstrained models (all but ConStruct). Notably, we outperform ConStruct on TLS validity with even 50 steps. For V.U.N., while ConStruct is hard-constrained to achieve 100% graph planarity, making it strongly biased toward high validity, DeFoG remarkably approaches these values without relying on such rigid constraints.

## 7 CONCLUSION

We introduce DeFoG, a flexible discrete flow-matching framework for graph generation. Our theoretical contributions ensure that the denoising process preserves the graph distribution. Extensive experiments show that DeFoG achieves state-of-the-art results across tasks such as synthetic, molecular, and TLS graph generation, demonstrating real-world applicability and efficiency with just 50 sampling steps. Although limited computational resources restricted a thorough hyperparameter search, leaving room for further optimization, DeFoG's performance highlights its potential. While our work focuses on graphs, the flexibility of DeFoG and the proposed theoretical guarantees suggest that some of our techniques may generalize to other data modalities, opening interesting avenues for future research.

## ETHICS STATEMENT

All authors of this paper have read and adhered to the ICLR Code of Ethics. We explicitly acknowledge our commitment to these ethical guidelines throughout the submission process.

Regarding impact, the primary objective of this paper is to advance graph generation under a more flexible framework, with applications spanning general graph generation, molecular design, and digital pathology. The ability to generate graphs with discrete labels can have broad-reaching implications for fields such as drug discovery and diagnostic technologies. While this development has the potential to bring about both positive and negative societal or ethical impacts, particularly in areas like biomedical and chemical research, we currently do not foresee any immediate societal concerns associated with the proposed methodology.

## REPRODUCIBILITY STATEMENT

All experiments in this paper are fully reproducible, and the code is provided in the supplementary material to facilitate this process. We include detailed descriptions of the datasets, experimental setup and methodologies used in Sec. 6 and appendix F. In particular, the specific hyperparameters employed in our experiments are documented in this paper in Appendix F.3 and in the corresponding code repository. This ensures that researchers can easily replicate our results and build upon our work. The full version and proofs of theoretical results can be found in Appendix D.

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

# Appendix

## Table of Contents

## A    CONTEXTUALIZING RELATED RESEARCH

In this section, we further contextualize DeFoG within the scope of related work. We begin by introducing the methods used for comparison with DeFoG in Appendix A.1. Subsequently, we outline the key distinctions between DeFoG and existing diffusion-based graph generative models in Appendix A.2.

### A.1    OVERVIEW OF COMPARED METHODS

In Sec. 6, we evaluate DeFoG against a diverse set of graph generative models, which we introduce below:

- GraphRNN (You et al., 2018) and GRAN (Liao et al., 2019), two pioneering autoregressive models for graph generation;
- SPECTRE (Martinkus et al., 2022), a spectrally conditioned GAN-based model for graph generation;
- DiGress (Vignac et al., 2022), the first discrete diffusion model for graph generation;
- EDGE (Chen et al., 2023), a discrete diffusion model leveraging graph sparsity and degree guidance for scalability.
- BwR (Diamant et al., 2023), which focuses on efficient graph representations via bandwidth restriction schemes that are compatible with various graph generation models. We report its results in combination with EDP-GNN (Niu et al., 2020), which was the first graph diffusion model;
- BiGG (Dai et al., 2020), an autoregressive model that exploits graph sparsity and training parallelization to scale to larger graphs;
- GraphGen (Goyal et al., 2020), a scalable autoregressive approach utilizing graph canonization with minimum DFS codes, notable for being domain-agnostic and inherently supporting attributed graphs;
- HSpectre (Bergmeister et al., 2023), a hierarchical graph generation method that utilizes a score-based formulation for iterative local expansion steps;
- DisCo (Xu et al., 2024) and Cometh (Siraudin et al., 2024), two continuous-time discrete diffusion models for graph generation;
- GruM (Jo et al., 2024), which employs a diffusion mixture to explicitly learn the final graph topology and structure;
- CatFlow (Eijkelboom et al., 2024), which results from the instantiation of variational flow matching to graph generation.

### A.2    DEFOG AND GRAPH DIFFUSION MODELS

In this section, we contextualize DeFoG in relation to existing graph diffusion models.

#### A.2.1    FROM CONTINUOUS TO DISCRETE STATE-SPACES

Early diffusion-based graph generative models extended continuous diffusion and score-based methods from image generation to graphs by relaxing adjacency matrices into continuous state-spaces (Niu et al., 2020; Jo et al., 2022). However, this approach overlooks the inherent discreteness of graph-structured data, resulting in topologically uninformed noising processes. For instance, these methods often destroy graph sparsity and generate noisy complete graphs (Vignac et al., 2022; Xu et al., 2024; Siraudin et al., 2024), making it more challenging for denoising neural networks to recover meaningful structural properties from the noisy inputs. Some recent formulations operating on continuous state-spaces have tried to overcome these limitations: GruM (Jo et al., 2024) introduces an endpoint-conditioned diffusion mixture strategy to enhance accuracy by explicitly learning final graph structures, while CatFlow (Eijkelboom et al., 2024) proposes variational flow matching to handle categorical data more effectively.

Alternatively, discrete diffusion models have emerged as a more natural solution, directly preserving the discrete nature of graph data (Vignac et al., 2022; Haefeli et al.). These models have demonstrated state-of-the-art performance across a variety of applications, including neural architecture search (Asthana et al., 2024), combinatorial optimization (Sun & Yang, 2024), molecular generation (Irwin et al., 2024), and reaction pathway design (Igashov et al., 2024).

DeFoG aligns with this second family of methods, modeling nodes and edges in discrete state-spaces to leverage the structural properties of graph data effectively.

### A.2.2 FROM DISCRETE TO CONTINUOUS TIME

The initial discrete-time diffusion frameworks for graph generation (Vignac et al., 2022; Haefeli et al.) were built upon Discrete Denoising Diffusion Probabilistic Models (D3PMs) (Austin et al., 2021), which operate with a fixed partitioning of time. This discretization constrains the model to denoise at specific time points and ties the sampling process to the same fixed time steps used during training, leading to a rigid coupling between the training and sampling stages. Such inflexibility in time discretization can limit the quality of generated graphs.

In contrast, continuous-time discrete diffusion frameworks (Campbell et al., 2022; Sun et al., 2022) overcome these limitations by enabling the model to denoise at arbitrary time points within a continuous interval (typically between 0 and 1). This flexibility allows the time discretization strategy for sampling to be selected post-training, enabling the use of advanced sampling techniques (Jolicoeur-Martineau et al., 2021; Zhang & Chen, 2022; Salimans & Ho, 2022; Chung et al., 2022; Song et al., 2020; Dockhorn et al., 2021) to improve generation performance. These continuous-time frameworks have been successfully extended to graph generative models (Xu et al., 2024; Siraudin et al., 2024), achieving notable improvements.

DeFoG follows a continuous-time formulation, leveraging its flexibility in sampling to achieve enhanced performance while maintaining the strengths of discrete state-space modeling.

### A.2.3 FROM CONTINUOUS-TIME DISCRETE DIFFUSION TO DISCRETE FLOW MATCHING

While both continuous-time discrete diffusion and discrete flow matching (DFM) share the CTMC formulation for the denoising process, they differ fundamentally in the formulation of the noising process. Continuous-time discrete diffusion-based graph generative models (Xu et al., 2024; Siraudin et al., 2024) define the noising process as a CTMC, akin to the denoising process. However, this approach imposes two significant limitations:

1. **Incomplete Coupling of Training and Sampling**: The rate matrices of the noising and denoising processes are explicitly interrelated, and the noising rate matrix must be fixed during training. This restricts the sampling stage, preventing full decoupling of training and sampling.

2. **Limited Design Space**: The noising process must be derived analytically, which is not straightforward and is only feasible for rate matrices suitable for matrix exponentiation. Additionally, the denoising rate matrix is implicitly defined during training, constraining the flexibility of the denoising trajectory at sampling time (e.g., fixing the level of stochasticity).

In contrast, DeFoG allows for direct prescription of the noising process, $p_{t|1}$, without these constraints. The rate matrix for the denoising process is selected exclusively at sampling time, fully decoupling the training and sampling stages. This flexibility enables performance optimization during sampling, such as tuning the stochasticity of the denoising trajectory via $R_t^{\mathrm{DB}}$ or adjusting target guidance magnitude with $R_t^\omega$.

The benefits of this decoupled framework are evident in Figures 2, 6 and 7, which demonstrate that the vanilla DeFoG configuration alone does not outperform existing diffusion-based graph generative models. However, our extensive sampling optimization pipeline capitalizes on DeFoG's flexible design space to achieve state-of-the-art results. These observations align with findings in iterative refinement methods across other data modalities. For instance, Karras et al. (2022) elaborate on the benefits of stochasticity adjustment in denoising trajectories within diffusion models for image generation.

For a comprehensive discussion of the differences between continuous-time discrete diffusion and DFM frameworks, see Appendix H of (Campbell et al., 2024).

# B  SAMPLE OPTIMIZATION

In this section, we explore the proposed sampling optimization in more detail. We start by analysing the different time distortion functions in Appendix B.1. Next, in Appendix B.2, we prove that the proposed target guidance mechanism actually satisfies the Kolmogorov equation, thus yielding valid rate matrices and, in Appendix B.3, we provide more details about the detailed balance equation and how it widens the design space of rate matrices. In Appendix B.4, we also describe the adopted sampling optimization pipeline. Finally, in Appendix B.5, we provide more details to better clarify the dynamics of the sampling process.

## B.1  TIME DISTORTION FUNCTIONS

In Sec. 4, we explore the utilization of different *distortion functions*, i.e., functions that are used to transform time. The key motivation for employing such functions arises from prior work on flow matching in image generation, where skewing the time distribution during training has been shown to significantly enhance empirical performance (Esser et al., 2024). In practical terms, this implies that the model is more frequently exposed to specific time intervals. Mathematically, this transformation corresponds to introducing a time-dependent re-weighting factor in the loss function, biasing the model to achieve better performance in particular time ranges.

In our case, we apply time distortions to the probability density function (PDF) by introducing a function $f$ that transforms the original uniformly sampled time $t$, such that $t' = f(t)$ for $t \in [0, 1]$. These time distortion functions must satisfy certain conditions: they must be monotonic, with $f(0) = 0$ and $f(1) = 1$. Although the space of functions that satisfy these criteria is infinite, we focus on five distinct functions that yield fundamentally different profiles for the PDF of $t'$. Our goal is to gain intuition about which time ranges are most critical for graph generation and not to explore that function space exhaustively. Specifically:

- *Polyinc*: $f(t) = t^2$, yielding a PDF that decreases monotonically with $t'$;
- *Cos*: $f(t) = \frac{1-\cos \pi t}{2}$, creating a PDF with high density near the boundaries $t' = 0$ and $t' = 1$, and low for intermediate $t'$;
- *Identity*: $f(t) = t$, resulting in a uniform PDF for $t' \in [0, 1]$;
- *Revcos*: $f(t) = 2t - \frac{1-\cos \pi t}{2}$, leading to high PDF density for intermediate $t'$ and low density at the extremes $t' = 0$ and $t' = 1$;
- *Polydec*: $f(t) = 2t - t^2$, where the PDF increases monotonically with $t'$.

The PDF resulting from applying a monotonic function $f$ to a random variable $t$ is given by:

$$\phi_{t'}(t') = \phi_t(t) \left| \frac{\mathrm{d}}{\mathrm{d}t'} f^{-1}(t') \right|,$$

where $\phi_t(t)$ and $\phi_{t'}(t')$ denote the PDFs of $t$ and $t'$, respectively. In our case, $\phi_t(t) = 1$ for $t \in [0, 1]$. The distortion functions and their corresponding PDFs are illustrated in Figure 3.

One of the strategies the proposed in **sampling** optimization procedure is the use of variable step sizes throughout the denoising process. This is achieved by mapping evenly spaced time points (DeFoG's vanilla version) through a transformation that follows the same constraints as the training time distortions discussed earlier. We employ the same set of time distortion functions, again not to exhaustively explore the space of applicable functions, but to gain insight into how varying step sizes affect graph generation. The expected step sizes for each distortion can be directly inferred from Figure 3. For instance, the polydec function leads to progressively smaller time steps, suggesting more refined graph edits in the denoising process as $t'$ approaches 1.

Note that even though we apply the same time distortions for both training and sample stages, in each setting they have different objectives: in training, the time distortions skew the PDFs from where $t'$ is sampled, while in sampling they vary the step sizes.

## B.2  TARGET GUIDANCE

In this section, we demonstrate that the proposed *target guidance* design for the rate matrices violates the Kolmogorov equation with an error that is linear in $\omega$. This result indicates that a small guidance

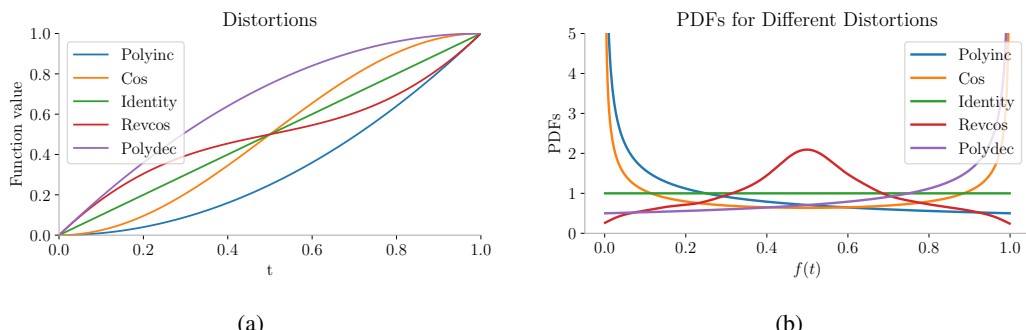

(a)             (b)

Figure 3: (a) The five distortion functions explored. (b) The resulting PDFs for the five distortion functions. For polydec, identity, and polyinc, they were computed in closed-form. For revcos and cos, they were simulated with $10^4$ repetitions.

factor effectively helps fit the distribution, whereas a larger guidance factor, as shown in Figure 8, while enhancing topological properties such as planarity, increases the distance between generated and training data on synthetic datasets according to the metrics of average ratio. Similarly, for molecular datasets, this also leads to an increase in validity and a decrease in novelty by forcing the generated data to closely resemble the training data.

**Lemma 10** (Rate matrices for target guidance). *Let $R_t^\omega(z_t, z_{t+\mathrm{d}t}|z_1)$ be defined as:*

$$R_t^\omega(z_t, z_{t+\mathrm{d}t}|z_1) = \omega \frac{\delta(z_1, z_{t+\mathrm{d}t})}{Z_t^{>0} \ p_{t|1}(z_t|z_1)}. \tag{10}$$

*Then, the univariate rate matrix $R_t^{\mathrm{TG}}(z_t, z_{t+\mathrm{d}t}|z_1) = R_t^*(z_t, z_{t+\mathrm{d}t}|z_1) + R_t^\omega(z_t, z_{t+\mathrm{d}t}|z_1)$ violates the Kolmogorov equation with an error of $-\frac{\omega}{Z_t^{>0}}$ when $z_t \neq z_1$, and an error of $\omega \frac{Z_t^{>0}-1}{Z_t^{>0}}$ when $z_t = z_1$.*

*Proof.* In the remaining of the proof, we consider the case $z_t \neq z_1$. We consider the same assumptions as Campbell et al. (2024):

- $p_{t|1}(z_t|z_1) = 0 \Rightarrow R_t^*(z_t, z_{t+\mathrm{d}t}|z_1) = 0$;

- $p_{t|1}(z_t|z_1) = 0 \Rightarrow \partial_t p_{t|1}(z_t|z_1) = 0$ ("dead states cannot ressurect").

The $z_1$-conditioned Kolmogorov equation is given by:

$$\partial_t p_{t|1}(z_t|z_1) = \sum_{z_{t+\mathrm{d}t} \neq z_t} R_t(z_{t+\mathrm{d}t}, z_t|z_1)p_{t+\mathrm{d}t|1}(z_{t+\mathrm{d}t}|z_1) - \sum_{z_{t+\mathrm{d}t} \neq z_t} R_t(z_t, z_{t+\mathrm{d}t}|z_1)p_{t|1}(z_t|z_1) \tag{11}$$

We denote by RHS and LHS the right-hand side and left-hand side, respectively, of Eq. (11). For the case in which $p_{t|1}(z_t|z_1) > 0$, we have:

$$\text{RHS} = \sum_{z_{t+dt} \neq z_t, p_{t+dt|1}(z_{t+dt}|z_1)>0} (R_t^*(z_{t+dt}, z_t|z_1) + R_t^\omega(z_{t+dt}, z_t|z_1))p_{t+dt|1}(z_{t+dt}|z_1)$$

$$- \sum_{z_{t+dt} \neq z_t, p_{t+dt|1}(z_{t+dt}|z_1)>0} (R_t^*(z_t, z_{t+dt}|z_1) + R_t^\omega(z_t, z_{t+dt}|z_1))p_{t|1}(z_t|z_1)$$

$$= \sum_{z_{t+dt} \neq z_t, p_{t+dt|1}(z_{t+dt}|z_1)>0} R_t^*(z_{t+dt}, z_t|z_1)p_{t+dt|1}(z_{t+dt}|z_1) - R_t^*(z_t, z_{t+dt}|z_1)p_{t|1}(z_t|z_1)$$

$$+ \sum_{z_{t+dt} \neq z_t, p_{t+dt|1}(z_{t+dt}|z_1)>0} R_t^\omega(z_{t+dt}, z_t|z_1)p_{t+dt|1}(z_{t+dt}|z_1) - R_t^\omega(z_t, z_{t+dt}|z_1)p_{t|1}(z_t|z_1),$$

For the first sum, we have:

$$\sum_{z_{t+dt} \neq z_t, p_{t+dt|1}(z_{t+dt}|z_1)>0} R_t^*(z_{t+dt}, z_t|z_1)p_{t+dt|1}(z_{t+dt}|z_1) - R_t^*(z_t, z_{t+dt}|z_1)p_{t|1}(z_t|z_1)$$

$$= \partial_t p_{t|1}(z_t|z_1).$$

since the $z_1$-conditioned $R_t^*$ generates $p_{t|1}$.

For the second sum, we have:

$$\sum_{z_{t+dt} \neq z_t, p_{t+dt|1}(z_{t+dt}|z_1)>0} R_t^\omega(z_{t+dt}, z_t|z_1)p_{t+dt|1}(z_{t+dt}|z_1) - R_t^\omega(z_t, z_{t+dt}|z_1)p_{t|1}(z_t|z_1) =$$

$$= \sum_{z_{t+dt} \neq z_t, p_{t+dt|1}(z_{t+dt}|z_1)>0} \omega \frac{\delta(z_1, z_t)}{Z_t^{>0}} \frac{1}{p_{t+dt|1}(z_{t+dt}|z_1)} p_{t+dt|1}(z_{t+dt}|z_1)$$

$$- \sum_{z_{t+dt} \neq z_t, p_{t+dt|1}(z_{t+dt}|z_1)>0} \omega \frac{\delta(z_1, z_{t+dt})}{Z_t^{>0}} \frac{1}{p_{t|1}(z_t|z_1)} p_{t|1}(z_t|z_1)$$

$$= \frac{\omega}{Z_t^{>0}} \sum_{z_{t+dt} \neq z_t, p_{t+dt|1}(z_{t+dt}|z_1)>0} (\delta(z_1, z_t) - \delta(z_1, z_{t+dt}))$$

If $z_1 \neq z_t$, we have:

$$\sum_{z_{t+dt} \neq z_t, p_{t+dt|1}(z_{t+dt}|z_1)>0} R_t^\omega(z_{t+dt}, z_t|z_1)p_{t+dt|1}(z_{t+dt}|z_1) - R_t^\omega(z_t, z_{t+dt}|z_1)p_{t|1}(z_t|z_1) =$$

$$= \frac{\omega}{Z_t^{>0}} \sum_{z_{t+dt} \neq z_t, p_{t+dt|1}(z_{t+dt}|z_1)>0} (\delta(z_1, z_t) - \delta(z_1, z_{t+dt}))$$

$$= \frac{\omega}{Z_t^{>0}} \sum_{z_{t+dt} \neq z_t, p_{t+dt|1}(z_{t+dt}|z_1)>0} (0 - \delta(z_1, z_{t+dt}))$$

$$= -\frac{\omega}{Z_t^{>0}},$$

Here, we apply the property that $z_t \neq z_1$, which indicates that $\delta(z_1, z_t) = 0$ and that there exists one and only one $z_{t+dt} \in \{z_{t+dt}, z_{t+dt} \neq z_t\}$ such that $z_{t+dt} = z_1$, which verifies that $p_{t+dt|1}(z_{t+dt}|z_1) > 0$ (a condition satisfied by any initial distribution proposed in this work when $t$ strictly positive) the sum simplifies to $-\frac{\omega}{Z_t^{>0}}$.

If $z_1 = z_t$, we have:

$$\sum_{z_{t+dt} \neq z_t, p_{t+dt|1}(z_{t+dt}|z_1) > 0} R_t^\omega(z_{t+dt}, z_t|z_1) p_{t+dt|1}(z_{t+dt}|z_1) - R_t^\omega(z_t, z_{t+dt}|z_1) p_t(z_t|z_1) =$$

$$= \frac{\omega}{Z_t^{>0}} \sum_{z_{t+dt} \neq z_t, p_{t+dt|1}(z_{t+dt}|z_1) > 0} (\delta(z_1, z_t) - \delta(z_1, z_{t+dt}))$$

$$= \frac{\omega}{Z_t^{>0}} \sum_{z_{t+dt} \neq z_t, p_{t+dt|1}(z_{t+dt}|z_1) > 0} (1 - 0)$$

$$= \omega \frac{Z_t^{>0} - 1}{Z_t^{>0}},$$

$\square$

**Intuition** The aim of *target guidance* is to reinforce the transition rate to the state predicted by the probabilistic model, $z_1$. The $\omega$ term is an hyperparameter used to control the target guidance magnitude.

### B.3 DETAILED BALANCE, PRIOR INCORPORATION, AND STOCHASTICITY

Campbell et al. (2024) show that although their $z_1$-conditional formulation of $R_t^*$ generates $p_{t|1}$, it does not span the full space of valid rate matrices — those that satisfy the conditional Kolmogorov equation (Eq. (11)). They derive sufficient conditions for identifying other valid rate matrices. Notably, they demonstrate that matrices of the form

$$R_t^\eta := R_t^* + \eta R_t^{\mathrm{DB}},$$

with $\eta \in \mathbb{R}^{\geq 0}$ and $R_t^{\mathrm{DB}}$ any matrix that verifies the *detailed balance condition*:

$$p_{t|1}(z_t|z_1) R_t^{\mathrm{DB}}(z_t, z_{t+dt}|z_1) = p_{t|1}(z_{t+dt}|z_1) R_t^{\mathrm{DB}}(z_{t+dt}, z_t|z_1), \tag{12}$$

still satisfy the Kolmogorov equation. The detailed balance condition ensures that the outflow, $p_{t|1}(z_t|z_1) R_t^{\mathrm{DB}}(z_t, z_{t+dt}|z_1)$, and inflow, $p_{t|1}(z_{t+dt}|z_1) R_t^{\mathrm{DB}}(z_{t+dt}, z_t|z_1)$, of probability mass to any given state are perfectly balanced. Under these conditions, this additive component's contribution to the Kolmogorov equation becomes null (similar to the target guidance, as shown in the proof of of Lemma 10, in Appendix B.2).

A natural question is how to choose a suitable design for $R_t^{\mathrm{DB}}$ from the infinite space of detailed balance rate matrices. As depicted in Figure 4, this flexibility can be leveraged to incorporate priors into the denoising model by encouraging specific transitions between states. By adjusting the sparsity of the matrix entries, additional transitions beyond those prescribed by $R_t^*$ can be introduced. In the general case, transitions between all states are possible; in the column case, a specific state centralizes all potential transitions; and in the single-entry case, only transitions between two states are permitted. These examples merely illustrate some possibilities and do not exhaust the range of potential $R_t^{\mathrm{DB}}$ designs. The matrix entries can be structured by considering the following reorganization of terms of Eq. (12):

$$R_t^{\mathrm{DB}}(z_{t+dt}, z_t|z_1) = \frac{p_{t|1}(z_t|z_1)}{p_{t|1}(z_{t+dt}|z_1)} R_t^{\mathrm{DB}}(z_t, z_{t+dt}|z_1).$$

Therefore, a straightforward approach is to assign the lower triangular entries of the rate matrix as $R_t^{\mathrm{DB}}(z_t, z_{t+dt}|z_1) = p_{t|1}(z_{t+dt}|z_1)$, and the upper triangular entries as $R_t^{\mathrm{DB}}(z_{t+dt}, z_t|z_1) = p_{t|1}(z_t|z_1)$. The diagonal entries are computed last to ensure that $R_t(z_t, z_t) = -\sum_{z_{t+dt} \neq z_t} R_t(z_t, z_{t+dt})$.

We incorporated various types of priors into $R^{\mathrm{DB}}$ by preserving specific rows or entries in the matrix. Specifically, we experimented with retaining the column corresponding to the state with the highest marginal distribution (Column - Max Marginal), the column corresponding to the predicted $x_1$ states (Column - $x_1$), and the columns corresponding to the state with the highest probability in $p_{t|1}$. Additionally, we tested the approach of retaining only $R^{\mathrm{DB}}(x_t, i)$ where $i$ is the state with the highest

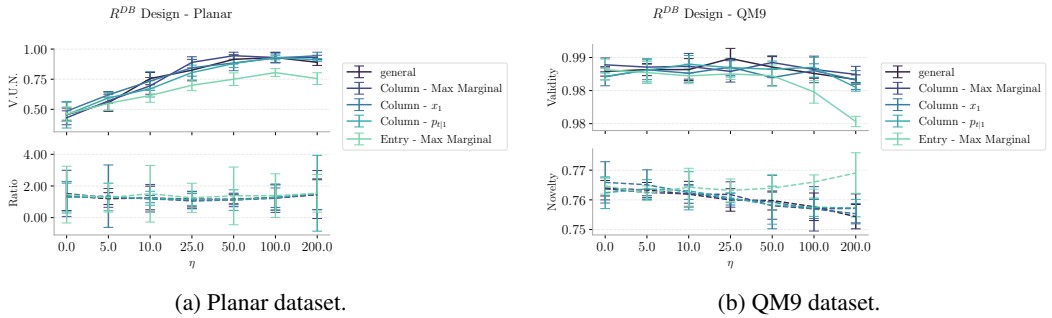

Figure 4: Examples of different rate matrices from the space of $3 \times 3$ matrices that satisfy the detailed balance condition. Here $p_i$ denotes $p_{t|1}(i|z_1)$.

Figure 5: Impact of $R^{\mathrm{DB}}$ with different level of sparsity.

(a) Planar dataset.

(b) QM9 dataset.

marginal distribution (Entry - Max Marginal). For instance, under the absorbing initial distribution, this state is the one to which all data is absorbed at $t = 0$. We note that there remains significant space for exploration by adjusting the weights assigned to different positions within $R^{\mathrm{DB}}$, as the only condition that must be satisfied is that symmetrical positions adhere to a specific proportionality. However, in practice, none of the specific designs illustrated in Figure 4 showed a clear advantage over others in the settings we evaluated. As a result, we chose the general case for our experiments, as it offers the most flexibility by incorporating the least prior knowledge.

Orthogonal to the design of $R_t^{\mathrm{DB}}$, we must also consider the hyperparameter $\eta$, which regulates the magnitude of stochasticity in the denoising process. Specifically, setting $\eta = 0$ (thereby relying solely on $R_t^*$) minimizes the expected number of jumps throughout the denoising trajectory under certain conditions, as shown by Campbell et al. (2024) in Proposition 3.4. However, in continuous diffusion models, some level of stochasticity has been demonstrated to enhance performance (Karras et al., 2022; Cao et al., 2023; Xu et al., 2023). Conversely, excessive stochasticity can negatively impact performance. Campbell et al. (2024) propose that there exists an optimal level of stochasticity that strikes a balance between exploration and accuracy. In our experiments, we observed varied behaviors as $\eta$ increases, resulting in different performance outcomes across datasets, as illustrated in Figure 9.

### B.4 HYPERPARAMETER OPTIMIZATION PIPELINE

A significant advantage of flow matching methods is their inherently greater flexibility in the sampling process compared to diffusion models, as they are more disentangled from the training stage. Each of the proposed optimization strategies exposed in Sec. 4.2.2 expands the search space for optimal performance. However, conducting a full grid search across all those methodologies is impractical for the computational resources available. To address this challenge, our sampling optimization pipeline consists of, for each of the proposed optimization strategies, all hyperparameters are held constant at their default values except for the parameter controlling the chosen strategy, over which we perform a sweep. The optimal values obtained for each strategy are combined to form the final configuration. In Tab. 6, we present the final hyperparameter values obtained for each dataset. This pipeline is sufficient to achieve state-of-the-art performance, which reinforces the ex-

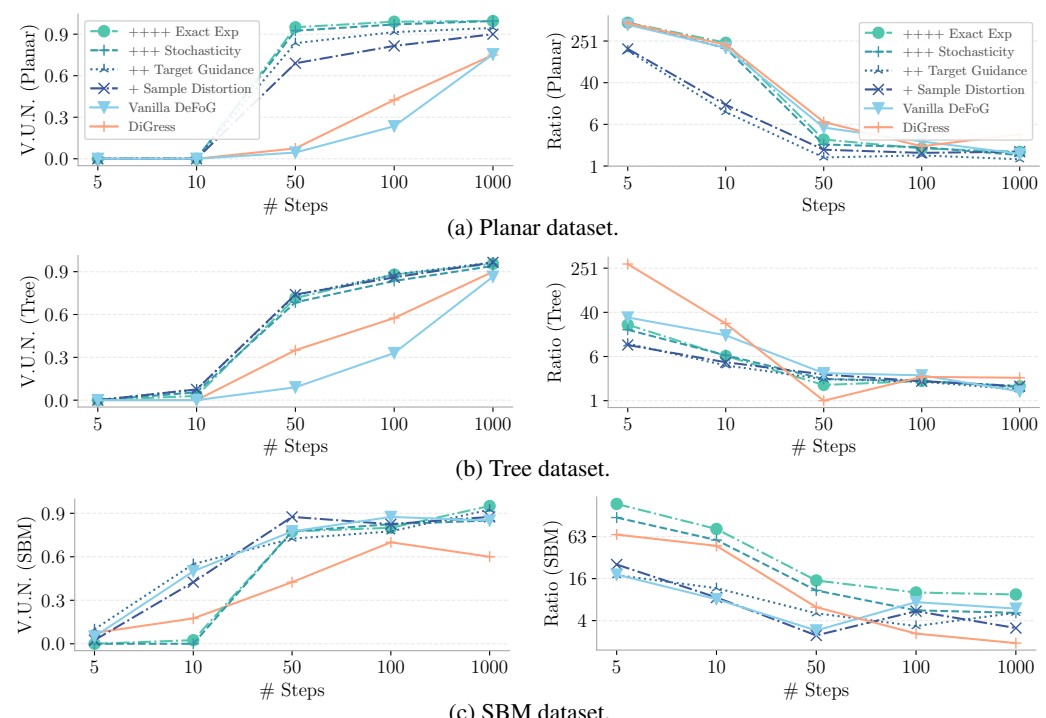

Figure 6: Sampling efficiency improvement over all synthetic datasets.

pressivity of DeFoG. We expect to achieve even better results if a more comprehensive search of the hyperparameter space was carried out.

To better mode detailedly illustrate the influence of each sampling optimization, we show in Figures 6 and 7 in present the impact of varying parameter values across the synthetic datasets Figure 6 and molecular datasets in Figure 7 used in this work.

While Figures 6 and 7 are highly condensed, we provide a more fine-grained version that specifically illustrates the influence of the hyperparameters $\eta$ and $\omega$. This version highlights their impact when generating with the full number of steps (500 and 1000 for molecular and synthetic data, respectively) and with 50 steps. As emphasized in Figures 8 and 9, the influence of these hyperparameters varies across datasets and exhibits distinct behaviors depending on the number of steps used.

Several key observations can be made here. First, since the stochasticity is designed around the detailed balance condition, which holds more rigorously with increased precision, it generally provides greater benefits with the full generation steps but leads to a more pronounced performance decrease when generating with only 50 steps. Additionally, for datasets such as Planar, MOSES, and Guacamol, the stochasticity shows an increasing-then-decreasing behavior, indicating the presence of an optimal value. Furthermore, while target guidance significantly improves validity across different datasets, it can negatively affect novelty and the average ratio when set too high. This suggests that excessive target guidance may promote overfitting to high-probability regions of the training set, distorting the overall distribution. In conclusion, each hyperparameter should be carefully chosen based on the specific objective.

To demonstrate the benefit of each designed optimization step, we report the step-wise improvements by sequentially adding each tuned step across the primary datasets—synthetic datasets in Figure 10 and molecular datasets in Figure 11—used in this work.

## B.5 UNDERSTANDING THE SAMPLING DYNAMICS OF $R_t^*$

In this section, we aim to provide deeper intuition into the sampling dynamics imposed by the design of $R_t^*$, as proposed by Campbell et al. (2024). The explicit formulation of $R_t^*$ can be found in Eq. (3).

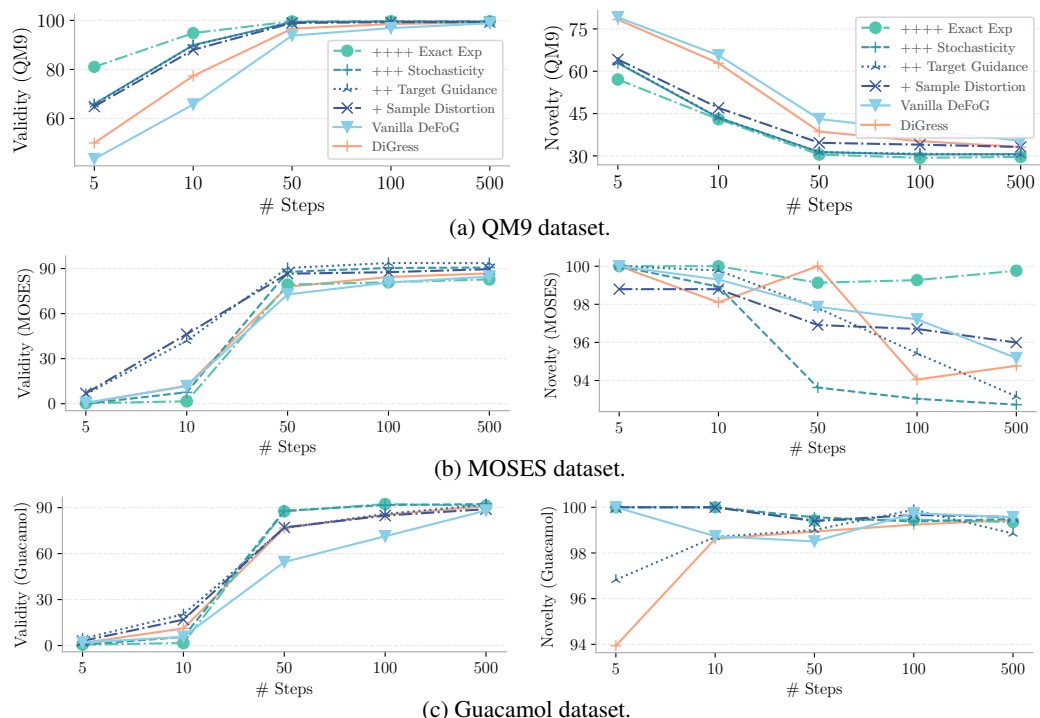

Figure 7: Sampling efficiency improvement over all molecular datasets. The Guacamol and MOSES datasets are evaluated with 2,000 samples, and validity and novelty are computed using local implementations for efficiency instead of the original benchmarks. For consistency across the three datasets, we apply a validity metric that ignores charged molecules. The values for Guacamol differ from Tab. 2, as its benchmark accommodates charged molecules, leading to higher reported validity.

Notably, the denominator in the expression serves as a normalizing factor, meaning the dynamics of each sampling step are primarily influenced by the values in the numerator. Specifically, we observe the following relationship:

$$\partial_t p_{t|1}(z_t|z_1) = \delta(z_t, z_1) + p_0(z_t),$$

derived by directly differentiating Eq. (1). Based on this, the possible values of $R_t^*$ for different combinations of $z_t$ and $z_{t+dt}$ are outlined in Tab. 4.

Table 4: Values of $R_t^*$ for different $z_t$ and $z_{t+dt}$.

| CONDITION | $R^*(x_t, j|z_1)$ | INTUITION |
|---|---|---|
| $z_t = z_1, z_{t+dt} = z_1$ | $\mathrm{ReLU}(p_0(z_1) - p_0(z_1)) = 0$ | NO TRANSITION |
| $z_t = z_1, z_{t+dt} \neq z_1$ | $\mathrm{ReLU}(p_0(z_t) - 1 - p_0(z_{t+dt})) = 0$ | NO TRANSITION |
| $z_t \neq z_1, z_{t+dt} = z_1$ | $\mathrm{ReLU}(p_0(z_t) - p_0(z_{t+dt}) + 1) > 0$ | TRANSITION TO $z_1$ |
| $z_t \neq z_1, z_{t+dt} \neq z_1$ | $\mathrm{ReLU}(p_0(z_t) - p_0(z_{t+dt}))$ | TRANSITION TO $z_{t+dt}$ IF $p_0(z_t) > p_0(z_{t+dt})$ |

From the first two lines of Tab. 4, we observe that once the system reaches the predicted state $z_1$, it remains there. If not, $R_t^*$ only encourages transitions to other states under two conditions: either the target state is $z_1$ (third line), or the corresponding entries in the initial distribution for potential next states have smaller values than the current state (fourth line). As a result, the sampling dynamics are heavily influenced by the initial distribution, as discussed further in Appendix C.1.

For instance, with the masking distribution, the fourth line facilitates transitions to states other than the virtual "mask" state, whereas for the uniform distribution, no transitions are allowed. For the marginal distribution, transitions are directed toward less likely states. Note that while these behaviors hold when the rate matrix consists solely of $R_t^*$, additional transitions can be introduced through $R_t^{DB}$ (as detailed in Appendix B.3) or by applying target guidance (see Appendix B.2).

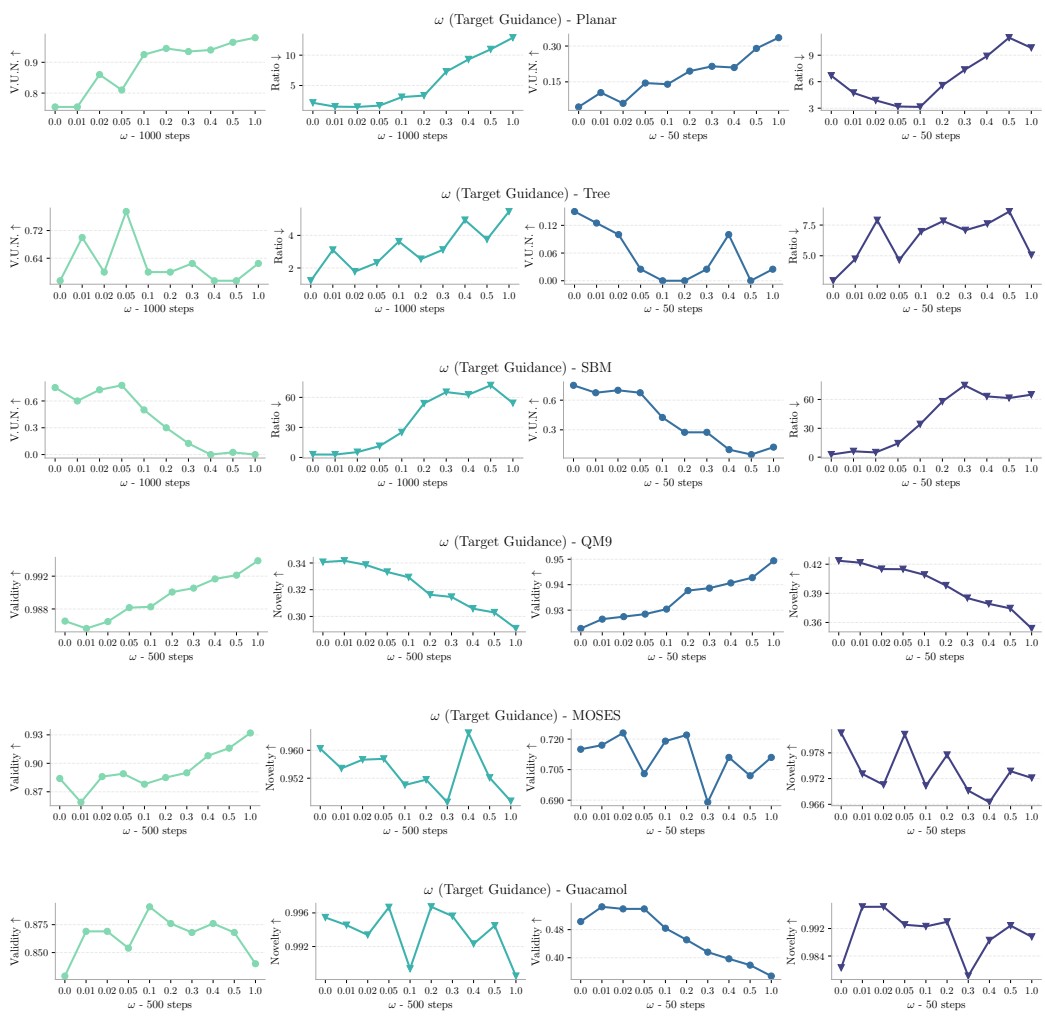

Figure 8: Influence of target guidance over all datasets.

## B.6 PERFORMANCE IMPROVEMENT FOR UNDERTRAINED MODELS

In this section, we present the performance of a model trained on the QM9 dataset and the Planar dataset using only 30% of the epochs compared to the final model being reported. We employ the same hyperparameters with Tab. 8 and Tab. 1 for the sampling setup, as reported in Tab. 6.

Compared to fully trained models, our model achieves 99.0 validity (vs. 99.3) and 96.4 uniqueness (vs. 96.3) on the QM9 dataset. For the Planar dataset, it attains 95.5 validity (vs. 99.5) and an average ratio of 1.4 (vs. 1.6). These results demonstrate that, even with significantly fewer training epochs, the model maintains competitive performance under a well-designed sampling procedure, although extended training can still further improve performance. Notably, all metrics surpass the discrete-time diffusion benchmark DiGress (Vignac et al., 2022). As a result, the optimization in the sampling stage proves particularly beneficial when computational resources are limited, by enhancing the performance of an undertrained model.

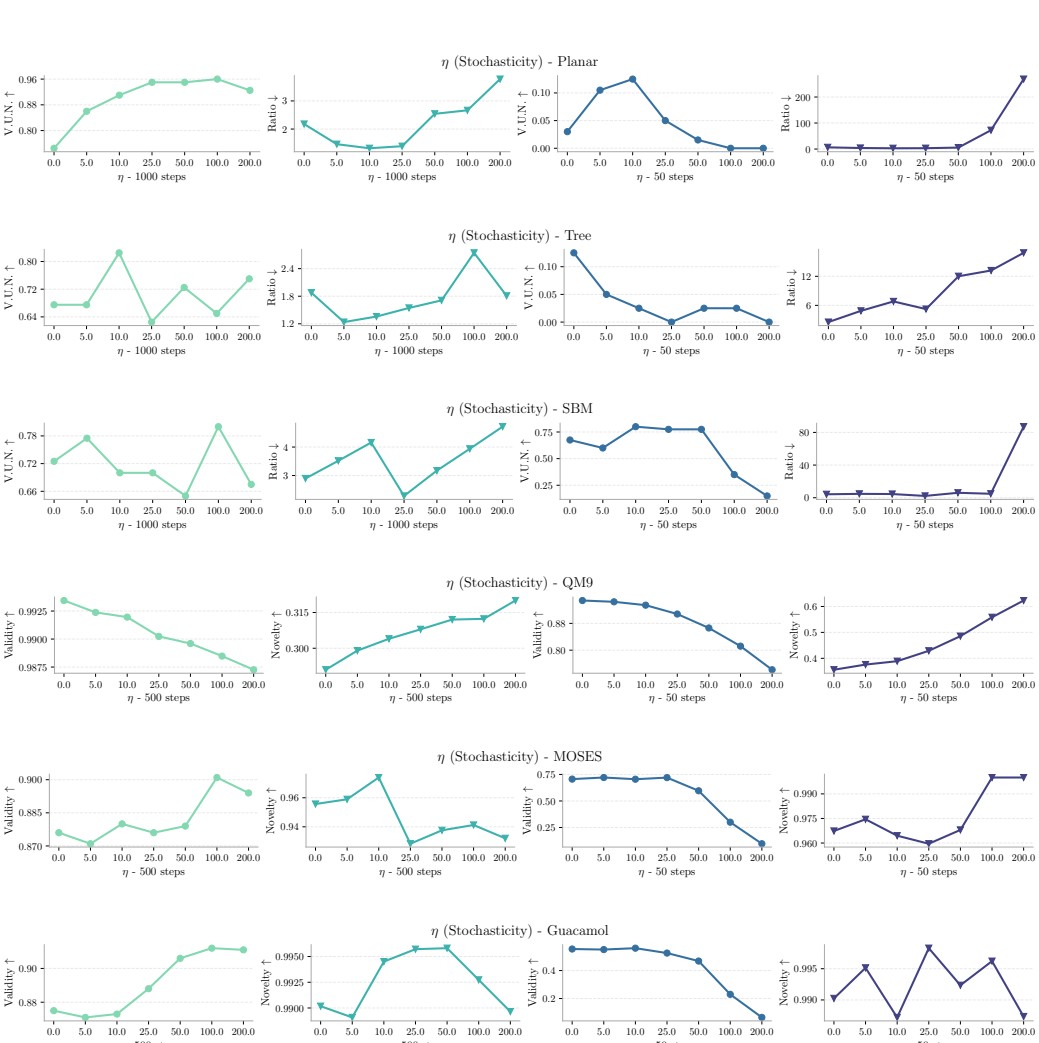

Figure 9: Influence of stochasticity level over all datasets.

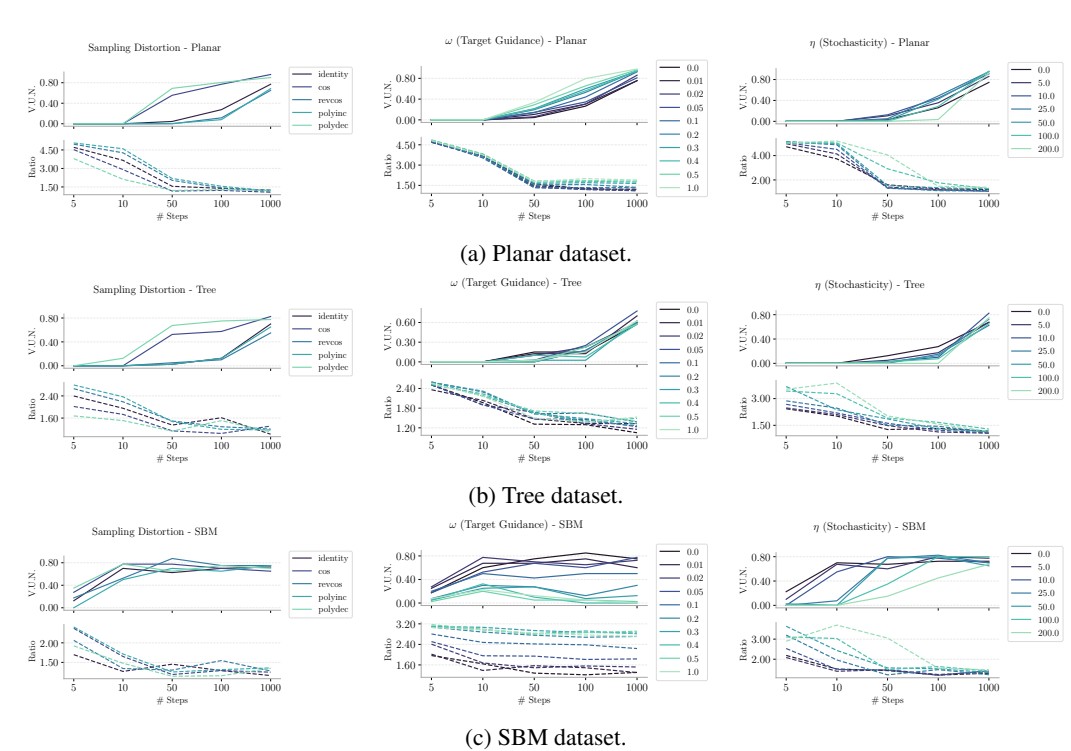

Figure 10: Stepwise parameter search for sampling optimization across synthetic datasets.

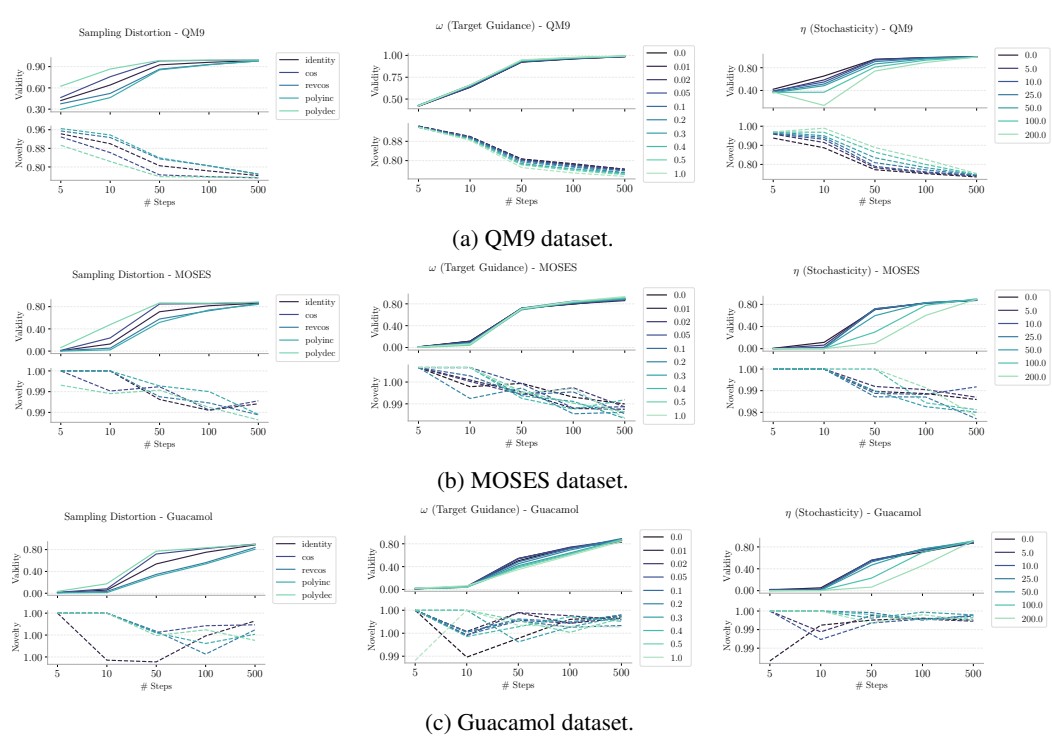

Figure 11: Stepwise parameter search for sampling optimization across molecular datasets.

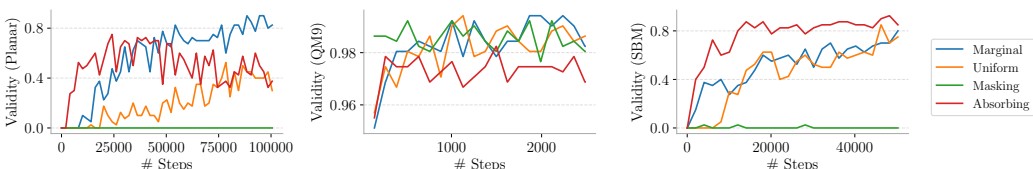

Figure 12: Influence of initial distribution over different datasets at different training steps.

## C  TRAIN OPTIMIZATION

In this section, we provide a more detailed analysis of the influence of the various training optimization strategies introduced in Sec. 4.2.1. In Appendix C.1, we empirically demonstrate the impact of selecting different initial distributions on performance, while in Appendix C.2, we examine the interaction between training and sampling optimization.

### C.1  INITIAL DISTRIBUTIONS

Under DeFoG's framework, the noising process for each dimension is modeled as a linear interpolation between the clean data distribution (the one-hot representation of the current state) and an initial distribution, $p_0$. As such, it is intuitive that different initial distributions result in varying performances, depending on the denoising dynamics they induce. In particular, they have a direct impact on the sampling dynamics through $R_t^*$ (see Appendix B.5) and may also pose tasks of varying difficulty for the graph transformer. In this paper, we explore four distinct initial distributions[2]:

**Uniform**: $p_0 = \left[\frac{1}{Z}, \frac{1}{Z}, \ldots, \frac{1}{Z}\right] \in \Delta^{Z-1}$. Here, the probability mass is uniformly distributed across all states, as proposed by Campbell et al. (2024).

**Masking**: $p_0 = [0, 0, \ldots, 0, 1] \in \Delta^Z$. In this setting, all the probability mass collapses into a new "mask" state at $t = 0$, as introduced by Campbell et al. (2024).

**Marginal**: $p_0 = [m_1, m_2, \ldots, m_Z] \in \Delta^{Z-1}$, where $m_i$ denotes the marginal probability of the $i$-th state in the dataset. This approach is widely used in state-of-the-art graph generation models (Vignac et al., 2022; Xu et al., 2024; Siraudin et al., 2024).

**Absorbing**: $p_0 = [0, \ldots, 1, \ldots, 0] \in \Delta^{Z-1}$, representing a one-hot encoding of the most common state (akin to applying an argmax operator to the marginal initial distribution).

In Figure 12, we present the training curves for each initial distribution for three different datasets.

We observe that the marginal distribution consistently achieves at least as good performance as the other initial distributions. This, along with the theoretical reasons outlined in Sec. 4.2.1, reinforces its use as the default initial distribution for DeFoG. The only dataset where marginal was surpassed was the SBM dataset, which we attribute to its inherently different nature (stochastic *vs.* deterministic). In this case, the absorbing distribution emerged as the best-performing choice. Interestingly, the absorbing distribution also tends to converge faster across datasets.

Lastly, it is worth noting that in discrete diffusion models for graphs, predicting the best limit noise distribution based solely on dataset characteristics remains, to our knowledge, an open question (Tseng et al., 2023). We expect this complexity to extend to discrete flow models as well. Although this is outside the scope of our work, we view this as an exciting direction for future research.

### C.2  INTERACTION BETWEEN SAMPLE AND TRAIN DISTORTIONS

From Appendix B.4, we observe that time distortions applied during the sampling stage can significantly affect performance. This suggests that graph discrete flow models do not behave evenly across time and are more sensitive to specific time intervals, where generative performance benefits from finer updates achieved by using smaller time steps. Building on this observation, we extended our analysis to the training stage, exploring two main questions:

---

[2]Recall that $Z$ represents the cardinality of the state space, and $\Delta^{Z-1}$ the associated probability simplex.

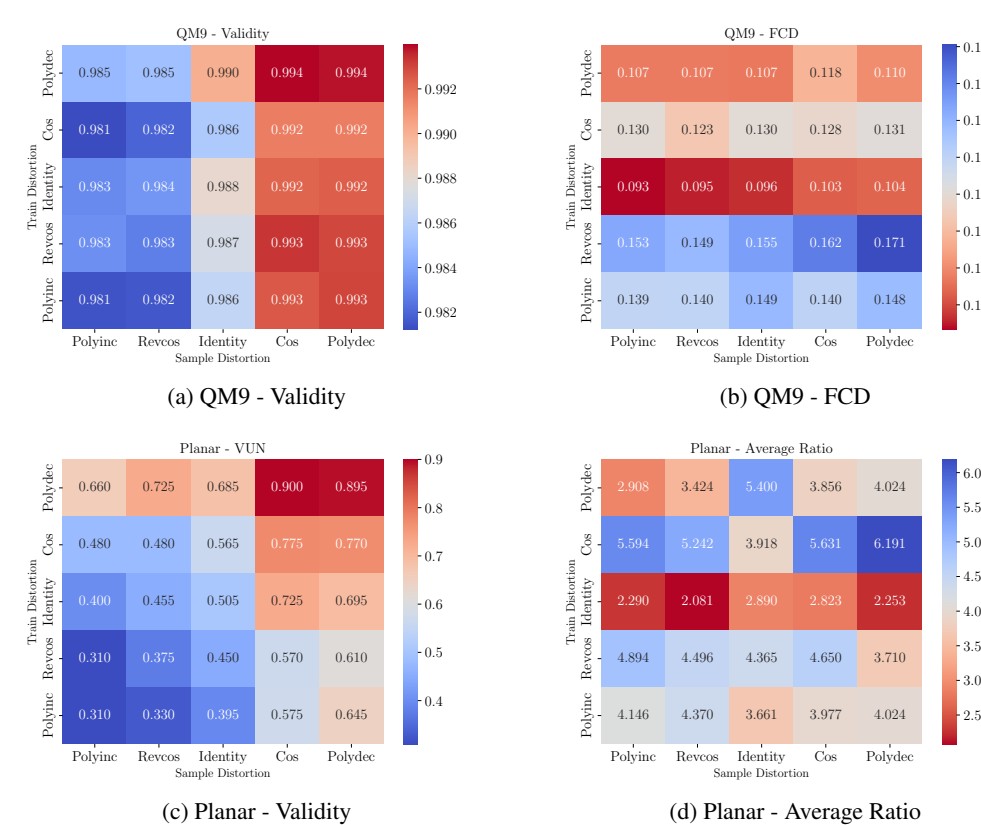

Figure 13: Interaction between training and sampling time distortions.

- Is there an universally optimal training time distortion for graph generation across different datasets?

- How do training and sampling time distortions interact? Is there alignment between the two? Specifically, if we understand the effect of a time distortion at one stage (training or sampling), can we infer its impact at the other?

To investigate these questions, we conducted a grid search. For two datasets, we trained five models, each with a different time distortion applied during training. Subsequently, we tested each model by applying the five different distortions at the sampling stage. The results are presented in Figure 13.

The results vary by dataset. For QM9, validity appears primarily influenced by the sampling distortion method, with a preference for distortions that encourage smaller steps at the end of the denoising process (such as polydec and cos). However, for FCD[3], the training distortion plays a more significant role.

For the planar dataset, we observe a near-perfect alignment between training and sampling distortions in terms of validity, with a clear preference for more accurate training models and finer sampling predictions closer to $t = 1$. The results for the average ratio metric, however, are less consistent and show volatility.

These findings help address our core questions: The interaction between training and sampling distortions, as well as the best training time distortion, is dataset-dependent. Nonetheless, for the particular case of the planar dataset, we observe a notable alignment between training and sampling distortions. This alignment suggests that times close to $t = 1$ are critical for correctly generating planar graphs. We conjecture that this alignment can be attributed to planarity being a global property that arises from local constraints, as captured by Kuratowski's Theorem, which states that

---

[3]FCD is calculated only for valid molecules, so this metric may inherently reflect survival bias.

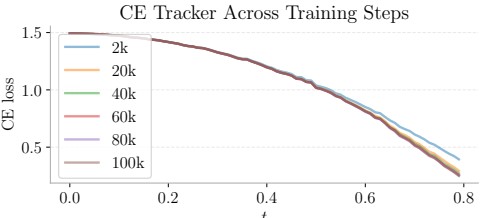 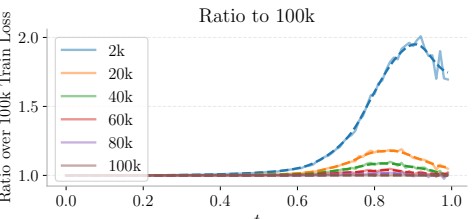

Figure 14: In the left figure, we present the cross-entropy (CE) loss used for training at different time steps across various stages of the training process. The right figure shows the ratio of each CE loss trajectory relative to the last one, illustrating the overall training trend and emphasizing which parts of the model are predominantly learned over time.

a graph is non-planar if and only if it contains a subgraph reducible to $K_5$ or $K_{3,3}$ through edge contraction (Kuratowski, 1930).

**Loss Tracker**   To determine if the structural properties observed in datasets like the planar dataset can be detected and exploited without requiring an exhaustive sweep over all possibilities, we propose developing a metric that quantifies the difficulty of predicting the clean graph for any given time point $t \in [0, 1)$. For this, we perform a sweep over $t$ for a given model, where for each $t$, we noise a sufficiently large batch of clean graphs and evaluate the model's training loss on them. This yields a curve that shows how the training loss varies as a function of $t$. We then track how this curve evolves across epochs. To make the changes more explicit, we compute the ratio of the loss curve relative to the fully trained model's values. These curves are shown in Figure 14.

As expected, the curve of training loss as a function of $t$ (left in Figure 14) is monotonically decreasing, indicating that as graphs are decreasingly noised, the task becomes simpler. However, the most interesting insights arise from the evolution of this curve across epochs (right in Figure 14). We observe that for smaller values of $t$, the model reaches its maximum capacity early in the training process, showing no significant improvements after the initial few epochs. In contrast, for larger values of $t$ (closer to $t = 1$), the model exhibits substantial improvements throughout the training period. This suggests that the model can continue to refine its predictions in the time range where the task is easier. These findings align with those in Figure 13, reinforcing our expectation that training the model to be more precise in this range or providing more refined sampling steps will naturally enhance performance in the planar dataset.

These insights offer a valuable understanding of the specific dynamics at play within the planar dataset. Nevertheless, the unique structural characteristics of each dataset may influence the interaction between training and sampling time distortions in ways that are not captured here. Future work could explore these dynamics across a wider range of datasets to assess the generalizability of our findings.

# D THEORETICAL RESULTS

In this section, we provide the proofs of the different theoretical results of the paper. We first provide the results that are domain agnostic, i.e., that hold for any data modality that lies in discrete state-spaces, and then the graph specific ones.

## D.1 DOMAIN AGNOSTIC THEORETICAL RESULTS

Here, we first provide the proof of Theorem 2 in Appendix D.1.1, and then the proof of Theorem 3 in Appendix D.1.2.

### D.1.1 BOUNDED ESTIMATION ERROR OF UNCONDITIONAL MULTIVARIATE RATE MATRIX

We start by introducing two important concepts, which will reveal important for the proof of the intended result.

**Unconditional Multivariate Rate Matrix** As exposed in Sec. 3.1, the marginal distribution and the rate matrix of a CTMC are related by the Kolmogorov equation:

$$
\begin{aligned}
\partial_t p_t &= R_t^T p_t \\
&= \underbrace{\sum_{z_{t+\mathrm{dt}} \neq z_t} R_t(z_{t+\mathrm{dt}}, z_t) p_t(z_{t+\mathrm{dt}})}_{\text{Probability Inflow}} - \underbrace{\sum_{z_{t+\mathrm{dt}} \neq z_t} R_t(z_t, z_{t+\mathrm{dt}}) p_t(z_t)}_{\text{Probability Outflow}}.
\end{aligned}
$$

The expansion in the second equality reveals the conservation law inherent in the Kolmogorov equation, illustrating that the time derivative of the marginal distribution represents the net balance between the inflow and outflow of probability mass at a given state.

Importantly, in the multivariate case, the (joint) rate matrix can be expressed through the following decomposition:

$$
\begin{aligned}
R_t(z_t^{1:D}, z_{t+\mathrm{dt}}^{1:D}) &= \sum_{d=1}^{D} \delta(z_t^{1:D \setminus (d)}, z_{t+\mathrm{dt}}^{1:D \setminus (d)}) \, R_t^{(d)}(z_t^{1:D}, z_{t+\mathrm{dt}}^{(d)}) \\
&= \sum_{d=1}^{D} \delta(z_t^{1:D \setminus (d)}, z_{t+\mathrm{dt}}^{1:D \setminus (d)}) \, \mathbb{E}_{p_{1|t}(z_1^{(d)} | z_t^{1:D})} \left[ R_t^{*(d)}(z_t^{(d)}, z_{t+\mathrm{dt}}^{(d)} | z_1^{(d)}) \right]. \quad (13)
\end{aligned}
$$

In the first equality, $1 : D \setminus (d)$ refers to all dimensions except $d$ and the $\delta$ term restricts contributions to rate matrices that account for at most one dimension transitioning at a time, since the probability of two or more independently noised dimensions transitioning simultaneously is zero under a continuous time framework (Campbell et al., 2022; 2024). In the second equality, the unconditional rate matrix is retrieved by taking the expectation over the $z_1$-conditioned rate matrices. Specifically, $R_t^{*(d)}(z_t^{(d)}, z_{t+\mathrm{dt}}^{(d)} | z_1^{(d)})$ denotes the univariate rate matrix corresponding to dimension $d$ (see Eq. (3))

**Total Variation** The total variation (TV) distance is a distance measure between probability distributions. While it can be defined more generally, this paper focuses on its application to discrete probability distributions over a finite sample space $\mathcal{Z}$. In particular, for two discrete probability distributions $P$ and $Q$, their total variation distance is defined as:

$$
\|P - Q\|_{\mathrm{TV}} = \frac{1}{2} \sum_{z \in \mathcal{Z}} |P(z) - Q(z)| \quad (14)
$$

We are now prepared to proceed with the proof of Theorem 2.

**Theorem 2** (Bounded estimation error of unconditional multivariate rate matrix). *Given $t \in [0, 1]$, $z_t^{1:D}, z_{t+\mathrm{dt}}^{1:D} \in \mathcal{Z}^D$, and $z_1^{1:D} \sim p_1(z_1^{1:D})$, let $R_t(z_t^{1:D}, z_{t+\mathrm{dt}}^{1:D})$ be the groundtruth rate matrix of the CTMC, which we approximate with $R_t^{\theta}(z_t^{1:D}, z_{t+\mathrm{dt}}^{1:D})$. The corresponding estimation error is upper-bounded as follows:*

$$
|R_t(z_t^{1:D}, z_{t+\mathrm{dt}}^{1:D}) - R_t^{\theta}(z_t^{1:D}, z_{t+\mathrm{dt}}^{1:D})|^2 \leq C_0 + C_1 \mathbb{E}_{p_1(z_1^{1:D})} \left[ p_{t|1}(z_t^{1:D} | z_1^{1:D}) \sum_{d=1}^{D} - \log p_{1|t}^{\theta, (d)}(z_1^{(d)} | z_t^{1:D}) \right], \quad (15)
$$

*where*

- $C_0 = 2D \sup_{d \in \{1,\ldots,D\}} \{C^2_{z^{(d)}}\} \sum_{z_1^{(d)} \in \mathcal{Z}} p_{1|t}^{(d)}(z_1^{(d)}|z_t^{1:D}) \log p_{1|t}^{(d)}(z_1^{(d)}|z_t^{1:D})$;

- $C_1 = 2D \sup_{d \in \{1,\ldots,D\}} \{C^2_{z^{(d)}}\}/p_1(z_1^{1:D})$;

with $C_{z^{(d)}} = \delta(z_t^{1:D\backslash(d)}, z_{t+dt}^{1:D\backslash(d)}) \sup_{z_1^{(d)} \in \mathcal{Z}} \{R_t^{*(d)}(z_t^{(d)}, z_{t+dt}^{(d)}|z_1^{(d)})\}$.

*Proof.* This proof is an adaptation of the proof of Theorem 3.3 from Xu et al. (2024) to the discrete flow matching setting.

By definition (Eq. (13)), we have:

$$
\begin{aligned}
R_t(z_t^{1:D}, z_{t+dt}^{1:D}) &= \sum_{d=1}^{D} \delta(z_t^{1:D\backslash(d)}, z_{t+dt}^{1:D\backslash(d)}) \, R_t^{(d)}(z_t^{1:D}, z_{t+dt}^{(d)}) \\
&= \sum_{d=1}^{D} \delta(z_t^{1:D\backslash(d)}, z_{t+dt}^{1:D\backslash(d)}) \, \mathbb{E}_{p_{1|t}^{(d)}(z_1^{(d)}|z_t^{1:D})} \left[ R_t^{*(d)}(z_t^{(d)}, z_{t+dt}^{(d)}|z_1^{(d)}) \right] \\
&= \sum_{d=1}^{D} \delta(z_t^{1:D\backslash(d)}, z_{t+dt}^{1:D\backslash(d)}) \sum_{z_1^{(d)}} p_{1|t}^{(d)}(z_1^{(d)}|z_t^{1:D}) R_t^{*(d)}(z_t^{(d)}, z_{t+dt}^{(d)}|z_1^{(d)})
\end{aligned}
$$

Thus:

$$
\begin{aligned}
|R_t(z_t^{1:D}, z_{t+dt}^{1:D}) &- R_t^\theta(z_t^{1:D}, z_{t+dt}^{1:D})| = \\
&= \left| \sum_{d=1}^{D} \delta(z_t^{1:D\backslash(d)}, z_{t+dt}^{1:D\backslash(d)}) \sum_{z_1^{(d)}} [R_t^{*(d)}(z_t^{(d)}, z_{t+dt}^{(d)}|z_1^{(d)}) \left( p_{1|t}^{(d)}(z_1^{(d)}|z_t^{1:D}) - p_{1|t}^{\theta,(d)}(z_1^{(d)}|z_t^{1:D}) \right)] \right| \\
&\leq \sum_{d=1}^{D} \delta(z_t^{1:D\backslash(d)}, z_{t+dt}^{1:D\backslash(d)}) \left| \sum_{z_1^{(d)}} \left[ R_t^{*(d)}(z_t^{(d)}, z_{t+dt}^{(d)}|z_1^{(d)}) \left( p_{1|t}^{(d)}(z_1^{(d)}|z_t^{1:D}) - p_{1|t}^{\theta,(d)}(z_1^{(d)}|z_t^{1:D}) \right) \right] \right| \\
&\leq \sum_{d=1}^{D} \delta(z_t^{1:D\backslash(d)}, z_{t+dt}^{1:D\backslash(d)}) \sup_{z_1^{(d)}} \{R_t^{*(d)}(z_t^{(d)}, z_{t+dt}^{(d)}|z_1^{(d)})\} \sum_{z_1^{(d)}} \left| p_{1|t}^{(d)}(z_1^{(d)}|z_t^{1:D}) - p_{1|t}^{\theta,(d)}(z_1^{(d)}|z_t^{1:D}) \right| \\
&= \sum_{d=1}^{D} 2\, C_{z^{(d)}} \, \|p_{1|t}^{(d)}(z_1^{(d)}|z_t^{1:D}) - p_{1|t}^{\theta,(d)}(z_1^{(d)}|z_t^{1:D})\|_{\text{TV}} \quad &(16) \\
&\leq \sum_{d=1}^{D} C_{z^{(d)}} \sqrt{2 D_{\text{KL}} \left( p_{1|t}^{(d)}(z_1^{(d)}|z_t^{1:D}) \,\|\, p_{1|t}^{\theta,(d)}(z_1^{(d)}|z_t^{1:D}) \right)} \quad &(17) \\
&= \sum_{d=1}^{D} \sqrt{2\, C^2_{z^{(d)}} \sum_{z_1^{(d)}} p_{1|t}^{(d)}(z_1^{(d)}|z_t^{1:D}) \log \frac{p^{(d)}(_{1|t}z_1^{(d)}|z_t^{1:D})}{p_{1|t}^{\theta,(d)}(z_1^{(d)}|z_t^{1:D})}}.
\end{aligned}
$$

In Eq. (16), we use the definition of TV distance as defined in Eq. (14) and Eq. (17) results from direct application of Pinsker's inequality. Now, we change the ordering of the sum and of the square root through the Cauchy-Schwarz inequality:

$$
\sum_{d=1}^{D} \sqrt{x_d} \leq \sum_{d=1}^{D} \sqrt{x_d} \cdot 1 \leq \sqrt{\sum_{d=1}^{D} \sqrt{x_d}^2} \sqrt{\sum_{d=1}^{D} \sqrt{1}^2} \leq \sqrt{D \sum_{d=1}^{D} x_d}
$$

So, we obtain:

$$|R_t(z_t^{1:D}, z_{t+dt}^{1:D}) - R_t^\theta(z_t^{1:D}, z_{t+dt}^{1:D})| \leq \sqrt{2\,D\,\sum_{d=1}^{D} C_{z^{(d)}}^2 \sum_{z_1^{(d)}} p_{1|t}^{(d)}(z_1^{(d)}|z_t^{1:D}) \log \frac{p_{1|t}^{(d)}(z_1^{(d)}|z_t^{1:D})}{p_{1|t}^{\theta,(d)}(z_1^{(d)}|z_t^{1:D})}}$$

$$\leq \sqrt{2\,D \sup_{d \in \{1,\ldots,D\}} \{C_{z^{(d)}}^2\} \sum_{d=1}^{D} \sum_{z_1^{(d)}} p_{1|t}^{(d)}(z_1^{(d)}|z_t^{1:D}) \left[ \log p_{1|t}^{(d)}(z_1^{(d)}|z_t^{1:D}) - \log p_{1|t}^{\theta,(d)}(z_1^{(d)}|z_t^{1:D}) \right]}$$

$$= \sqrt{C_2 \left( C_3 - \underbrace{\sum_{d=1}^{D} \sum_{z_1^{(d)}} p_{1|t}^{(d)}(z_1^{(d)}|z_t^{1:D}) \log p_{1|t}^{\theta,(d)}(z_1^{(d)}|z_t^{1:D})}_{} \right)},$$

where in the last step we rearrange the terms independent of the approximation parametrized by $\theta$ as constants: $C_2 = 2D \sup_{d \in \{1,\ldots,D\}} \{C_{z^{(d)}}^2\}$, $C_3 = \sum_{z_1^{(d)} \in \mathcal{Z}} p_{1|t}^{(d)}(z_1^{(d)}|z_t^{1:D}) \log p_{1|t}^{(d)}(z_1^{(d)}|z_t^{1:D})$. We now develop the underbraced term in the last equation[4]:

$$\sum_{d=1}^{D} \sum_{z_1^{(d)}} p_{1|t}^{(d)}(z_1^{(d)}|z_t^{1:D}) \log p_{1|t}^{\theta,(d)}(z_1^{(d)}|z_t^{1:D}) =$$

$$= \sum_{d=1}^{D} \sum_{z_1^{(d)}} \frac{p(z_1^{(d)}, z_t^{1:D})}{p_t(z_t^{1:D})} \log p_{1|t}^{\theta,(d)}(z_1^{(d)}|z_t^{1:D})$$

$$= \frac{1}{p_t(z_t^{1:D})} \sum_{d=1}^{D} \sum_{z_1^{(d)}} \sum_{z_1^{1:D \setminus (d)}} p(z_1^{(d)}, z_t^{1:D}, z_1^{1:D \setminus (d)}) \log p_{1|t}^{\theta,(d)}(z_1^{(d)}|z_t^{1:D})$$

$$= \frac{1}{p_t(z_t^{1:D})} \sum_{d=1}^{D} \sum_{z_1^{1:D}} p(z_1^{1:D}, z_t^{1:D}) \log p_{1|t}^{\theta,(d)}(z_1^{(d)}|z_t^{1:D})$$

$$= \frac{1}{p_t(z_t^{1:D})} \sum_{d=1}^{D} \sum_{z_1^{1:D}} p_1(z_1^{1:D}) p_{t|1}(z_t^{1:D}|z_1^{1:D}) \log p_{1|t}^{\theta,(d)}(z_1^{(d)}|z_t^{1:D})$$

$$= \frac{1}{p_t(z_t^{1:D})} \sum_{z_1^{1:D}} p_1(z_1^{1:D}) p_{t|1}(z_t^{1:D}|z_1^{1:D}) \sum_{d=1}^{D} \log p_{1|t}^{\theta,(d)}(z_1^{(d)}|z_t^{1:D})$$

$$= -C_4 \, \mathbb{E}_{p_1(z_1^{1:D})} \left[ p_{t|1}(z_t^{1:D}|z_1^{1:D}) \underbrace{\sum_{d=1}^{D} - \log p_{1|t}^{\theta,(d)}(z_1^{(d)}|z_t^{1:D})}_{\text{Cross-entropy}} \right],$$

where $C_4 = 1/p(z_1^{1:D})$.

Replacing back the obtained expression into the original equation, we obtain:

$$|R_t(z_t^{1:D}, z_{t+dt}^{1:D}) - R_t^\theta(z_t^{1:D}, z_{t+dt}^{1:D})|^2 \leq C_2 C_3 + C_2 C_4 \mathbb{E}_{p_1(z_1^{1:D})} \left[ p_{t|1}(z_t^{1:D}|z_1^{1:D}) \sum_{d=1}^{D} - \log p_{1|t}^{\theta,(d)}(z_1^{(d)}|z_t^{1:D}) \right],$$

retrieving the intended result. □

---

[4]In this step, we omit some subscripts from joint probability distributions as they are not defined in the main paper, but they can be inferred from context.

### D.1.2 BOUNDED DEVIATION OF THE GENERATED DISTRIBUTION

As proceeded in Appendix D.1.1, we start by introducing the necessary concepts that will reveal useful for the proof of the intended result.

**On the Choice of the CTMC Sampling Method**  Generating new samples using DFM amounts to simulate a multivariate CTMC according to:

$$p_{t+\mathrm{d}t|t}(z_{t+\mathrm{d}t}^{1:D}|z_t^{1:D}) = \delta(z_t^{1:D}, z_{t+\mathrm{d}t}^{1:D}) + R_t(z_t^{1:D}, z_{t+\mathrm{d}t}^{1:D})\mathrm{d}t, \tag{18}$$

where $R_t(z_t^{1:D}, z_{t+\mathrm{d}t}^{1:D})$ denotes the unconditional multivariate rate matrix defined in Eq. (13). This process can be simulated exactly using Gillespie's Algorithm (Gillespie, 1976; 1977). However, such an algorithm does not scale for large $D$ (Campbell et al., 2022). Although $\tau$-leaping is a widely adopted approximate algorithm to address this limitation (Gillespie, 2001), it requires ordinal discrete state spaces, which is suitable for cases like text or images but not for graphs. Therefore, we cannot apply it in the context of this paper. Additionally, directly replacing the infinitesimal step $\mathrm{d}t$ in Eq. (13) with a finite time step $\Delta t$ *à la* Euler method is inappropriate, as $R_t(z_t^{1:D}, z_{t+\mathrm{d}t}^{1:D})$ prevents state transitions in more than one dimension per step under the continuous framework. Instead, Campbell et al. (2024) propose an approximation where the Euler step is applied independently to each dimension, as seen in Eq. (4).

In this section, we theoretically demonstrate that, despite its approximation, the independent-dimensional Euler sampling method error remains bounded and can be made arbitrarily small by reducing the step size $\Delta t$ or by reducing the estimation error of the rate matrix.

**Markov Kernel of a CTMC**  For this proof, we also introduce the notion of Markov kernel of a CTMC. As previously seen, the *rate matrix* (or *generator*), $R_t$, characterizes the infinitesimal transition rates between states, governing the dynamics of the process. The *Markov kernel*, $\mathcal{R}_t$, is a function that provides the transition probabilities between states over a *finite time interval*. For example, for a univariate CTMC with a state space $Z$, the Markov kernel $\mathcal{R}_t(s, t)$ is a matrix where each entry $\mathcal{R}_{t,ij}(s, s')$ represents the probability that the single variable transitions from state $i$ to state $j$ during the time interval $[s, s']$. These matrices are stochastic, i.e., for fixed $s$ and $s'$, $\sum_{j \in S} \mathcal{R}_{t,ij}(s, s') = 1, \ \forall i$. This contrasts with the rate matrix where rows sum to 1. Additionally, Markov kernels must also respect the initial condition $\mathcal{R}_{t,ij}(s, s) = I_{Z \times Z}$[5]. Importantly, a constant rate matrix, $R$, between $t$ and $t + \Delta t$ yields the following Markov kernel:

$$\mathcal{R}(t, t + \Delta t) = e^{R\Delta t} \tag{19}$$

We are now in conditions of proceeding to the proof of Theorem 3. We start by first proving that, in the univariate case, the time derivatives of the *conditional* rate matrices are upper bounded.

**Lemma 5** (Upper bound time derivative of conditional univariate rate matrix). *For* $t \in (0, 1)$, $z_t, z_{t+\mathrm{d}t}, z_1 \in \mathbb{Z}$, *with* $z_t \neq z_{t+\mathrm{d}t}$, *then we have:*

$$|\partial_t R_t^*(z_t, z_{t+\mathrm{d}t}|z_1)| \leq \frac{2}{p_{t|1}(z_t|z_1)^2}.$$

*Proof.* Recall that $p_{t|1}(z_t|z_1) = t\,\delta(z_t, z_1) + (1-t)\,p_0(z_t)$ (from Eq. (1)). Two different cases must then be considered.

In the first case, $p_{t|1}(z_t|z_1) = 0$. This implies that both extremes of the linear interpolation are 0. In that case, the linear interpolation will be identically 0 for $t \in (0, 1)$. Thus, by definition, $R_t^*(z_t, z_{t+\mathrm{d}t}|z_1) = 0$ for $t \in (0, 1)$, which implies that $|\partial_t R_t^*(z_t, z_{t+\mathrm{d}t}|z_1)| = 0$.

Otherwise ($p_{t|1}(z_t|z_1) > 0$), we recall that $R_t^*(z_t, z_{t+\mathrm{d}t}|z_1)$ with $z_t \neq z_{t+\mathrm{d}t}$ has the following form:

$$R_t^*(z_t, z_{t+\mathrm{d}t}|z_1) = \frac{\mathrm{ReLU}\left(\partial_t p_{t|1}(z_{t+\mathrm{d}t}|z_1) - \partial_t p_{t|1}(z_t|z_1)\right)}{\mathbb{Z}_t^{>0} p_{t|1}(z_t|z_1)}, \tag{20}$$

where $\mathbb{Z}_t^{>0} = |\{z_t : p_{t|1}(z_t|z_1) > 0\}|$.

---

[5] $I_{Z \times Z}$ denotes the identity matrix of dimension $Z \times Z$.

By differentiating the explicit form of $p_{t|1}(z_t|z_1)$, we have that $\partial^2 p_{t|1}(z_t|z_1) = 0$. As a consequence, the numerator of eq. (20) has zero derivative. Additionally, we also note that $\mathbb{Z}_t^{>0}$ is constant. Again, since $p_{t|1}(z_t|z_1)$ is a linear interpolation between $z_1$ and $p_0$ and, therefore, it is impossible for $p_{t|1}(z_t|z_1)$ to suddenly become 0 for $t \in (0,1)$.

Consequently, we have:

$$\partial_t R_t^*(z_t, z_{t+\mathrm{d}t}|z_1) = \frac{\mathrm{ReLU}\left(\partial_t p_{t|1}(z_{t+\mathrm{d}t}|z_1) - \partial_t p_{t|1}(z_t|z_1)\right)}{\mathbb{Z}_t^{>0}} \partial_t \left(\frac{1}{p_{t|1}(z_t|z_1)}\right)$$

$$= -\frac{\mathrm{ReLU}\left(\partial_t p_{t|1}(z_{t+\mathrm{d}t}|z_1) - \partial_t p_{t|1}(z_t|z_1)\right)}{\mathbb{Z}_t^{>0}} \frac{\partial_t p_{t|1}(z_t|z_1)}{p_{t|1}(z_t|z_1)^2}.$$

We necessarily have $|\partial_t p_{t|1}(z_t|z_1)| = |\delta(z_t, z_1) - p_0(z_t)| \leq 1$, $\mathrm{ReLU}\left(\partial_t p(z_{t+\mathrm{d}t}|z_1) - \partial_t p(z_t|z_1)\right) \leq 2$, $\mathbb{Z}_t^{>0} \geq 1$, and, necessarily, $p(z_t|z_1) > 0$. Thus:

$$|\partial_t R_t^*(z_t, z_{t+\mathrm{d}t}|z_1)| \leq \frac{\mathrm{ReLU}\left(\partial_t p_{t|1}(z_{t+\mathrm{d}t}|z_1) - \partial_t p_{t|1}(z_t|z_1)\right)}{\mathbb{Z}_t^{>0}} \frac{|\delta(z_t, z_1) - p_0(z_t)|}{p_{t|1}(z_t|z_1)^2}$$

$$\leq \frac{2}{p_{t|1}(z_t|z_1)^2}.$$

$\square$

We now upper bound the time derivative of the *unconditonal* multivariate rate matrix. We use Lemma 5 as an intermediate result to accomplish so. Additionally, we consider the following assumption.

**Assumption 6.** For $z_t^{1:D} \in \mathcal{Z}^D, z_1^{(d)} \in \mathcal{Z}$ and $t \in [0,1]$, for each variable $z^{(d)}$ of a joint variable $z^{1:D}$, there exists a constant $B_t^{(d)} > 0$ such that $p_{1|t}(z_1^{(d)}|z_t^{1:D}) \leq B_t^{(d)} p_{t|1}(z_t^{(d)}|z_1^{(d)})^2$.

This assumption states that the denoising process is upper bounded by a quadratic term on the noising process. This assumption is reasonable because, while the noising term applies individually to each component of the data, the denoising process operates on the joint variable, allowing for a more comprehensive and interdependent correction that reflects the combined influence of all components.

**Proposition 7** (Upper bound time derivative of unconditional multivariate rate matrix). *For $z_t^{1:D}, z_{t+\mathrm{d}t}^{1:D} \in \mathcal{Z}^D$ and $t \in (0,1)$, under Assumption 6, we have:*

$$|\partial_t R_t^{1:D}(z_t^{1:D}, z_{t+\mathrm{d}t}^{1:D})| \leq 2B_t Z D,$$

*with $B_t = \sup_{d \in 1,\dots,D} B_t^{(d)}$.*

*Proof.* From Eq. (13), the unconditional rate matrix is given by:

$$R_t(z_t^{1:D}, z_{t+\mathrm{d}t}^{1:D}) = \sum_{d=1}^{D} \delta(z_t^{1:D\backslash(d)}, z_{t+\mathrm{d}t}^{1:D\backslash(d)}) R_t^{(d)}(z_t^{1:D}, z_{t+\mathrm{d}t}^{(d)})$$

$$= \sum_{d=1}^{D} \delta(z_t^{1:D\backslash(d)}, z_{t+\mathrm{d}t}^{1:D\backslash(d)}) \mathbb{E}_{p_{1|t}^{(d)}(z_1^{(d)}|z_t^{1:D})} \left[ R_t^{*(d)}(z_t^{(d)}, z_{t+\mathrm{d}t}^{(d)}|z_1^{(d)}) \right]$$

$$= \sum_{d=1}^{D} \delta(z_t^{1:D\backslash(d)}, z_{t+\mathrm{d}t}^{1:D\backslash(d)}) \sum_{z_1^{(d)} \in \mathcal{Z}} p_{1|t}^{(d)}(z_1^{(d)}|z_t^{1:D}) R_t^{*(d)}(z_t^{(d)}, z_{t+\mathrm{d}t}^{(d)}|z_1^{(d)}).$$

So, by linearity of the time derivative, we have:

$$\left|\partial_t R_t(z_t^{1:D}, z_{t+\mathrm{d}t}^{1:D})\right| = \left|\sum_{d=1}^{D} \delta(z_t^{1:D\backslash(d)}, z_{t+\mathrm{d}t}^{1:D\backslash(d)}) \sum_{z_1^{(d)} \in \mathcal{Z}} p_{1|t}^{(d)}(z_1^{(d)}|z_t^{1:D}) \, \partial_t R_t^{*(d)}(z_t^{(d)}, z_{t+\mathrm{d}t}^{(d)}|z_1^{(d)})\right|$$

$$\leq \sum_{d=1}^{D} \delta(z_t^{1:D\backslash(d)}, z_{t+\mathrm{d}t}^{1:D\backslash(d)}) \sum_{z_1^{(d)} \in \mathcal{Z}} p_{1|t}^{(d)}(z_1^{(d)}|z_t^{1:D}) \left|\partial_t R_t^{*(d)}(z_t^{(d)}, z_{t+\mathrm{d}t}^{(d)}|z_1^{(d)})\right|$$

$$\leq \sum_{d=1}^{D} \delta(z_t^{1:D\backslash(d)}, z_{t+\mathrm{d}t}^{1:D\backslash(d)}) \sum_{z_1^{(d)} \in \mathcal{Z}} B_t^{(d)} \, p_{t|1}(z_t^{(d)}|z_1^{(d)})^2 \frac{2}{p_{t|1}(z_t^{(d)}|z_1^{(d)})^2}$$

$$\leq \sum_{d=1}^{D} \delta(z_t^{1:D\backslash(d)}, z_{t+\mathrm{d}t}^{1:D\backslash(d)}) \, 2B_t Z$$

$$\leq 2B_t ZD,$$

where in the first inequality triangular we apply triangular inequality; in the second inequality, we use Lemma 5 and in Assumption 6 to upper bound $|\partial_t R_t^{*(d)}(z_t^{(d)}, z_{t+\mathrm{d}t}^{(d)}|z_1^{(d)})|$ and $p_{1|t}(z_1^{(d)}|z_t^{1:D})$, respectively. $\qquad\square$

Now, we finally start the proof of Theorem 3

**Theorem 3** (Bounded deviation of the generated distribution). *Let $\{z_t^{1:D}\}_{t\in[0,1]} \in \mathcal{Z}^D \times [0,1]$ be a CTMC starting with $p(z_0^{1:D}) = p_\epsilon$ and ending with $p(z_1^{1:D}) = p_{data}$, whose groundtruth rate matrix is $R_t$. Additionally, let $(y_k^{1:D})_{k=0,1,\ldots,K}$ be a Euler sampling approximation of that CTMC, with maximum step size $\Delta T = \sup_k \Delta t_k$ and an approximate rate matrix $R_t^\theta$. Then, under Assumption 6, the following total variation bound holds:*

$$\|p(y_K^{1:D}) - p_{data}\|_{TV} \leq UZD + B(ZD)^2 \Delta t + O(\Delta t), \tag{21}$$

*where* $U = \sup\limits_{\substack{t\in[0,1], \\ z_t^{1:D}, z_{t+\mathrm{d}t}^{1:D} \in \mathcal{Z}^D}} \sqrt{C_0 + C_1 \mathbb{E}_{p_1(z_1^{1:D})}\left[p_{t|1}(z_t^{1:D}|z_1^{1:D}) \sum_{d=1}^{D} -\log p_{1|t}^\theta(z_1^{(d)}|z_t^{1:D})\right]}$ *and*

$$B = \sup_{\substack{t\in[0,1], z_1^{(d)} \in \mathcal{Z} \\ z_t^{1:D} \in \mathcal{Z}^D}} B_t^{(d)}.$$

*Proof.* We start the proof by clarifying the notation for the Euler sampling approximation process. We denote its discretization timesteps by $0 = t_0 < t_1 < \ldots < t_K = 1$, with $\Delta t_k = t_k - t_{k-1}$. It is initiated at the same limit distribution as the groundtruth CTMC, $p_\epsilon$, and the bound to be proven will quantify the deviation that the approximated procedure incurs in comparison to the groundtruth CTMC. To accomplish so, we define $\mathcal{R}_k^{\theta,E}$ as the Markov kernel that corresponds to apply Euler sampling with the approximated rate matrix $R_t^\theta$, moving from $t_{k-1}$ to $t_k$. Therefore, $\mathcal{R}^{\theta,E} = \mathcal{R}_1^{\theta,E} \mathcal{R}_2^{\theta,E} \ldots \mathcal{R}_K^{\theta,E}$ and $p(y_K^{1:D}) = p_\epsilon \mathcal{R}^{\theta,E}$.

We first apply the same decomposition to the left-hand side of Theorem 3, as Campbell et al. (2022), Theorem 1:

$$\|p(y_K^{1:D}) - p_{\text{data}}\|_{\text{TV}} = \|p_\epsilon \mathcal{R}^{\theta,E} - p_{\text{data}}\|_{\text{TV}}$$

$$\leq \|p_\epsilon \mathcal{R}^{\theta,E} - p_\epsilon \mathbb{P}_{1|0}\|_{\text{TV}} + \|p_\epsilon - \underbrace{p(z_0^{1:D})}_{=p_\epsilon}\|_{\text{TV}} \tag{22}$$

$$\leq \|p_\epsilon \mathcal{R}^{\theta,E} - p_\epsilon \mathbb{P}_{1|0}\|_{\text{TV}}$$

$$\leq \sum_{k=1}^{K} \sup_\nu \|\nu \mathcal{R}_k^{\theta,E} - \nu \mathcal{P}_k\|_{\text{TV}}, \tag{23}$$

where, in Eq. (22), $\mathbb{P}_{1|0}$ denotes the path measure of the exact groundtruth CTMC and the difference between limit distributions (second term from Eq. (22)) is zero since in flow matching the convergence to the limit distribution via linear interpolation is not asymptotic (as in diffusion models) but actually attained at $t = 0$. In Eq. (23), we introduce the stepwise path measure, i.e., $\mathcal{P}_k = \mathbb{P}_{t_k|t_{k-1}}$, such that $\mathbb{P}_{T|0} = \mathcal{P}_1\mathcal{P}_2 \ldots \mathcal{P}_K$. Therefore, finding the intended upper bound amounts to establish bounds on the total variation distance for each interval $[t_{k-1}, t_k]$.

For any distribution $\nu$:

$$\|\nu\mathcal{R}_k^{\theta,E} - \nu\mathcal{P}_k, \|_{\text{TV}} \leq \|\nu\mathcal{R}_k^{\theta,E} - \nu\mathcal{R}_k^{\theta} + \nu\mathcal{R}_k^{\theta} - \nu\mathcal{P}_k\|_{\text{TV}}$$
$$\leq \|\nu\mathcal{P}_k - \nu\mathcal{R}_k^{\theta}\|_{\text{TV}} + \|\nu\mathcal{R}_k^{\theta} - \nu\mathcal{R}_k^{\theta,E}\|_{\text{TV}}, \tag{24}$$

where $\mathcal{R}_k^{\theta}$ denotes the resulting Markov kernel of running a CTMC with constant rate matrix $R_{t_{k-1}}^{\theta}$ between $t_{k-1}$ and $t_k$.

For the first term, we use Proposition 5 from Campbell et al. (2022) to relate the total variation distance imposed by the Markov kernels with the difference between the corresponding rate matrices:

$$\|\nu\mathcal{P}_k - \nu\mathcal{R}_k^{\theta}\|_{\text{TV}} \leq \int_{t_{k-1}}^{t_k} \sup_{z_t^{1:D} \in \mathcal{Z}^D} \left\{ \sum_{z_{t+dt}^{1:D} \neq z_t^{1:D}} \left| R_t(z_t^{1:D}, z_{t+dt}^{1:D}) - R_{t_{k-1}}^{\theta}(z_t^{1:D}, z_{t+dt}^{1:D}) \right| \right\} dt$$

$$\leq \underbrace{\int_{t_{k-1}}^{t_k} \sup_{z_t^{1:D} \in \mathcal{Z}^D} \left\{ \sum_{z_{t+dt}^{1:D} \neq z_t^{1:D}} \left| R_t(z_t^{1:D}, z_{t+dt}^{1:D}) - R_{t_{k-1}}(z_t^{1:D}, z_{t+dt}^{1:D}) \right| \right\} dt}_{\text{Discretization Error}}$$

$$+ \underbrace{\int_{t_{k-1}}^{t_k} \sup_{z_t^{1:D} \in \mathcal{Z}^D} \left\{ \sum_{z_{t+dt}^{1:D} \neq z_t^{1:D}} \left| R_{t_{k-1}}(z_t^{1:D}, z_{t+dt}^{1:D}) - R_{t_{k-1}}^{\theta}(z_t^{1:D}, z_{t+dt}^{1:D}) \right| \right\} dt}_{\text{Estimation Error}}$$

The first term consists of the discretization error, where we compare the chain with groundtruth rate matrix changing continuously between $t_{k-1}$ and $t_k$ with its discretized counterpart, i.e., a chain where the rate matrix is held constant to its value at the beginning of the interval. The second corresponds to the estimation error, where we compare the chain generated by the discretized groundtruth rate matrix with an equally discretized chain but that uses an estimated rate matrix instead. For the former, we have:

$$\int_{t_{k-1}}^{t_k} \sup_{z_t^{1:D} \in \mathcal{Z}^D} \left\{ \sum_{z_{t+dt}^{1:D} \neq z_t^{1:D}} \left| R_t(z_t^{1:D}, z_{t+dt}^{1:D}) - R_{t_{k-1}}(z_t^{1:D}, z_{t+dt}^{1:D}) \right| \right\} dt$$

$$\leq \int_{t_{k-1}}^{t_k} \sup_{z_t^{1:D} \in \mathcal{Z}^D} \left\{ \sum_{z_{t+dt}^{1:D} \neq z_t^{1:D}} \left| \partial_t R_{t_c}(z_t^{1:D}, z_{t+dt}^{1:D})(t - t_{k-1})) \right| \right\} dt \tag{25}$$

$$\leq \int_{t_{k-1}}^{t_k} ZD \sup_{z_t^{1:D}, z_{t+dt}^{1:D} \in \mathcal{Z}^D} \left\{ \left| \partial_t R_{t_c}(z_t^{1:D}, z_{t+dt}^{1:D}) \right| \right\} |t - t_{k-1}| \, dt \tag{26}$$

$$\leq 2Z^2D^2 \int_{t_{k-1}}^{t_k} B_t \, |t - t_{k-1}| \, dt, \tag{27}$$

$$= B_k(ZD\Delta t_k)^2, \tag{28}$$

where, in Eq. (25), we use the Mean Value Theorem, with $t_c \in (t_{k-1}, t_k)$; in Eq. (26), we use the fact that there are $ZD$ values of $z_{t+dt}^{1:D}$ that differ at most in only one coordinate from $z_t^{1:D}$; in Eq. (27), we

use the result from Proposition 7 to upper bound the time derivative of the multivariate unconditional rate matrix; and finally, in Eq. (28), we define $B_k = \sup_{t \in (t_{k-1}, t_k)} B_t$ and $\Delta t_k = t_k - t_{k-1}$.

For the estimation error term, we have:

$$\int_{t_{k-1}}^{t_k} \sup_{z_t^{1:D} \in \mathcal{Z}^D} \left\{ \sum_{z_{t+\mathrm{d}t}^{1:D} \neq z_t^{1:D}} \left| R_{t_{k-1}}(z_t^{1:D}, z_{t+\mathrm{d}t}^{1:D}) - R_{t_{k-1}}^\theta(z_t^{1:D}, z_{t+\mathrm{d}t}^{1:D}) \right| \right\} \mathrm{d}t$$

$$\leq \int_{t_{k-1}}^{t_k} U_k Z D \, \mathrm{d}t, \tag{29}$$

$$\leq U_k Z D \Delta t_k, \tag{30}$$

where, in Eq. (29), we use again the fact that there are $ZD$ values of $z_{t+\mathrm{d}t}^{1:D}$ that differ at most in only one coordinate from $z_t^{1:D}$ along with the estimation error upper bound from Theorem 2. In particular, we consider $U_k = \sup_{\substack{t \in [t_{k-1}, t_k], \\ z_t^{1:D}, z_{t+\mathrm{d}t}^{1:D} \in \mathcal{Z}^D}} U_k^{z_t^{1:D} \rightarrow z_{t+\mathrm{d}t}^{1:D}}$, with:

$$U_k^{z_t^{1:D} \rightarrow z_{t+\mathrm{d}t}^{1:D}} = \sqrt{C_0 + C_1 \mathbb{E}_{p_1(z_1^{1:D})} \left[ p_{t|1}(z_t^{1:D} | z_1^{1:D}) \sum_{d=1}^{D} - \log p_{1|t}^{\theta,(d)}(z_1^{(d)} | z_t^{1:D}) \right]},$$

i.e., the square root of the right-hand side of Eq. (15).

It remains to bound the second term from Eq. (24). We start by analyzing the Markov kernel corresponding to a Markov chain with *constant* rate matrix $R_{t_{k-1}}^\theta$ between $t_{k-1}$ and $t_k$. In that case, from Eq. (19) we obtain:

$$\mathcal{R}_k^\theta(t_{k-1}, t_k) = e^{R_{t_{k-1}}^\theta \Delta t_k}$$

$$= \sum_{i=0}^{\infty} \frac{(R_{t_{k-1}}^\theta \Delta t_k)^i}{i!}$$

$$= I + R_{t_{k-1}}^\theta \Delta t_k + \frac{(R_{t_{k-1}}^\theta \Delta t_k)^2}{2!} + \frac{(R_{t_{k-1}}^\theta \Delta t_k)^3}{3!} + \cdots$$

On the other hand, we have from Eq. (4) that sampling with the Euler approximation in multivariate Markov chain corresponds to:

$$\mathcal{R}_k^{\theta,E}(t_{k-1}, t_k) = \tilde{p}_{t_k|t_{k-1}}(z_{t_k}^{1:D} | z_{t_{k-1}}^{1:D})$$

$$= \prod_{d=1}^{D} \tilde{p}_{t_k|t_{k-1}}^{(d)}(z_{t_k}^{(d)} | z_{t_{k-1}}^{1:D})$$

$$= \prod_{d=1}^{D} \delta(z_{t_{k-1}}^{(d)}, z_{t_k}^{(d)}) + R_t^{\theta,d}(z_{t_{k-1}}^{1:D}, z_{t_k}^{(d)}) \Delta t_k \tag{31}$$

$$= \underbrace{\delta(z_{t_{k-1}}^{1:D}, z_{t_k}^{1:D})}_{=I} + \Delta t_k \underbrace{\sum_{d=1} \delta\left(z_{t_{k-1}}^{1:D \setminus (d)}, z_{t_k}^{1:D \setminus (d)}\right) R_t^{\theta,d}(z_{t_{k-1}}^{1:D}, z_{t_k}^{(d)})}_{=R_{t_{k-1}}^\theta} + O(\Delta t_k^2),$$

where in Eq. (31) we have that the approximated transition rate matrix is computed according to Eq. (13) but using $p_{1|t}^{\theta,(d)}(z_1^{(d)} | z_t^{1:D})$ instead of $p_{1|t}^{(d)}(z_1^{(d)} | z_t^{1:D})$.

Consequently, we have:

$$\left\| \nu \mathcal{R}_k^\theta - \nu \mathcal{R}_k^{\theta,E} \right\|_{\mathrm{TV}} = O(\Delta t_k^2). \tag{32}$$

Therefore, we get the intended result by gathering the results from Eq. (28), Eq. (30), and Eq. (32).

$$\|p(y_K^{1:D}) - p_{\text{data}}\|_{\text{TV}} \leq \sum_{k=1}^{K} \left( U_k ZD\Delta t_k \; + \; B_k (ZD\Delta t_k)^2 \; + \; O(\Delta t_k^2) \right)$$
$$\leq UZD \; + \; B(ZD)^2\Delta t + \; O(\Delta t),$$

where $\Delta t = \sup_k \Delta t_k$ is the maximum step size, $\sum_{k=1}^{K} \Delta t_k = 1$, $U = \sup_k U_k$ and $B = \sup_k B_k$.

$\square$

## D.2 GRAPH SPECIFIC THEORETICAL RESULTS

We now proceed to the graph specific theoretical results.

### D.2.1 ADDITIONAL FEATURES EXPRESSIVITY

This section explains the expressivity of the RRWP features used in DeFoG. We summarize the findings of Ma et al. (2023) in Proposition 7, who establish that, by encoding random walk probabilities, the RRWP positional features can be used to arbitrarily approximate several essential graph properties when fed into an MLP. Specifically, point 1 shows that RRWP with $K - 1$ steps encodes all shortest path distances for nodes up to $K - 1$ hops. Additionally, points 2 and 3 indicate that RRWP features effectively capture diverse graph propagation dynamics.

**Proposition 7** (Expressivity of an MLP with RRWP encoding (Ma et al., 2023)). *For any $n \in \mathbb{N}$, let $G_n \subseteq \{0,1\}^{n \times n}$ denote all adjacency matrices of $n$-node graphs. For $K \in \mathbb{N}$, and $A \in \mathbb{G}_n$, consider the RRWP:*

$$P = [I, M, \ldots, M^{K-1}] \in \mathbb{R}^{n \times n \times K}$$

*Then, for any $\epsilon > 0$, there exists an* MLP $: \mathbb{R}^{K-1} \to \mathbb{R}$ *acting independently across each $n$ dimension such that* MLP$(P)$ *approximates any of the following to within $\epsilon$ error:*

*1. MLP$(P)_{ij} \approx SPD_{K-1}(i, j)$*

*2. MLP$(P) \approx \sum_{k=0}^{K-1} \theta_k (D^{-1} A)^k$*

*3. MLP$(P) \approx \theta_0 I + \theta_1 A$*

*in which $SPD_{K-1}(i, j)$ is the $K - 1$ truncated shortest path distance, and $\theta_k \in \mathbb{R}$ are arbitrary coefficients.*

Siraudin et al. (2024) experimentally validate the effectiveness of RRWP features for graph diffusion models and propose extending their proof to additional graph properties that GNNs fail to capture (Xu et al., 2019; Morris et al., 2019). For example, an MLP with input $M^k$ with $k = N - 1$ for an $N$-node graph can approximate the connected component of each node and the number of vertices in the largest connected component. Additionally, RRWP features can be used to capture cycle-related information.

### D.2.2 NODE PERMUTATION EQUIVARIANCE AND INVARIANCE PROPERTIES

The different components of a graph generative model have to respect different graph symmetries. For example, the permutation equivariance of the model architecture ensures the output changes consistently with any reordering of input nodes, while permutation-invariant loss evaluates the model's performance consistently across isomorphic graphs, regardless of node order. We provide a proof for related properties included in Lemma 1 as follows.

**Lemma 1** (Node Permutation Equivariance and Invariance Properties of DeFoG). *The DeFoG model is permutation equivariant, its loss function is permutation invariant, and its sampling probability is permutation invariant.*

*Proof.* Recall that we denote an undirected graph with $N$ nodes by $G = (x^{1:n:N}, e^{1:i<j:N})$. Here, each node variable is represented as $x^n \in \mathcal{X} = \{1, \ldots, X\}$, and each edge variable as $e^{(ij)} \in \mathcal{E} = \{1, \ldots, E\}$. We also treat $G$ as a multivariate data point consisting of $D$ discrete variables including all nodes and all edges.

We then consider a permutation function $\sigma$, which is applied to permute the graph's node ordering. Under this permutation, the index $n$ will be mapped to $\sigma(n)$. We denote the ordered set of nodes and edges in the original ordering by $x^{(1:n:N)}$ and $e^{(1:i<j:N)}$, respectively, and by $x'^{(1:n:N)}$ and $e'^{(1:i<j:N)}$ after permutation. Additionally, $x^{(n)}$ denotes the $n$-th entry of the corresponding ordered set (and analogously for edges). By definition, the relationship between the original and permuted entries of the ordered sets is given by: $x'^{(n)} = x^{\sigma^{-1}(n)}$ and $e'^{(ij)} = e^{(\sigma^{-1}(i), \sigma^{-1}(j))}$.

**Permutation Equivariant Model**    We begin by proving that the DeFoG model is permutation-equivariant, including the network architecture and the additional features employed.

- Permutation Equivariance of RRWP Features: Recall that the RRWP features until $K - 1$ steps are defined as $\text{RRWP}(M) = P = [I, M, \ldots, M^{K-1}] \in \mathbb{R}^{n \times n \times K}$, where $M^k = (D^{-1}A)^k$, $0 \leq k < K$.

  We first prove that $M(A) = D^{-1}A$ is permutation equivariant:

  $$
  \begin{aligned}
  M(A')^{(ij)} &= (D'^{-1}A')^{(ij)} \\
  &= (1/(D')^{(ii)})(A')^{(ij)} \\
  &= (1/D^{(\sigma^{-1}(i), \sigma^{-1}(i))})(A)^{(\sigma^{-1}(i), \sigma^{-1}(j))} \\
  &= (D^{-1}A)^{(\sigma^{-1}(i), \sigma^{-1}(j))} \\
  &= (M(A)')^{(ij)}.
  \end{aligned}
  $$

  To facilitate notation, in the following proofs, we consider the matrix $\pi \in \{0, 1\}^{N \times N}$ representing the same permutation function $\sigma$, with the permuted features represented as $\pi M \pi^T$. We then prove that RRWP is permutation equivariant.

  $$
  \begin{aligned}
  P(\pi M \pi^T) &= [\pi I \pi^T, \pi M \pi^T, \ldots, (\pi M \pi^T)^{K-1}] \\
  &= [\pi I \pi^T, \pi M \pi^T, \ldots, (\pi M^{K-1} \pi^T)], \quad \text{since } \pi^T \pi = \mathbf{I}_N \\
  &= \pi [I, M, \ldots, M^{K-1}] \pi^T \\
  &= \pi P(M) \pi^T.
  \end{aligned}
  $$

- Permutation Equivariance of Model Layers: The model layers (MLP, FiLM, PNA, and self-attention) preserve permutation equivariance, as shown in prior work (e.g., Vignac et al. (2022) in Lemma 3.1).

Hence, since all of its components are permutation-equivariant, so is the DeFoG full architecture.

**Permutation Invariant Loss Function**    DeFoG's loss consists of summing the cross-entropy loss between the predicted clean graph and the true clean graph (node and edge-wise). Vignac et al. (2022), Lemma 3.2, provide a concise proof that this loss is permutation invariant.

**Permutation Invariant Sampling Probability**    Given a noisy graph $G_0$ sampled from the initial distribution, $p_0$, the rate matrix at any time point $t \in [0, 1]$ defines the denoising process.

Recall that the conditional rate matrix for a variable $z_t$ at each time step $t$ is defined as:

$$
R_t^*(z_t, z_{t+dt}|z_1) = \frac{\text{ReLU}\left[\partial_t p_{t|1}(z_{t+dt}|z_1) - \partial_t p_{t|1}(z_t|z_1)\right]}{Z_t^{>0} \, p_{t|1}(z_t|z_1)}.
$$

In our multivariate formulation, we compute this rate matrix independently for each variable inside the graph. We denote the concatenated rate matrix entries for all nodes with $R_t^*(x_t'^{(1:N)}, x_{t+dt}'^{(1:N)}|x_1'^{(1:N)})$.

In the following part, we demonstrate the node permutation equivariance of the rate matrix predicted by the trained equivariant network, denoted by $f_\theta$. The proof for edges follows a similar logic. Suppose that the noisy graph $G_t$ is permuted, and the permuted graph has nodes denoted by $x_t'^{(1:N)}$, i.e., $x_t'^{(n)} = x_t^{\sigma^{-1}(n)}$. We have:

$$R_t^* \left( x_t'^{(1:N)}, x_{t+\mathrm{d}t}'^{(1:N)} | x_1'^{(1:N)} \right)^{(n)}$$

$$= R_t^* \left( x_t'^{(n)}, x_{t+\mathrm{d}t}'^{(n)} | x_1'^{(n)} \right)$$

$$= \frac{\mathrm{ReLU} \left[ \partial_t p_{t|1}(x_{t+\mathrm{d}t}'^{(n)} | x_1'^{(n)}) - \partial_t p_{t|1}(x_t'^{(n)} | x_1'^{(n)}) \right]}{Z_t^{>0} \, p_{t|1}(x_t'^{(n)} | x_1'^{(n)})}$$

$$= \frac{\mathrm{ReLU} \left[ \partial_t p_{t|1}(x_{t+\mathrm{d}t}^{\sigma^{-1}(n)} | x_1'^{(n)}) - \partial_t p_{t|1}(x_t^{\sigma^{-1}(n)} | x_1'^{(n)}) \right]}{Z_t^{>0} \, p_{t|1}(x_t^{\sigma^{-1}(n)} | x_1'^{(n)})}, \tag{33}$$

$$= \frac{\mathrm{ReLU} \left[ \partial_t p_{t|1}(x_{t+\mathrm{d}t}^{\sigma^{-1}(n)} | x_1^{\sigma^{-1}(n)}) - \partial_t p_{t|1}(x_t^{\sigma^{-1}(n)} | x_1^{\sigma^{-1}(n)}) \right]}{Z_t^{>0} \, p_{t|1}(x_t^{\sigma^{-1}(n)} | x_1^{\sigma^{-1}(n)})}, \tag{34}$$

$$= R_t^*(x_t^{\sigma^{-1}(n)}, x_{t+\mathrm{d}t}^{\sigma^{-1}(n)} | x_1^{\sigma^{-1}(n)})$$

$$= R_t^*(x_t^{(1:N)}, x_{t+\mathrm{d}t}^{(1:N)} | x_1^{(1:N)})^{\sigma^{-1}(n)}.$$

In Eq. (33), we use the definition of permuted ordered set for $x_t'$ and $x_{t+\mathrm{d}t}'$, and, in Eq. (34), we use that $f_\theta$ is equivariant.

Furthermore, the transition probability at each time step $t$ is given by:

$$\tilde{p}_{t+\Delta t|t}(G_{t+\Delta t}^{1:D} | G_t^{1:D}) = \prod_{d=1}^{D} \tilde{p}_{t+\Delta t|t}^{(d)}(G_{t+\Delta t}^{(d)} | G_t^{1:D})$$

$$= \prod_{d=1}^{D} \left( \delta(G_t^{(d)}, G_{t+\Delta t}^{(d)}) + \mathbb{E}_{p^\theta(G_1^{(d)} | G_t^{1:D})} \left[ R_t^{(d)}(G_t^{(d)}, G_{t+\Delta t}^{(d)} | G_1^{(d)}) \right] \Delta t \right).$$

The transition probability $\tilde{p}_{t+\Delta t|t}(G_{t+\Delta t}^{1:D} | G_t^{1:D})$ is expressed as a product over all nodes and edges, an operation that is inherently a permutation invariant function with respect to node ordering. Furthermore, as demonstrated earlier, the term $\mathbb{E}_{p^\theta(G_1^{(d)} | G^{1:D}t)} \left[ R_t^{(d)}(G_t^{(d)}, G_{t+\Delta t}^{(d)} | G_1^{(d)}) \right]$ is permutation equivariant since $R_t^{(d)}$ and $p_{t|1}^\theta(G_1^{(d)} | G_t^{1:D})$ are both permutation equivariant if model $f_\theta$ is permutation equivariant. Consequently, since the composition of these components yields a permutation invariant function, we conclude that the transition probability of the considered CTMC is permutation invariant.

We finally verify the final sampling probability, $\tilde{p}_1(G_1^{1:D})$, is permutation-invariant. To simulate $G_1^{1:D}$ over $K$ time steps $[0 = t_0, t_1, \ldots, t_K = 1]$, we can marginalize it first by taking the expectation over the state at the last time step $t_{K-1}$. Specifically, we have $p_1(G_1^{1:D}) = \mathbb{E}_{p_{t_{K-1}}(G_{t_{K-1}}^{1:D})} \left[ \tilde{p}_{1|t_{K-1}}(G_1^{1:D} | G_{t_{K-1}}^{1:D}) \right]$. Since the process is Markovian, this expression can be sequentially extended over the $T$ steps through successive expectations.

$$p_1(G_1^{1:D}) = \mathbb{E}_{p_{t_{K-1}}(G_{t_{K-1}}^{1:D})} \left[ \tilde{p}_{1|t_{K-1}}(G_1^{1:D} | G_{t_{K-1}}^{1:D}) \right]$$

$$= \mathbb{E}_{p_{t_{K-2}}(G_{t_{K-2}}^{1:D})} \underbrace{\left[ \mathbb{E}_{\tilde{p}_{t_{K-1}|t_{K-2}}(G_{t_{K-1}}^{1:D} | G_{t_{K-2}}^{1:D})} \left[ \tilde{p}_{1|t_{K-1}}(G_1^{1:D} | G_{t_{K-1}}^{1:D}) \right] \right]}_{\tilde{p}_{1|t_{K-2}}(G_1^{1:D} | G_{t_{K-2}}^{1:D})}$$

$$\ldots$$

$$= \mathbb{E}_{p_0(G_0^{1:D})} \left[ \mathbb{E}_{\tilde{p}_{t_1|0}(G_{t_1}^{1:D} | G_0^{1:D})} \left[ \ldots \mathbb{E}_{\tilde{p}_{t_{K-1}|t_{K-2}}(G_{t_{K-1}}^{1:D} | G_{t_{K-2}}^{1:D})} \left[ \tilde{p}_{1|t_{K-1}}(G_1^{1:D} | G_{t_{K-1}}^{1:D}) \right] \right] \right]$$

Due to the fact that each function in the sequence is itself permutation-invariant and that the initial distribution $p_0(G_0^{1:D})$ is permutation invariant (see Sec. 4.2.1), the composition of permutation-invariant functions preserves this invariance throughout. Thus, the final sampling probability is invariant over isomorphic graphs. $\square$

# E  CONDITIONAL GENERATION

In this section, we describe how to seamlessly integrate DeFoG with existing methods for CTMC-based conditioning mechanisms. In this setting, all the examples are assumed to have a label. The objective of conditional generation is to steer the generative process based on that label, so that at sampling time we can guide the model to which class of samples we are interested in obtaining.

We focus on classifier-free guidance methods, as these models streamline training by avoiding task-specific classifiers. This approach has been widely adopted for continuous state-space models, e.g., for image generation, where it has been shown to enhance the generation quality of the generative model (Ho & Salimans, 2021; Sanchez et al., 2024). Recently, Nisonoff et al. (2024) extended this method to discrete flow matching in a principled manner. In this paper, we adopt their formulation.

In this framework, 90% of the training is performed with the model having access to the label of each noisy sample. This allows the model to learn the conditional rate matrix, $R_t^\theta(z_t, z_{t+\mathrm{d}t}|y)$. In the remaining 10% of the training procedure, the labels of the samples are masked, forcing the model to learn the unconditional generative rate matrix $R_t^\theta(z_t, z_{t+\mathrm{d}t})$. The conditional training enables targeted and accurate graph generation, while the unconditional phase ensures robustness when no conditions are specified. The combination of both conditional and unconditional training offers a more accurate pointer to the conditional distribution, typically described by the distance between the conditional and unconditional prediction. In our framework, this pointer is defined through the ratio between the conditional and unconditional rate matrices, as follows:

$$R_t^{\theta,\gamma}(x,\tilde{x}|y) = R_t^\theta(x,\tilde{x}|y)^\gamma R_t^\theta(x,\tilde{x})^{1-\gamma} = R_t^\theta(x,\tilde{x}|y)\left(\frac{R_t^\theta(x,\tilde{x}|y)}{R_t^\theta(x,\tilde{x})}\right)^{\gamma-1},$$

where $\gamma$ denotes the guidance weight. In particular, the case with $\gamma = 1$ corresponds to standard conditional generation, while $\gamma = 0$ represents standard unconditional generation. As $\gamma$ increases, the conditioning effect described by $\left(\frac{R_t^\theta(x,\tilde{x}|y)}{R_t^\theta(x,\tilde{x})}\right)^{\gamma-1}$ is strengthened, thereby enhancing the quality of the generated samples. We observed $\gamma = 2.0$ to be the best performing value for our digital pathology experiments (Sec. 6.4), as detailed in Tab. 6.

Overall, conditional generation is pivotal for guiding models to produce graphs that meet specific requirements, offering tailored solutions for complex real-world tasks. The flexibility of DeFoG, being well-suited for conditional generation, marks an important step forward in advancing this direction, promising greater adaptability and precision in future graph-based applications.

# F EXPERIMENTAL DETAILS

This section provides further details on the experimental settings used in the paper.

## F.1 DATASET DETAILS

### F.1.1 SYNTHETIC DATASETS

Here, we describe the datasets employed in our experiments and outline the specific metrics used to evaluate model performance on each dataset. Additional visualizations of example graphs from each dataset, along with generated graphs, are provided in Figures 15 to 17.

**Description** We use three synthetic datasets with distinct topological structures. The first is the *planar* dataset (Martinkus et al., 2022), which consists of connected planar graphs—graphs that can be drawn on a plane without any edges crossing. The second dataset, *tree* (Bergmeister et al., 2023), contains tree graphs, which are connected graphs with no cycles. Lastly, the *Stochastic Block Model (SBM)* dataset (Martinkus et al., 2022) features synthetic clustering graphs where nodes within the same cluster have a higher probability of being connected.

The *planar* and *tree* datasets exhibit well-defined deterministic graph structures, while the *SBM* dataset, commonly used in the literature, stands out due to its stochasticity, resulting from the random sampling process that governs its connectivity.

**Metrics** We follow the evaluation procedures described by Martinkus et al. (2022); Bergmeister et al. (2023), using both dataset-agnostic and dataset-specific metrics.

First, dataset-agnostic metrics assess the alignment between the generated and training distributions for specific general graph properties. We map the graphs to their node degrees (Deg.), clustering coefficients (Clus.), orbit count (Orbit), eigenvalues of the normalized graph Laplacian (Spec.), and statistics derived from a wavelet graph transform (Wavelet). We then compute the distance to the corresponding statistics calculated for the test graphs. For each statistic, we measure the distance between the empirical distributions of the generated and test sets using Maximum Mean Discrepancy (MMD). These distances are aggregated into the *Ratio* metric. To compute this, we first calculate the MMD distances between the training and test sets for the same graph statistics. The final Ratio metric is obtained by dividing the average MMD distance between the generated and test sets by the average MMD distance between the training and test sets. A Ratio value of 1 is ideal, as the distance between the training and test sets represents a lower-bound reference for the generated data's performance.

Next, we report dataset-specific metrics using the V.U.N. framework, which assesses the proportion of graphs that are valid (V), unique (U), and novel (N). Validity is assessed based on dataset-specific properties: the graph must be planar, a tree, or statistically consistent with an SBM for the planar, tree, and SBM datasets, respectively. Uniqueness captures the proportion of non-isomorphic graphs within the generated graphs, while novelty measures how many of these graphs are non-isomorphic to any graph in the training set.

### F.1.2 MOLECULAR DATASETS

**Description** Molecular generation is a key real-world application of graph generation. It poses a challenging task to current graph generation models to their rich chemistry-specific information, involving several nodes and edges classes and leaning how to generate them jointly, and more complex evaluation pipelines. To assess DeFoG's performance on molecular datasets, we use three benchmarks that progressively increase in molecular complexity and size.

First, we use the QM9 dataset (Wu et al., 2018), a subset of GDB9 (Ruddigkeit et al., 2012), which contains molecules with up to 9 heavy atoms.

Next, we evaluate DeFoG on the Moses benchmark (Polykovskiy et al., 2020), derived from the ZINC Clean Leads collection (Sterling & Irwin, 2015), featuring molecules with 8 to 27 heavy atoms, filtered by specific criteria.

Finally, we include the Guacamol benchmark (Brown et al., 2019), based on the ChEMBL 24 database (Mendez et al., 2019). This dataset comprises synthesized molecules, tested against biological targets, with sizes ranging from 2 to 88 heavy atoms.

Next, we evaluate DeFoG on the Moses benchmark (Polykovskiy et al., 2020), derived from the ZINC Clean Leads collection (Sterling & Irwin, 2015), which contains molecules ranging from 8 to 27 heavy atoms, filtered according to certain criteria. Lastly, we include the Guacamol benchmark (Brown et al., 2019), derived from the ChEMBL 24 database (Mendez et al., 2019). This benchmark contains only molecules that have been synthesized and tested against biological targets, with 2 to 88 heavy atoms.

**Metrics**    For the QM9 dataset, we follow the dataset splits and evaluation metrics outlined by Vignac et al. (2022). For the Moses and Guacamol benchmarks, we adhere to the training setups and evaluation metrics proposed by Polykovskiy et al. (2020) and Brown et al. (2019), respectively. Note that Guacamol includes molecules with charges; therefore, the generated graphs are converted to charged molecules based on the relaxed validity criterion used by Jo et al. (2022) before being translated to their corresponding SMILES representations. The validity, uniqueness, and novelty metrics reported by the Guacamol benchmark are actually V, V.U., and V.U.N., and are referred to directly as V, V.U., and V.U.N. in the table for clarity.

### F.1.3    DIGITAL PATHOLOGY DATASETS

**Description**    Graphs, with their natural ability to represent relational data, are widely used to capture spatial biological dependencies in tissue images. This approach has proven successful in digital pathology tasks such as microenvironment classification (Wu et al., 2022), cancer classification (Pati et al., 2022), and decision explainability (Jaume et al., 2021). More recently, graph-based methods have been applied to generative tasks (Madeira et al., 2023), and an open-source dataset was made available by Madeira et al. (2024). This dataset consists of cell graphs where the nodes represent biological cells, categorized into 9 distinct cell types (node classes), and edges model local cell-cell interactions (a single class). For further details, refer to Madeira et al. (2024).

**Metrics**    Each cell graph in the dataset can be mapped to a TLS (Tertiary Lymphoid Structure) embedding, denoted as $\kappa = [\kappa_0, \ldots, \kappa_5] \in \mathbb{R}^6$, which quantifies its TLS content. A graph $G$ is classified as having low TLS content if $\kappa_1(G) < 0.05$, and high TLS content if $\kappa_2(G) > 0.05$. Based on these criteria, the dataset is split into two subsets: high TLS and low TLS. In prior work, TLS generation accuracy was evaluated by training generative models on these subsets separately, and verifying if the generated graphs matched the corresponding TLS content label. We compute TLS accuracy as the average accuracy across both subsets. For DeFoG, we conditionally train it on both subsets simultaneously, as described in Appendix E, and compute TLS accuracy based on whether the generated graphs adhere to the conditioning label. Additionally, we report the V.U.N. metric (valid, unique, novel), similar to what is done for the synthetic datasets (see Appendix F.1.1). A graph is considered valid in this case if it is a connected planar graph, as the graphs in these datasets were constructed using Delaunay triangulation.

### F.2    RESOURCES

The training and sampling times for the different datasets explored in this paper are provided in Tab. 5. All the experiments in this work were run on a single NVIDIA A100-SXM4-80GB GPU.

### F.3    HYPERPARAMETER TUNING

The default hyperparameters for training and sampling for each dataset can be found in the provided code repository. In Tab. 6, we specifically highlight their values for the proposed training (see Sec. 4.2.1) and sampling (see Sec. 4.2.2) strategies, and conditional guidance parameter (see Appendix E). As the training process is by far the most computationally costly stage, we aim to minimize changes to the default model training configuration. Nevertheless, we demonstrate the effectiveness of these modifications on certain datasets:

Table 5: Training and sampling time on each dataset.

| Dataset | Min Nodes | Max Nodes | Training Time (h) | Graphs Sampled | Sampling Time (h) |
|---|---|---|---|---|---|
| Planar | 64 | 64 | 29 | 40 | 0.07 |
| Tree | 64 | 64 | 8 | 40 | 0.07 |
| SBM | 44 | 187 | 75 | 40 | 0.07 |
| QM9 | 2 | 9 | 6.5 | 10000 | 0.2 |
| QM9(H) | 3 | 29 | 55 | 10000 | 0.4 |
| Moses | 8 | 27 | 46 | 25000 | 5 |
| Guacamol | 2 | 88 | 141 | 10000 | 7 |
| TLS | 20 | 81 | 38 | 80 | 0.15 |

Table 6: Training and sampling parameters for full-step sampling (500 or 1000 steps for synthetic and molecular datasets respectively).

| | Train | | Sampling | | | | Conditional |
|---|---|---|---|---|---|---|---|
| Dataset | Initial Distribution | Train Distortion | Sample Distortion | $\omega$ (Target Guidance) | $\eta$ (Stochasticity) | Exact Exp | $\gamma$ |
| Planar | Marginal | Identity | Polydec | 0.05 | 50 | True | — |
| Tree | Marginal | Polydec | Polydec | 0.0 | 0.0 | False | — |
| SBM | Absorbing | Identity | Identity | 0.0 | 0.0 | False | — |
| QM9 | Marginal | Identity | Polydec | 0.0 | 0.0 | False | — |
| QM9(H) | Marginal | Identity | Polydec | 0.05 | 0.0 | False | — |
| Moses | Marginal | Polydec | Polydec | 0.5 | 200 | False | — |
| Guacamol | Marginal | Polydec | Polydec | 0.1 | 300 | False | — |
| TLS | Marginal | Identity | Polydec | 0.05 | 0.0 | False | 2.0 |

1. SBM performs particularly well with absorbing distributions, likely due to its distinct clustering structure, which differs from other graph properties. Additionally, when tested with a marginal model, SBM can achieve a V.U.N. of 80.5% and an average ratio of 2.5, which also reaches state-of-the-art performance.

2. Guacamol and MOSES are trained directly with polydec distortion to accelerate convergence, as these datasets are very large and typically require a significantly longer training period.

3. For the tree dataset, standard training yielded suboptimal results (85.3% for V.U.N. and 1.8 for average ratio). However, a quick re-training using polydec distortion achieved state-of-the-art performance with 7 hours of training.

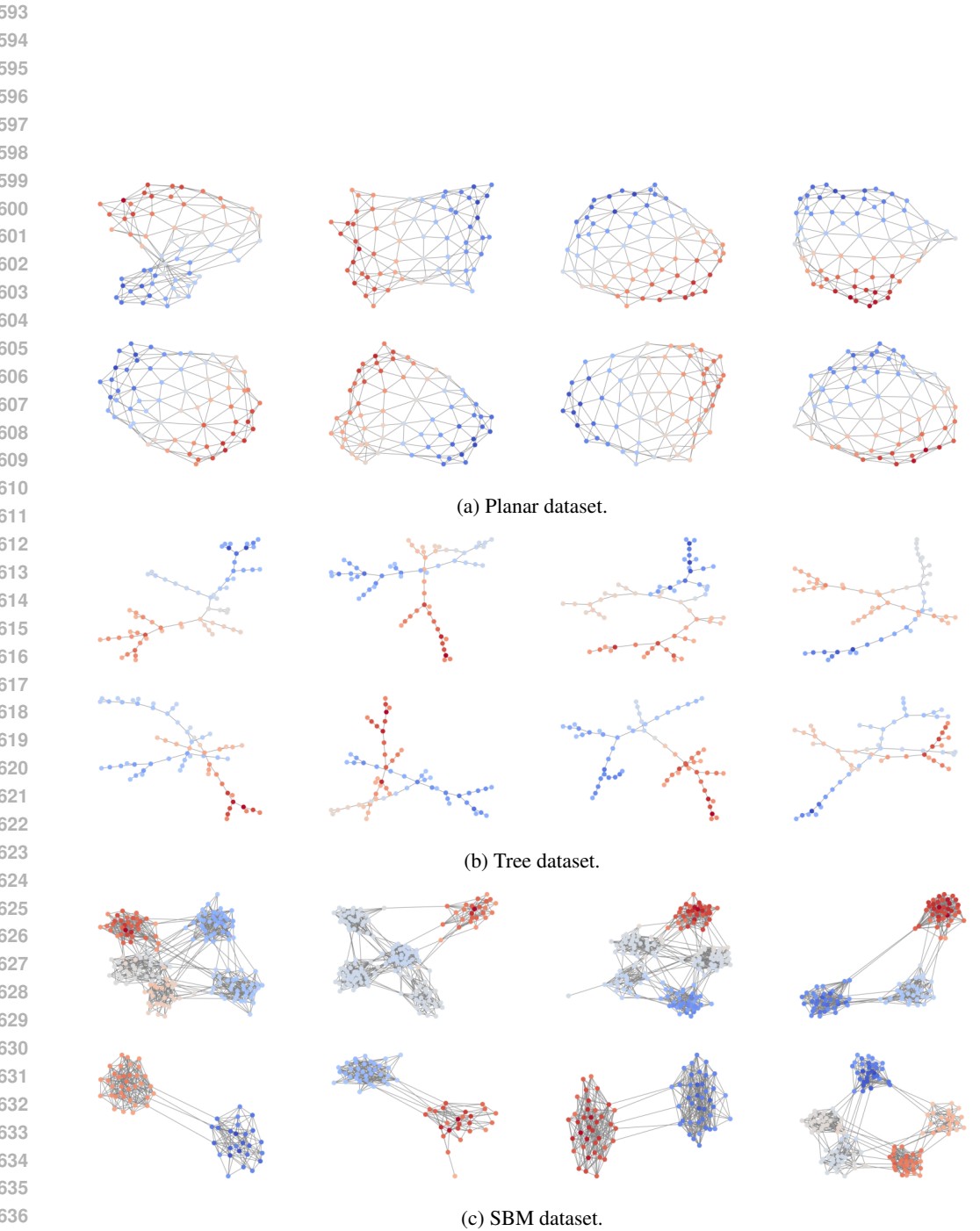

(a) Planar dataset.

(b) Tree dataset.

(c) SBM dataset.

Figure 15: Uncurated set of dataset graphs (top) and generated graphs by DeFoG (bottom) for the synthetic datasets.

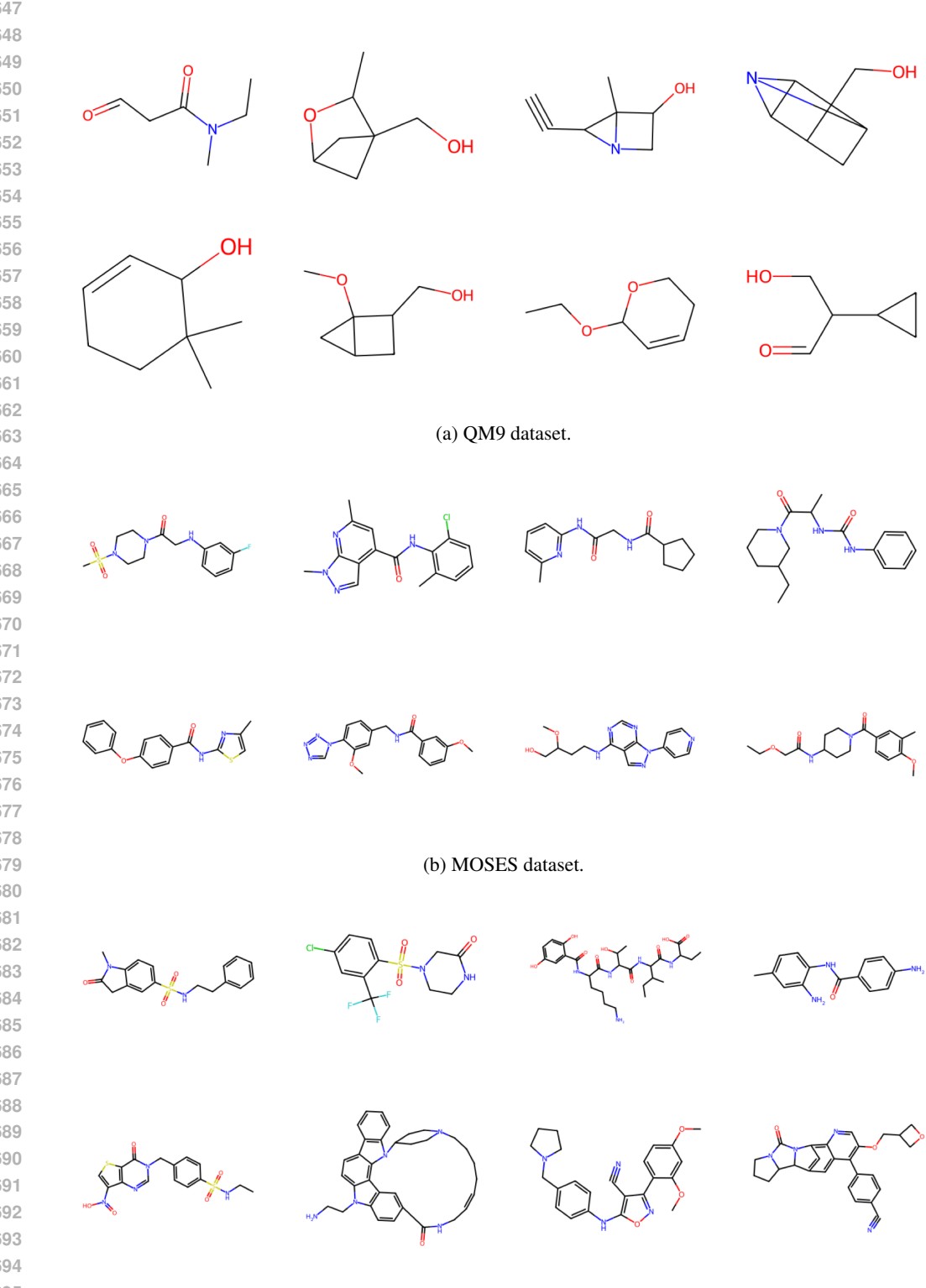

(a) QM9 dataset.

(b) MOSES dataset.

(c) Guacamol dataset.

Figure 16: Uncurated set of dataset graphs (top) and generated graphs by DeFoG (bottom) for the molecular datasets.

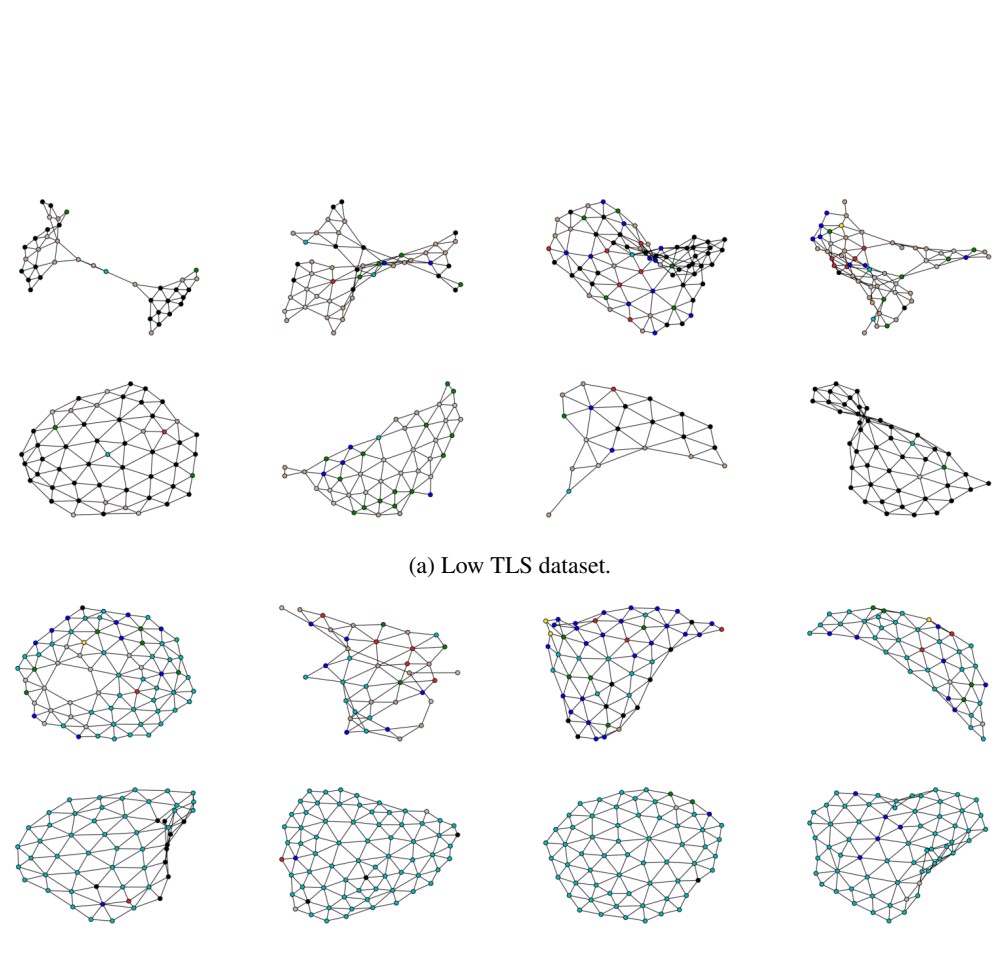

(a) Low TLS dataset.

(b) High TLS dataset.

Figure 17: Uncurated set of dataset graphs (top) and generated graphs by DeFoG (bottom) for the TLS dataset.

# G ADDITIONAL RESULTS

In this section, we present additional experimental results. We begin with the complete tables showcasing results for synthetic datasets in Appendix G.1. Next, Appendix G.2 focuses on molecular generation tasks, including results for the QM9 dataset and the full tables for MOSES and Guacamol. Finally, in Appendix G.3, we analyze the time complexity of various additional features that enhance expressivity in graph diffusion models.

## G.1 SYNTHETIC GRAPH GENERATION

In Tab. 7, we present the full results for DeFoG for the three different datasets: planar, tree, and SBM.

Table 7: Graph Generation Performance on Synthetic Graphs. We present the results of DeFoG across five sampling runs, each generating 40 graphs, reported as mean ± standard deviation. The remaining values are obtained from Bergmeister et al. (2023). Additionally, we include results for Cometh (Siraudin et al., 2024) and DisCo (Xu et al., 2024). For the average ratio computation, we adhere to the method outlined by Bergmeister et al. (2023), excluding statistics where the training set MMD is zero.

| Model | Deg. ↓ | Clus. ↓ | Orbit ↓ | Spec. ↓ | Wavelet ↓ | Ratio ↓ | Valid ↑ | Unique ↑ | Novel ↑ | V.U.N. ↑ |
|---|---|---|---|---|---|---|---|---|---|---|
| *Planar Dataset* | | | | | | | | | | |
| Train set | 0.0002 | 0.0310 | 0.0005 | 0.0038 | 0.0012 | 1.0 | 100 | 100 | 0.0 | 0.0 |
| GraphRNN (You et al., 2018) | 0.0049 | 0.2779 | 1.2543 | 0.0459 | 0.1034 | 490.2 | 0.0 | 100 | 100 | 0.0 |
| GRAN (Liao et al., 2019) | 0.0007 | 0.0426 | 0.0009 | 0.0075 | 0.0019 | 2.0 | 97.5 | 85.0 | 2.5 | 0.0 |
| SPECTRE (Martinkus et al., 2022) | 0.0005 | 0.0785 | 0.0012 | 0.0112 | 0.0059 | 3.0 | 25.0 | 100 | 100 | 25.0 |
| DiGress (Vignac et al., 2022) | 0.0007 | 0.0780 | 0.0079 | 0.0098 | 0.0031 | 5.1 | 77.5 | 100 | 100 | 77.5 |
| EDGE (Chen et al., 2023) | 0.0761 | 0.3229 | 0.7737 | 0.0957 | 0.3627 | 431.4 | 0.0 | 100 | 100 | 0.0 |
| BwR (Diamant et al., 2023) | 0.0231 | 0.2596 | 0.5473 | 0.0444 | 0.1314 | 251.9 | 0.0 | 100 | 100 | 0.0 |
| BiGG (Dai et al., 2020) | 0.0007 | 0.0570 | 0.0367 | 0.0105 | 0.0052 | 16.0 | 62.5 | 85.0 | 42.5 | 5.0 |
| GraphGen (Goyal et al., 2020) | 0.0328 | 0.2106 | 0.4236 | 0.0430 | 0.0989 | 210.3 | 7.5 | 100 | 100 | 7.5 |
| HSpectre (one-shot) (Bergmeister et al., 2023) | 0.0003 | 0.0245 | 0.0006 | 0.0104 | 0.0030 | 1.7 | 67.5 | 100 | 100 | 67.5 |
| HSpectre (Bergmeister et al., 2023) | 0.0005 | 0.0626 | 0.0017 | 0.0075 | 0.0013 | 2.1 | 95.0 | 100 | 100 | 95.0 |
| GruM (Jo et al., 2024) | 0.0004 | 0.0301 | 0.0002 | 0.0104 | 0.0020 | 1.8 | — | — | — | 90.0 |
| CatFlow (Eijkelboom et al., 2024) | 0.0003 | 0.0403 | 0.0008 | — | — | — | — | — | — | 80.0 |
| DisCo (Xu et al., 2024) | 0.0002 ±0.0001 | 0.0403 ±0.0155 | 0.0009 ±0.0004 | — | — | — | 83.6 ±2.1 | 100.0 ±0.0 | 100.0 ±0.0 | 83.6 ±2.1 |
| Cometh - PC (Siraudin et al., 2024) | 0.0006 ±0.0005 | 0.0434 ±0.0093 | 0.0016 ±0.0006 | 0.0049 ±0.0008 | — | — | 99.5 ±0.9 | 100.0 ±0.0 | 100.0 ±0.0 | 99.5 ±0.9 |
| DeFoG | 0.0005 ±0.0002 | 0.0501 ±0.0149 | 0.0006 ±0.0004 | 0.0072 ±0.0011 | 0.0014 ±0.0002 | 1.6 ±0.4 | 99.5 ±1.0 | 100.0 ±0.0 | 100.0 ±0.0 | 99.5 ±1.0 |
| *Tree Dataset* | | | | | | | | | | |
| Train set | 0.0001 | 0.0000 | 0.0000 | 0.0075 | 0.0030 | 1.0 | 100 | 100 | 0.0 | 0.0 |
| GRAN (Liao et al., 2019) | 0.1884 | 0.0080 | 0.0199 | 0.2751 | 0.3274 | 607.0 | 0.0 | 100 | 100 | 0.0 |
| DiGress (Vignac et al., 2022) | 0.0002 | 0.0000 | 0.0000 | 0.0113 | 0.0043 | 1.6 | 90.0 | 100 | 100 | 90.0 |
| EDGE (Chen et al., 2023) | 0.2678 | 0.0000 | 0.7357 | 0.2247 | 0.4230 | 850.7 | 0.0 | 7.5 | 100 | 0.0 |
| BwR (Diamant et al., 2023) | 0.0016 | 0.1239 | 0.0003 | 0.0480 | 0.0388 | 11.4 | 0.0 | 100 | 100 | 0.0 |
| BiGG (Dai et al., 2020) | 0.0014 | 0.0000 | 0.0000 | 0.0119 | 0.0058 | 5.2 | 100 | 87.5 | 50.0 | 75.0 |
| GraphGen (Goyal et al., 2020) | 0.0105 | 0.0000 | 0.0000 | 0.0153 | 0.0122 | 33.2 | 95.0 | 100 | 100 | 95.0 |
| HSpectre (one-shot) (Bergmeister et al., 2023) | 0.0004 | 0.0000 | 0.0000 | 0.0080 | 0.0055 | 2.1 | 82.5 | 100 | 100 | 82.5 |
| HSpectre (Bergmeister et al., 2023) | 0.0001 | 0.0000 | 0.0000 | 0.0117 | 0.0047 | 4.0 | 100 | 100 | 100 | **100** |
| DeFoG | 0.0002 ±0.0001 | 0.0000 ±0.0000 | 0.0000 ±0.0000 | 0.0108 ±0.0028 | 0.0046 ±0.0004 | 1.6 ±0.0 | 96.5 ±2.6 | 100.0 ±0.0 | 100.0 ±0.0 | 96.5 ±2.6 |
| *Stochastic Block Model ($n_{max} = 187$, $n_{avg} = 104$)* | | | | | | | | | | |
| Training set | 0.0008 | 0.0332 | 0.0255 | 0.0027 | 0.0007 | 1.0 | 85.9 | 100 | 0.0 | 0.0 |
| GraphRNN (You et al., 2018) | 0.0055 | 0.0584 | 0.0785 | 0.0065 | 0.0431 | 14.7 | 5.0 | 100 | 100 | 5.0 |
| GRAN (Liao et al., 2019) | 0.0113 | 0.0553 | 0.0540 | 0.0054 | 0.0212 | 9.7 | 25.0 | 100 | 100 | 25.0 |
| SPECTRE (Martinkus et al., 2022) | 0.0015 | 0.0521 | 0.0412 | 0.0056 | 0.0028 | 2.2 | 52.5 | 100 | 100 | 52.5 |
| DiGress (Vignac et al., 2022) | 0.0018 | 0.0485 | 0.0415 | 0.0045 | 0.0014 | 1.7 | 60.0 | 100 | 100 | 60.0 |
| EDGE (Chen et al., 2023) | 0.0279 | 0.1113 | 0.0854 | 0.0251 | 0.1500 | 51.4 | 0.0 | 100 | 100 | 0.0 |
| BwR (Diamant et al., 2023) | 0.0478 | 0.0638 | 0.1139 | 0.0169 | 0.0894 | 38.6 | 7.5 | 100 | 100 | 7.5 |
| BiGG (Dai et al., 2020) | 0.0012 | 0.0604 | 0.0667 | 0.0059 | 0.0370 | 11.9 | 10.0 | 100 | 100 | 10.0 |
| GraphGen (Goyal et al., 2020) | 0.0550 | 0.0623 | 0.1189 | 0.0182 | 0.1193 | 48.8 | 5.0 | 100 | 100 | 5.0 |
| HSpectre (one-shot) (Bergmeister et al., 2023) | 0.0141 | 0.0528 | 0.0809 | 0.0071 | 0.0205 | 10.5 | 75.0 | 100 | 100 | 75.0 |
| HSpectre (Bergmeister et al., 2023) | 0.0119 | 0.0517 | 0.0669 | 0.0067 | 0.0219 | 10.2 | 45.0 | 100 | 100 | 45.0 |
| GruM (Jo et al., 2024) | 0.0015 | 0.0589 | 0.0450 | 0.0077 | 0.0012 | 1.1 | — | — | — | 85.0 |
| CatFlow (Eijkelboom et al., 2024) | 0.0012 | 0.0498 | 0.0357 | — | — | — | — | — | — | 85.0 |
| DisCo (Xu et al., 2024) | 0.0006 ±0.0002 | 0.0266 ±0.0133 | 0.0510 ±0.0128 | — | — | — | 66.2 ±1.4 | 100.0 ±0.0 | 100.0 ±0.0 | 66.2 ±1.4 |
| Cometh (Siraudin et al., 2024) | 0.0020 ±0.0003 | 0.0498 ±0.0000 | 0.0383 ±0.0051 | 0.0024 ±0.0003 | — | — | 75.0 ±3.7 | 100.0 ±0.0 | 100.0 ±0.0 | 75.0 ±3.7 |
| DeFoG | 0.0006 ±0.0023 | 0.0517 ±0.0012 | 0.0556 ±0.0739 | 0.0054 ±0.0012 | 0.0080 ±0.0024 | 4.9 ±1.3 | 90.0 ±5.1 | 90.0 ±5.1 | 90.0 ±5.1 | **90.0** ±5.1 |

## G.2 MOLECULAR GRAPH GENERATION

For the molecular generation tasks, we begin by examining the results for QM9, considering both implicit and explicit hydrogens (Vignac et al., 2022). In the implicit case, hydrogen atoms are inferred to complete the valencies, while in the explicit case, hydrogens must be explicitly modeled, making it an inherently more challenging task. The results are presented in Tab. 8. Notably, DeFoG achieves training set validity in both scenarios, representing the theoretical maximum. Furthermore, DeFoG consistently outperforms other models in terms of FCD. Remarkably, even with only 10% of the sampling steps, DeFoG surpasses many existing methods.

Table 8: Molecule generation on QM9. We present the results over five sampling runs of 10000 generated graphs each, in the format mean ± standard deviation.

| Model | Without Explicit Hydrogenes | | | With Explicit Hydrogenes | | |
|---|---|---|---|---|---|---|
| | Valid ↑ | Unique ↑ | FCD ↓ | Valid ↑ | Unique ↑ | FCD ↓ |
| Training set | 99.3 | 99.2 | 0.03 | 97.8 | 99.9 | 0.01 |
| SPECTRE (Martinkus et al., 2022) | 87.3 | 35.7 | — | — | — | — |
| GraphNVP (Madhawa et al., 2019) | 83.1 | **99.2** | — | — | — | — |
| GDSS (Jo et al., 2022) | 95.7 | 98.5 | 2.9 | — | — | — |
| DiGress (Vignac et al., 2022) | 99.0±0.0 | 96.2±0.1 | — | 95.4±1.1 | **97.6**±0.4 | — |
| GruM(Jo et al., 2024) | 99.2 | 96.7 | **0.11** | — | — | — |
| DisCo (Xu et al., 2024) | 99.3±0.6 | — | — | — | — | — |
| Cometh (Siraudin et al., 2024) | **99.6**±0.1 | 96.8±0.2 | 0.25 ±0.01 | — | — | — |
| DeFoG (# sampling steps = 50) | 98.9±0.1 | 96.2±0.2 | 0.26±0.00 | 97.1±0.0 | 94.8±0.0 | 0.31±0.00 |
| DeFoG (# sampling steps = 500) | 99.3±0.0 | 96.3±0.3 | 0.12±0.00 | **98.0**±0.0 | 96.7±0.0 | **0.05**±0.00 |

Additionally, we provide the complete version of Tab. 2, presenting the results for MOSES and Guacamol separately in Tab. 9 and Tab. 10, respectively. We include models from classes beyond diffusion models to better contextualize the performance achieved by DeFoG.

Table 9: Molecule generation on MOSES.

| Model | Class | Val.↑ | Unique. ↑ | Novelty↑ | Filters ↑ | FCD ↓ | SNN ↑ | Scaf ↑ |
|---|---|---|---|---|---|---|---|---|
| Training set | — | 100.0 | 100.0 | 0.0 | 100.0 | 0.01 | 0.64 | 99.1 |
| VAE (Kingma, 2013) | Smiles | 97.7 | 99.8 | 69.5 | **99.7** | **0.57** | **0.58** | 5.9 |
| JT-VAE (Jin et al., 2018) | Fragment | **100.0** | **100.0** | **99.9** | 97.8 | 1.00 | 0.53 | 10.0 |
| GraphInvent (Mercado et al., 2021) | Autoreg. | 96.4 | 99.8 | — | 95.0 | 1.22 | 0.54 | 12.7 |
| DiGress (Vignac et al., 2022) | One-shot | 85.7 | **100.0** | 95.0 | 97.1 | 1.19 | 0.52 | 14.8 |
| DisCo (Xu et al., 2024) | One-shot | 88.3 | **100.0** | 97.7 | 95.6 | 1.44 | 0.50 | 15.1 |
| Cometh (Siraudin et al., 2024) | One-shot | 90.5 | 99.9 | 92.6 | 99.1 | 1.27 | 0.54 | 16.0 |
| DeFoG (# sampling steps = 50) | One-shot | 83.9 | 99.9 | 96.9 | 96.5 | 1.87 | 0.50 | **23.5** |
| DeFoG (# sampling steps = 500) | One-shot | 92.8 | 99.9 | 92.1 | 98.9 | 1.95 | 0.55 | 14.4 |

Table 10: Molecule generation on GuacaMol. We present the results over five sampling runs of 10000 generated graphs each, in the format mean ± standard deviation.

| Model | Class | Val. ↑ | V.U. ↑ | V.U.N. ↑ | KL div↑ | FCD↑ |
|---|---|---|---|---|---|---|
| Training set | — | 100.0 | 100.0 | 0.0 | 99.9 | 92.8 |
| LSTM (Graves & Graves, 2012) | Smiles | 95.9 | 95.9 | 87.4 | **99.1** | **91.3** |
| NAGVAE (Kwon et al., 2020) | One-shot | 92.9 | 88.7 | 88.7 | 38.4 | 0.9 |
| MCTS (Brown et al., 2019) | One-shot | **100.0** | **100.0** | 95.4 | 82.2 | 1.5 |
| DiGress (Vignac et al., 2022) | One-shot | 85.2 | 85.2 | 85.1 | 92.9 | 68.0 |
| DisCo (Xu et al., 2024) | One-shot | 86.6 | 86.6 | 86.5 | 92.6 | 59.7 |
| Cometh (Siraudin et al., 2024) | One-shot | 98.9 | 98.9 | 97.6 | 96.7 | 72.7 |
| DeFoG (# steps = 50) | One-shot | 91.7 | 91.7 | 91.2 | 92.3 | 57.9 |
| DeFoG (# steps = 500) | One-shot | 99.0 | 99.0 | **97.9** | 97.7 | 73.8 |

Table 11: Performance comparison of RRWP-based graph encoding within the DiGress framework.

| Method | Planar | | SBM | | QM9 | | |
|---|---|---|---|---|---|---|---|
| | V.U.N. ↑ | Ratio ↓ | V.U.N. ↑ | Ratio ↓ | Valid ↑ | Unique ↑ | FCD ↓ |
| DiGress | 77.5 | 5.1 | 60.0 | 1.7 | $99.0 \pm 0.0$ | $96.2 \pm 0.1$ | - |
| DiGress (RRWP) | 90.0 | 4.0 | 70.0 | 1.7 | $99.1 \pm 0.1$ | $96.6 \pm 0.2$ | - |
| DeFoG (RRWP, 50 steps) | $95.0 \pm 3.2$ | $3.2 \pm 1.1$ | $86.5 \pm 5.3$ | $2.2 \pm 0.3$ | $98.9 \pm 0.1$ | $96.2 \pm 0.2$ | $0.26 \pm 0.00$ |
| DeFoG (RRWP) | $99.5 \pm 1.0$ | $1.6 \pm 0.4$ | $90.0 \pm 5.1$ | $4.9 \pm 1.3$ | $99.3 \pm 0.0$ | $96.3 \pm 0.3$ | $0.12 \pm 0.00$ |

## G.3 IMPACT OF ADDITIONAL FEATURES

In graph diffusion methods, the task of graph generation is decomposed into a mapping of a graph to a set of marginal probabilities for each node and edge. This problem is typically addressed using a Graph Transformer architecture, which is augmented with additional features to capture structural aspects that the base architecture might struggle to model effectively (Vignac et al., 2022; Xu et al., 2024; Siraudin et al., 2024) otherwise. In this section, we evaluate the impact of using RRWP encodings as opposed to the spectral and cycle encodings (up to 6-cycles) proposed in DiGress (Vignac et al., 2022).

In Tab. 11, we present a performance comparison of these two variants across three datasets: QM9, Planar, and SBM. The results show that RRWP achieves comparable or superior performance within the DiGress framework, validating its effectiveness as a graph encoding method. Notably, despite these improvements, DiGress's performance remains significantly below that of DeFoG on the Planar and SBM datasets, while achieving similar validity and uniqueness on the QM9 dataset.

To further demonstrate the impact of using RRWP on sampling efficiency, we compare the performances of DeFoG, DiGress, and DiGress augmented with RRWP (replacing the original additional features) across a varying number of sampling steps. These results are shown in Figure 18.

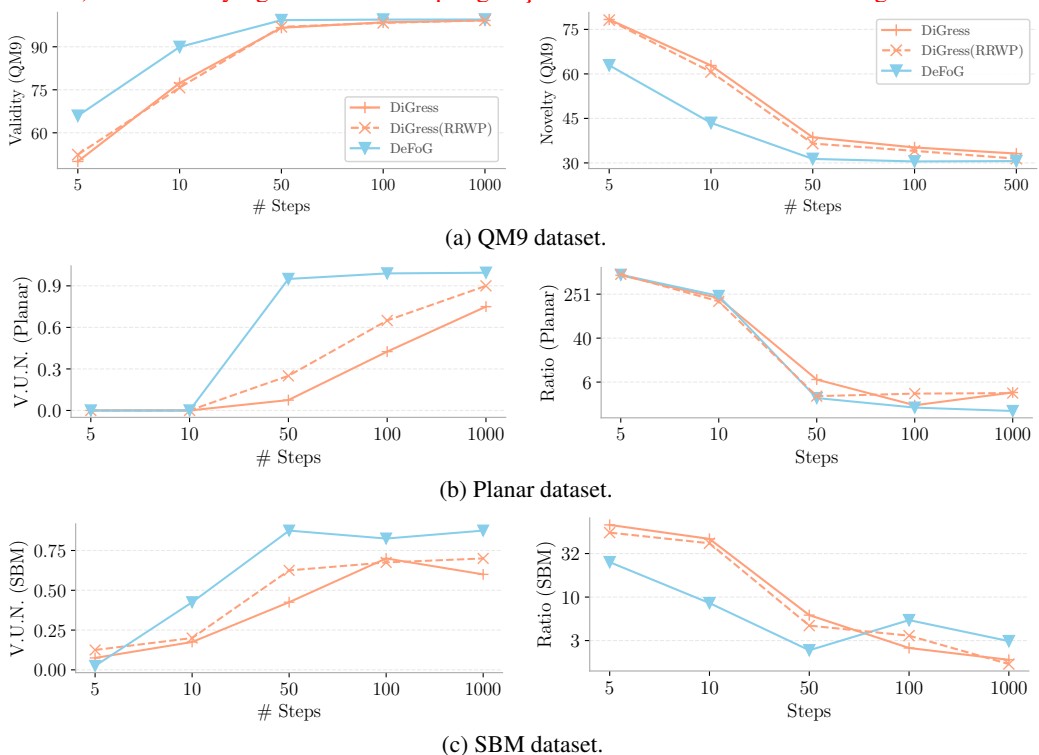

(a) QM9 dataset.

(b) Planar dataset.

(c) SBM dataset.

Figure 18: Impact of RRWP features for sampling efficiency.

We observe that, while RRWP provides improvements on the Planar and SBM datasets with fewer generation steps, it is still significantly outperformed by the optimized DeFoG framework. This highlights that although RRWP is an efficient and effective graph encoding method, the primary

Table 12: Computation time for different additional features. The RRWP features are computed with 12 steps.

| Dataset | Min Nodes | Max Nodes | RRWP (ms) | Cycles (ms) | Spectral (ms) |
|---------|-----------|-----------|-----------|-------------|---------------|
| Moses   | 8         | 27        | 0.6       | 1.2         | 2.5           |
| Planar  | 64        | 64        | 0.5       | 1.2         | 93.0          |
| SBM     | 44        | 187       | 0.6       | 2.7         | 146.9         |

performance gains of DeFoG stem from its continuous-time formulation featuring fully decoupled training and sampling stages.

We then perform a time complexity analysis of these methods. While both cycle and RRWP encodings primarily involve matrix multiplications, spectral encodings require more complex algorithms for eigenvalue and eigenvector computation. As shown in Tab. 12, cycle and RRWP encodings are more computationally efficient, particularly for larger graphs where eigenvalue computation becomes increasingly costly. These results also support the use of RRWP encodings over the combined utilization of cycle and spectral features.

For the graph sizes considered in this work, the additional feature computation time remains relatively small compared to the model's forward pass and backpropagation. However, as graph sizes increase - a direction beyond the scope of this paper - this computational gap could become significant, making RRWP a suitable encoding for scalable graph generative models.

### G.4 COMPUTATION COST FOR EXACT EXPECTATION OF RATE MATRIX

The proposed denoising process iteratively applies the dimension-independent Euler step described in Eq. (4). This requires computing the expectation $\mathbb{E}_{p_{1|t}^{\theta,d}(z_1^d|z_t^{1:D})}\left[R_t^d(z_t^d, z_{t+\Delta t}^d|z_1^d)\right]$. Two procedures are available for this computation: an approximated version, by sequential sampling of conditional distributions, as suggested by Campbell et al. (2024), and an exact version, by evaluating a weighted sum over possible rate matrices:

$$\mathbb{E}_{p_{1|t}^{\theta,d}(z_1^d|z_t^{1:D})}\left[R_t^d(z_t^d, z_{t+\Delta t}^d|z_1^d)\right] = \sum_{z_1^d} p_{1|t}^{\theta,d}(z_1^d|z_t^{1:D})R_t^d(z_t^d, z_{t+\Delta t}^d|z_1^d).$$

This exact computation involves a sweep over all possible states of $z_1^d$. For graphs, this corresponds to iterating over all possible node and edge states at each denoising step. However, since the denoising neural network computes $p_{1|t}^{\theta,d}$ for all $z_1^d$ in a single forward pass (outputting marginal distributions for each node and edge), the exact expectation does not incur additional network calls.

In our experiments, the cardinalities of the node ($X$) and edge ($E$) state spaces are relatively small:

- Synthetic datasets: $X = 1$, $E = 2$;
- Molecular datasets: $X = 4$, $E = 5$;
- Digital Pathology datasets: $X = 2$, $E = 9$.

Tab. 13 summarizes the runtimes for representative datasets. These results show that the computational overhead of the exact expectation in our setting is minimal. Importantly, as anticipated, the overhead increases with the cardinalities of the state spaces.

Table 13: Runtimes for approximate and exact expectation computations across representative datasets. Results are reported in the format mean ± standard deviation over 5 runs.

| Dataset | Approximate (s) | Exact (s) | Overhead (%) |
|---------|-----------------|-----------|--------------|
| Planar (Synthetic)  | $117.92 \pm 0.74$ | $119.83 \pm 0.40$ | $+1.62$ |
| QM9 (Molecular)     | $269.65 \pm 0.90$ | $275.18 \pm 0.86$ | $+2.05$ |
| Digital Pathology   | $629.00 \pm 30.57$ | $667.45 \pm 32.84$ | $+6.11$ |

