# OpenReview forum: "DeFoG: Defogging Discrete Flow Matching for Graph Generation"
_ICLR.cc/2025/Conference — Submitted to ICLR 2025_

### Official Review · Reviewer_G38o · 2024-10-29

**Soundness:** 3
**Presentation:** 2
**Contribution:** 1
**Rating:** 3
**Confidence:** 5

**Summary:**

This paper adapts the Discrete Flow Model (DFM) for graph generation tasks and introduces the DeFog model, which utilizes a graph transformer as its backbone for handling graph data. The authors highlight the importance of a disentangled design for training and inference procedures, which enhances sampling efficiency and performance. Their approach achieves state-of-the-art results across extensive evaluation tasks.

**Strengths:**

1. This paper proposes Discrete Flow Matching (DFM) and achieves good generation results with significant sampling speed ups, on tasks raning from synthetic graph to molecular graph.
2. The writing of the paper is clear and organized, making it easy for readers to follow and understand.

**Weaknesses:**

The primary concern with this paper is its novelty, as it appears to be a straightforward application of Discrete Flow Models (DFMs) as in paper [1]. The technical advancements in the DFM model are either too trivial or lack sufficient justification.

1. The crucial decoupling technique that enhances efficiency and performance, using a sampling procedure different from training, is based on prior work rather than being an original invention.
2. The decoupling technique lacks justification. Theoretically, the paper does not adequately explain why the proposed decoupling would improve sampling efficiency and performance. Empirically, it lacks quantitative measures demonstrating its impact on performance.
3. In Table 1, metrics that measure the similarity between generated population and data distribution are not present, other MMD indicators, for example, Degree, Clustering coefficient etc are not considered in this paper.
4. The necessity of several Lemmas and Theorems in the paper is questionable. For instance, Lemma 1 concerning permutation equivariance and invariance for a transformer without positional embedding seems too obvious to formalize. Additionally, the practical need for an equivariance guarantee in a logical graph is not apparent, and the theoretical guarantees in Section 5 do not clearly connect with the main idea of the paper.
5. The paper's notation requires careful revision. For example, in Equation 4 (#189), the subscript of the expectation $p_{t|1}$ should be $p_{1|t}$. The notation $x^{1:n:N}$ at #219 seems peculiar; $x^{(n)}$ might be clearer than $x^n$. At #229, $p_{1|t}^\theta$ is a scalar rather than a set, and it is suggested to use boldface for $\theta$ as it represents a non-scalar value.

**Questions:**

1. In line #223, while the relationship between nodes and edges is managed by a graph transformer, what additional graph-specific challenges should be considered?
2. In line #278, what does the term "design space" refer to? Is it related to the data distribution or the noisy exploration space?
3. In line #280, the paper claims that flow models enable greater flexibility and efficiency due to decoupling the training and sampling procedures. This isn't immediately apparent; why would a decoupled procedure enhance sampling speed?
4. In lines #288-289, how is the factorization of graph probability related to permutation invariance?
5. In lines #334-337, shouldn't the new $R_t$ be normalized?

[1] Andrew Campbell, Jason Yim, Regina Barzilay, Tom Rainforth, Tommi Jaakkola. "Generative Flows on Discrete State-Spaces:
Enabling Multimodal Flows with Applications to Protein Co-Design"

---

> ### Author Response · Authors · 2024-11-20
>
> We appreciate the detailed comments and pertinent questions. We address the raised concerns below:
>
> Regarding the novelty concerns of our work, we refer to the points **Novelty** and **Graph-specificity** of the general comment for a detailed response. Our work introduces new theoretical results and algorithmic innovations specifically designed for graph generation. We provide new theoretical guarantees for both training and sampling stages, including a direct link between the training loss and rate matrix estimation error, along with an upper bound on the sampling error. Furthermore, we address graph-specific challenges by ensuring DeFoG’s permutation equivariance in a fundamentally different context from the original DFM framework. We also explore under-studied aspects of the sampling process for graph generation, such as stochasticity configurations, and propose novel techniques, including target guidance, to enhance DeFoG’s design space, which are critical for achieving state-of-the-art performance. These contributions highlight DeFoG's suitability as a robust framework for graph generative modeling.
>
> **W1**: Indeed, the DFM framework, which we build upon, allows us to decouple the training and sampling procedures, a key property for efficiently optimizing the sampling process. However, DFM does not work out of the box for graph generation, as it can be observed for the performance of vanilla DeFoG for the different settings analysed (see, for example, Figs. 2, 6 and 7). Our exploration of DeFoG's design space is crucial for the observed state-of-the-art performance.
> While components like stochasticity during sampling via $R^\text{DB}$ [6] build on prior work, our key innovations, such as target guidance and sample distortion, significantly improve the performance of the vanilla framework.
> Overall, we believe that our improvements to DeFoG's design space and its extensive exploration are important new contributions to the community, as illustrated by the state-of-the-art results.
>
> **W2**:
> Having a decoupled training and sampling procedure allows for a more efficient exploration and optimization of DeFoG's design space. As mentioned in the Introduction (lines 51-53), this decoupling enables optimization at the sampling stage without requiring re-training, a process that is typically the most computationally expensive. Additionally, this also allows for a larger sampling optimization space, since the rate matrices are no longer implicitly defined at training time. This allows us to explore varying stochasticities and target guidance mechanisms at sampling time, which were not possible in continuous-time discrete diffusion. It is due to these procedures that we obtain better performance with fewer steps, as shown in our experiments.
>
> We highlight in Section 4.2, Exploring DeFoG's Design Space, that this efficient optimization is a significant advantage over existing diffusion-based graph generative models:.
>
>     "While discrete diffusion models offer a broader design space compared to one-shot models, including noise schedules and diffusion trajectories (Austin et al., 2021), its exploration is costly. In contrast, flow models enable greater flexibility and efficiency due to their decoupled noising and sampling procedures. For instance, the number of sampling steps and their sizes are not fixed as in discrete-time diffusion (Vignac et al., 2022), and the rate matrix design can be adjusted independently, unlike continuous-time diffusion models (Siraudin et al., 2024; Xu et al., 2024). In this section, we exploit this rich design space and propose key algorithmic improvements to optimize both the training and sampling stages of DeFoG."
>
> To further address the reviewer’s concerns, we have added a new section in the appendix (Appendix A.2) contextualizing the benefits of the decoupled stages in DeFoG, relative to existing diffusion-based graph generative models. In particular, we added a new Section(A.2.3) that provides a detailed discussion of how the full decoupling of sampling from training enhances both efficiency and performance.
>
> We support this theoretical motivation with extensive experimental results. First, the decoupled optimization enables DeFoG to achieve state-of-the-art performance across all evaluated tasks (Tables 1, 2, and 3). Second, Figures 2, 6, and 7 demonstrate the critical role of the sampling optimization pipeline in DeFoG’s performance, where we also included curves for DiGress across all datasets to provide a more comprehensive comparison in the revised version of the manuscript. Specifically, these experiments show that vanilla DeFoG's configuration is insufficient to outperform existing diffusion-based graph generative models, and the proposed modifications are pivotal for achieving state-of-the-art results. Importantly, such extensive exploration would be infeasible for other diffusion-based graph generative models.

---

> > ### Author Response · Authors · 2024-11-20
> >
> > **W2**(cont.) Finally, Appendix C.2 analyzes the interplay between decoupled training and sampling procedures using different distortion functions for each stage. These experiments challenge the assumption that aligned training and sampling procedures are always beneficial, illustrating that decoupling can, in fact, enhance model performance.
> >
> > **W3** Those results are available in Table 7, as indicated in the caption of Table 1. We did not include them in Table 1 as the Av. Ratio aggregates these MMD indicators.
> >
> > **W4**: We believe Lemma 1 is essential for ensuring the proper behavior of the entire generative model, not just the transformer component. This formalization encompasses the permutation equivariance of the RRWP encodings and CTMC formulation, as well as the permutation invariance of the adopted loss. Additionally, permutation equivariance is a standard requirement in the literature on graph generative models [1, 2, 3, 4, 5].
> >
> > Regarding the term "logical graph," we are unclear about its intended meaning in this context. Could the reviewer kindly clarify or provide additional details?
> >
> > In Section 5, we provide new theoretical guarantees that strengthen DeFoG’s framework and substantiate its training and sampling procedures, which are essential for understanding its empirical success. Specifically:
> >
> > - Theorem 2 justifies the use of the cross-entropy loss function for training, directly linking it to the sampling algorithm.
> > - Theorem 3 validates the approximate CTMC simulation method used during sampling by providing an upper bound on its deviation from the exact simulation.
> >
> > These theoretical foundations enhance the soundness of DeFoG’s framework, offering guarantees for its high performance.
> >
> > **W5**: We thank the reviewer for the suggestions and for noticing the typo error in Equation 4 (line 189). For the remaining preferences on the notation, we address them as follows:
> >
> > - We clarified the notation $x^{1:n:N}$ in the updated version of the manuscript;
> > - We have decided not to replace the notation $x^n$ with $x^{(n)}$ as the notation is already dense and the change would hinder readability;
> > - In line 229, we initially defined $p_{1|t}^\theta(G_t)$ as the set of marginal distributions output by the neural network, not as a scalar. Nevertheless, we agree that the notation could be misleading and we have updated it to $p_{1|t}^\theta(\cdot|G_t)$ to avoid such confusion;
> > - We decided not to bold $\theta$ for consistency with previous literature [2, 3, 4, 5, 6].
> >
> > We have integrated these changes into the updated version of the manuscript.
> >
> > **Q1**: In line 223, we highlight challenges specific to graph modeling that go beyond jointly modeling a set of variables, as nodes and edges play distinct roles in the graph structure and adhere to certain symmetries. For instance, preserving *node* permutation equivariance is critical to ensure that the generative model is expressive yet consistent under graph isomorphisms. To address these challenges, we leverage a graph transformer and RRWP encodings, which are key to DeFoG’s state-of-the-art performance in graph generation. Additionally, we adopt a differentially weighted loss function for nodes and edges, enabling tailored optimization for their distinct contributions to the graph structure.
> >
> > **Q2**: Throughout the paper, we use term "design space" to refer to the range of possible configurations and hyperparameters that can be explored to optimize the performance of a generative model. In the particular context of the mentioned line, it refers to the design space of Discrete Denoising Difusion Probabilistic Models (D3PMs), which includes the noise schedules (how much noise is added at each timestep) and the diffusion trajectories (the different transition matrices thac can be chosen for the forward process). Note, nevertheless, that this design space changes for different models.
> >
> > **Q3**: Decoupling training and sampling enables efficient optimization of DeFoG's sampling design space without requiring computationally expensive retraining. For instance, unlike discrete-time diffusion models, the number and size of sampling steps are not fixed, and, unlike continuous-time diffusion models, the denoising rate matrix can be adjusted independently of training. This independence allows further enhancements to the denoising process, such as controllable trajectory stochasticity or guidance modifications (for continuous-time diffusion models, this is not possible as the rate matrix is implicitly fixed during training; see Appendix A.2.3 for details). By efficiently exploring this enriched sampling design space, DeFoG achieves strong performance with significantly fewer sampling steps, substantially improving sampling speed.

---

> ### Author Response · Authors · 2024-11-20
>
> **Q4**: In lines 288-289, we discuss what is a reasonable initial distribution for graphs.
> This distribution should be permutation invariant, as the order of nodes and edges in a graph is arbitrary. This means that the distribution should have the same probability of yielding any permutation of the same graph. This is a necessary condition to ensure permutation equivariance of the whole model.
> Therefore, we adopt the proposed formulation:
>
> $$
> p_\epsilon = \prod_n p_\epsilon^\mathcal{X} \prod_{i<j} p_\epsilon^\mathcal{E}
> $$
>
> where $p_\epsilon^\mathcal{X}$ and $p_\epsilon^{\mathcal{E}}$ represent shared distributions across nodes and edges, respectively.
>
> (We changed the notation of $p^n_\epsilon$ to $p_\epsilon^{\mathcal{X}}$ and $p_\epsilon^{ij}$ to $p_\epsilon^{\mathcal{E}}$ to make it clearer that is a shared distribution across nodes and edges, respectively).
> This formulation is trivially permutation invariant, while sufficiently flexible to allow different prior incorporation strategies, which we explore in the paper.
> We have updated the main text (lines 290-291) and the proof of Lemma 1 (lines 2372- 2373) to make this connection explicit.
>
> **Q5**:
> We are unclear about what is meant by "normalization" here.
> If it refers to preserving the Kolmogorov equation, we quantify how much deviation is incurred by that component, and there is no need for re-normalization.
> However, if the question refers to the normalization step
> $R_t(z_t, z_t) = -\sum_{z_{t+\mathrm{d}t} \neq z_t} R_t(z_t, z_{t+\mathrm{d}t}),$
> which ensures $\sum_{z_{t+\mathrm{d}t}} p_{t+\mathrm{d}t|t}(z_{t+\mathrm{d}t} | z_t) = 1$,
> then note that $R_t(z_t, z_t)$ is computed only at the end. Specifically, it is calculated after summing all the components of $R_t(z_t, z_{t+\mathrm{d}t})$ for $z_{t+\mathrm{d}t} \neq z_t$, including $R^*_t$, $R^\mathrm{DB}_t$, and $R^\omega_t$. So, it is necessarily normalized.
>
> We hope these clarifications address the raised concerns and provide a more comprehensive overview of our work.
> We have revised our manuscript to better highlight the novelty and contributions of our approach, as well as to clarify notational choices and ensure the accuracy of our framework.
> We remain at your disposal for any further questions or clarifications.
>
> [1] - *Permutation Invariant Graph Generation via Score-Based Generative Modeling*, Niu et al., AISTATS 2020
>
> [2] - *DiGress: Discrete Denoising Diffusion for Graph Generation*, Vignac et al., ICLR 2023
>
> [3] - *Variational Flow Matching for Graph Generation*, Eijkelboom et al., NeurIPS 2024
>
> [4] - *Discrete-state Continuous-time Diffusion for Graph Generation*, Xu et al., NeurIPS 2024
>
> [5] - *Cometh: A continuous-time discrete-state graph diffusion model*, Siraudin et al., ArXiv 2024
>
> [6] - *Generative Flows on Discrete State-Spaces: Enabling Multimodal Flows with Applications to Protein Co-Design*, Campbell et al., ICML 2024

---

> ### Comment · Reviewer_G38o · 2024-11-24
> **Response to the authors**
>
> I have read the response and I am not persuaded by the authors regarding Novelty. Specifically,
>
> - The work is essentially the approach first introduced by [1].
>     - The authors consider the permutation equivariance and invariance a technical contribution. However, equivariance can be trivially satisfied by the transformer architecture which is well-known and needs no further proof. The authors quote the work of [3] for the proof of invariant property. To conclude, The proof of Lemma 1 can not be regarded as a contribution to this paper.
>     - Again, the theoretical guarantees of Theorem 2 in Section 5 do not clearly connect with the main idea of the paper. i.e. what could potentially go wrong with the loss that it need the guarantee of Theorem 2? The authors did not provide a clear explanation. I personally don't like mistifying the paper with unnecessary theoretical guarantees, I didn't go through all the guarantees but at least Lemma 1 seems to be unnecessary, Theorem 2 is mysterious if not completely unnecessary.
>
> - The component RRWP which I believe contributed most of the performance improvement was first adopted to graph generation by [2]. This, in my opinion, can not be considered as an original contribution as well.
>
> The authors have requested for a clarification of the term "logical graph", it indeed is a misused term for "graph". Although I don't think this should be a major source of confusion given this paper is not involved with any other type of data structure, and the response of the authors seems to be in the right context, I apologize for the confusion.
>
> Regarding W5, I am satisfied that the authors have made some changes to the paper, however some more clarification from my side is needed.
> - $x^n$ can be easily identified as $x$ to the power of $n$. therefore suggest using $x^{(n)}$ instead.
> - At #229, I am well-aware that the authors have contextualized $p_{1|t}^\theta$ as a set. However, $p$ is usually used to represent some scalar probability. The presentation should be improved to avoid confusion, a simple change of the notation is not enough.
>
> Regarding Q4, I am not convinced by the authors' response.
> `initial distribution for graphs, This distribution should be permutation invariant`, 1. The distribution for graphs should be defined in the space of graphs, i.e. isomorphic graphs A and B are essentially one datapoint. 2. As an initial distribution to sample from, it is not necessary needed to be permutation invariant.
>
> Regarding Q5, I am convinced by the authors' response, but the authors should put this in the paper, specifically the text right before Eq. 7 (with $z_{t+dt} \neq z_t$),  to avoid confusion.
>
> ---
>
> [1] Andrew Campbell, Jason Yim, Regina Barzilay, Tom Rainforth, Tommi Jaakkola. "Generative Flows on Discrete State-Spaces:
> Enabling Multimodal Flows with Applications to Protein Co-Design"
>
> [2] Antoine Siraudin, Fragkiskos D. Malliaros, Christopher Morris. "COMETH: A CONTINUOUS-TIME DISCRETE-STATE GRAPH DIFFUSION MODEL"
>
> [3] Clement Vignac, Igor Krawczuk, Antoine Siraudin, Bohan Wang, Volkan Cevher, and Pascal Frossard. "Digress: Discrete denoising diffusion for graph generation"

---

> ### Author Response · Authors · 2024-11-24
>
> We thank the reviewer for their feedback and for acknowledging the improvements made in the revised version of our manuscript. We address the remaining concerns below.
>
> Regarding the novelty concerns:
>
> - **"The work is essentially the approach first introduced by [1]."**
>
>   While our work builds upon discrete flow matching, we introduce several key innovations that enable its first successful application to graph generation. These include a novel sampling optimization pipeline (including target guidance, sample distortion...), which are crucial for achieving state-of-the-art performance. Extensive experiments (see Figures 2, 6, and 7) demonstrate that a vanilla adaptation of DFM underperforms in this context, underscoring the necessity of our contributions for adapting DFM to the graph domain.
>
> - Regarding **Lemma 1**:
>
>   We respectfully clarify that the equivariance of the entire generative framework cannot be reduced to the equivariance of the transformer alone.
>   For instance, the applied loss function (node- and edge-wise cross-entropy) achieves permutation invariance only due to its specific formulation, which takes as input a clean graph and its independently noised version, both with the same node ordering.
>   Consequently, full equivariance arises from the combined effects of multiple components: (1) the equivariance of the denoising graph neural network (transformer), (2) the permutation invariance of the loss function, (3) the permutation equivariance of the RRWP encoding, and (4) the permutation invariance of the sampling probabilities.
>
>   While the first three components rely on results that are either straightforward or adapted from prior work, the proof for the fourth component is **unique to our framework**. This proof directly stems from the application of the discrete flow matching framework to graph generation. We consider this guarantee a key contribution, as it ensures the theoretical soundness of our model. Its importance is further underscored by the presence of similar guarantees in other diffusion-based graph generative models, as noted in our earlier response. Thus, we believe its formalization meaningfully strengthens our work.
>
> - Regarding **Theorem 2**:
>
>   Improper alignment between loss functions and objectives is a well-documented issue across various domains. In GANs, misaligned objectives often lead to mode collapse [1], where only a subset of the target distribution is captured. In VAEs, poorly designed losses can result in underexploration of the target distribution [2]. In reinforcement learning, inadequate reward shaping frequently produces suboptimal policies [3].
>
>   Theorem 2 establishes a principled foundation for the use of the cross-entropy loss in our framework by demonstrating its direct connection to minimizing errors in the estimated rate matrix.
>   This connection is pivotal for ensuring accurate sampling in the CTMC-based framework, as further reinforced by Theorem 3.
>
>   Importantly, discrete diffusion models were originally proposed with a loss function incorporating a variational upper bound on the negative log-likelihood.
>   However, when adapted to graphs, this loss was simplified to a node- and edge-wise cross-entropy without formal justification (in DiGress [4]).
>   By rigorously linking the cross-entropy loss to rate matrix estimation, we address this gap, thereby enhancing the theoretical robustness and empirical reliability of DeFoG.
>
> - Regarding **RRWP**:
>
>   We have included an Appendix section (G.3) that provides detailed results comparing DiGress with and without RRWP encodings.
>   These results demonstrate that, while RRWP encodings offer some benefits, their impact is relatively modest.
>   This is particularly evident in the sampling efficiency analysis (Fig. 18), where DiGress with RRWP encodings exhibits a similar trend to DiGress without them.
>   This indicates that the notable sampling efficiency of DeFoG with fewer steps is primarily driven by our optimized discrete flow matching formulation and sampling pipeline.
>
>   Figures 2, 6, and 7 further corroborate this conclusion, showing that our high performance cannot be solely attributed to the use of RRWP encodings.
>   Additionally, while we acknowledge in the paper that we are not the first to incorporate RRWP encodings into a generative framework, our method achieves superior results.
>   Importantly, we are the first to provide a detailed empirical ablation study on the benefits of RRWP within the generative setting.
>   These results justify that the **high performance of our method is not merely a consequence of RRWP encodings, but rather the result of the entire framework**.

---

> ### Author Response · Authors · 2024-11-24
>
> **W5 (Notation)**: We thank the reviewer for your further suggestions over the notation.
>   - We have modified the notation of $x^n$ to $x^{(n)}$ in the updated version of the manuscript.
>   - We believe we have addressed the noted notation ambiguities.
>     Specifically, we have explicitly clarified when $p$ represents a non-scalar, as stated, for example, in lines 171-172:
>     "Since we aim to reverse a $z_1$-conditional noising process $p_{t|1}(\cdot | z_1) \in \Delta^{Z-1}$, DFM instead considers a $z_1$-conditional rate matrix, $R_t(\cdot \, , \cdot|z_1) \in  \mathbb{R}^{Z \times Z}$ that generates it."
>
>     We also provide a specific definition for nodes and edges in a graph in lines 229-232 to reinforce this notation.
>     "The denoising of DFM requires a noisy graph $G_t$ as input and predicts the clean marginal probability for each node $x^{(n)}$ via $p_{1|t}^{\theta,(n)} (\cdot|G_t)\in \Delta^{X-1}$ and for each edge $e^{(ij)}$ via $p^{\theta,{(ij)}}_{1|t} (\cdot|G_t)\in \Delta^{E-1}$."
>
>     We believe the notation is consistent, since for all cases where we omit a dimension indicator (superscript $(d)$ or $(n)$) after $p_{1|t}$ in a multidimensional setting, we refer to the full joint variable. If the reviewer has specific additional suggestions, we would be happy to incorporate them.
>
> **Q4**: Thank you for requesting further clarification.
> The intention of the sentence is to emphasize that, for any two isomorphic graphs, $G_A$ and $G_B$, the probability of generating $G_A$ is identical to that of generating $G_B$ for any distribution within the defined class of distributions.
> Regarding your specific concerns:
> 1. We have revised the phrasing in lines 289-291 of the updated manuscript to eliminate any semantic ambiguity.
> 2. The initial distribution must generate any permutation of the same graph with equal probability to ensure the permutation invariance of the sampling probabilities of the model, thus crucial for Lemma 1 (detailed in lines 2373-2374).
> Additionally, this formulation allows for likelihood computation of our model, via exchangeability (see Lemma 3.3 of [4]). We do not explore this aspect in the current work, but it leaves the door open for future research.
>
> **Q5**: We have integrated the suggested modification into the updated version of the manuscript.
>
> As a final remark, we would like to emphasize that our framework achieves state-of-the-art
> performance and comparable performance with previous diffusion-based models
> with significantly fewer steps (5%).
> In our view, these results solidify our framework as a **substantial and innovative contribution to the field**.
>
> We believe that the constructive feedback provided by the reviewer has guided us toward addressing their concerns more effectively.
> In light of these improvements, we hope the reviewer might consider revising their score.
>
> ---
>
> [1] - *Wasserstein GAN*, Arjovsky et al., ICML 2017
>
> [2] - *Generating Sentences from a Continuous Space*, Bowman et al., CoNLL 2015
>
> [3] - *Policy invariance under reward transformations: Theory and application to reward shaping*, Ng et al., ICML 1999
>
> [4] - *DiGress: Discrete Denoising Diffusion for Graph Generation*, Vignac et al., ICLR 2023

---

> > ### Author Response · Authors · 2024-12-01
> >
> > Dear Reviewer,
> >
> > Thank you again for your feedback. As we approach the end of the rebuttal period, we wanted to check if our responses have addressed your concerns or if there are any remaining points we could clarify.
> >
> > Best regards,
> >
> > The authors

---

### Official Review · Reviewer_jSH4 · 2024-10-30

**Soundness:** 4
**Presentation:** 3
**Contribution:** 2
**Rating:** 6
**Confidence:** 4

**Summary:**

The paper adds to the growing literature of graph generation using diffusion/flow models by adapting a continuous-time discrete-space flow matching framework. This framework is tailored for graph generation through a graph transformer architecture with Relative Random Walk Probabilities encodings, a loss function that differentially weights node connectivity and features, a finely-tuned noise schedule, an appropriate prior distribution, and optimized rate matrices for sampling. The method is evaluated on a comprehensive set of benchmarks, demonstrating strong results in sample fidelity and efficiency compared to previous diffusion-based methods.

**Strengths:**

The paper is clearly written and well-organized. Several changes to the adapted flow matching framework are proposed, each of which is well-motivated and theoretically justified, with its empirical impact thoroughly evaluated in an ablation study. The extensive numerical experiments cover small to medium-sized synthetic and molecular graphs, consistently showing competitive or state-of-the-art performance across all benchmarks. Notably, the comparison on sampling and training efficiency underscores the value of the proposed method over previous discrete-time graph diffusion approaches.

**Weaknesses:**

My primary criticism is the limited novelty of the contribution. There is a substantial body of work on graph generation by adapting generative models to the graph domain, recently including various diffusion/flow models. The proposed method is incremental in this context, primarily confirming the effectiveness of the adapted flow matching approach by Campbell et al. in the graph generation setting.

Although the method is extensively evaluated against previous work, the related work section is rather sparse and does not mention several works the authors compare against in the experiments. I recommend expanding this section to provide a more comprehensive overview of the existing literature on graph generation.

**Questions:**

When comparing the efficiency of the proposed method to ablated models and DiGress (Figures 2, 6a, 7a), which version of DiGress is used for the comparison? Is it the original version? As the proposed method also includes architectural improvements (e.g., Relative Random Walk Probabilities encodings), I suggest including a comparison of DiGress with the same architecture as the proposed method to better assess the impact of the continuous-time vs. discrete-time formulation.

Additionally, why do Figures 6b, 6c, 7b, and 7c not include the comparison to DiGress?

---

> ### Author Response · Authors · 2024-11-20
>
> We sincerely appreciate the reviewer’s constructive feedback and requests for clarification. We have addressed the raised questions below:
>
> **W1**: We believe that our work offers substantial advances beyond existing frameworks in both theoretical insights and performance. Furthermore, our experiments emphasize that while DFM is not directly tailored for graph generation, it holds significant potential due to its flexible exploration space. Our comprehensive investigation into DeFoG's design space has been instrumental in realizing the observed state-of-the-art performance. For additional details, please refer to the "Novelty" section in the general comments. In summary, we present the first method to enable systematic, effective, and extensively studied sampling optimization, which our experiments demonstrate as crucial for achieving state-of-the-art performance.
>
> **W2**: We thank the reviewer for the suggestion. Unfortunately, due to the author guidelines for the rebuttal, we are constrained by space limitations in the main text. Nevertheless, to address the reviewer's concern we have added a new section in the appendix (Appendix A) that provides a more comprehensive and detailed overview of the related research methods. There, we introduce the several compared methods and provide a detailed and compact contextualization of DeFoG within the diffusion-based graph generative models landscape. We hope this provides a more detailed context for our work, but we remain open to further suggestions in case we missed any relevant work.
>
> **Q1**: Again, we thank the reviewer for the suggestion and we included a more comprehensive study about RRWP in Appendix G.3 in response. Please refer to the point **Isolate Effect of RRWP** of the general comment for the results of the ablation study comparing DiGress with the RRWP encoding.
>
> **Q2**: DiGress was initially excluded from Figures 6b, 6c, 7b, and 7c because Figures 6a and 7a were intended as illustrative examples for synthetic and molecular datasets, rather than exhaustive coverage. Additionally, the original code repository did not provide checkpoints for these datasets. However, we have now run the code and obtained the required checkpoints ourselves. The updated results, included in the revised version of our PDF, confirm that DeFoG outperforms DiGress in both efficiency and performance, while maintaining consistent trends across datasets.
>
> We hope these clarifications address the raised concerns and provide a more comprehensive overview of our work. We remain at your disposal for any further questions or clarifications.

---

> > ### Author Response · Authors · 2024-11-27
> >
> > Dear reviewer,
> >
> > We would like to know if we addressed your concerns or if there is any point that you would like us to further discuss. Your feedback is highly valued.
> >
> > Thank you for your time.
> >
> > Best regards,
> >
> > The authors

---

> > ### Comment · Reviewer_jSH4 · 2024-11-28
> >
> > I appreciate the authors' detailed response and the enhancements made to the manuscript.
> > The added experimental results (Digress with the same architecture as the proposed method) give a good insight into the impact of the continuous-time vs. discrete-time formulation.
> >
> > While I acknowledge the novelty of the proposed method, I still consider the contribution to be incremental within the expanding body of literature on graph generation using diffusion/flow models. Therefore, I maintain my score of 6.

---

> > > ### Author Response · Authors · 2024-12-01
> > >
> > > Dear reviewer,
> > >
> > > Thank you for your feedback and for settling your review. We greatly appreciate your time and insights.
> > >
> > > Best regards,
> > >
> > > The authors

---

### Official Review · Reviewer_Jrtv · 2024-10-30

**Soundness:** 3
**Presentation:** 3
**Contribution:** 2
**Rating:** 6
**Confidence:** 5

**Summary:**

The paper presents DeFoG, a graph generation framework based on discrete flow matching (DFM) that overcomes limitations in existing diffusion-based methods for graph generation. DeFoG introduces a linear interpolation noising process and a denoising method using continuous-time Markov chains (CTMCs). It study how to disentangle training and sampling stages, thus enhancing model optimization flexibility and efficiency. Result of DeFoG is shown to achieve state-of-the-art results across synthetic and molecular datasets.

**Strengths:**

The paper make a first attempt on applying discrete flow matching on the graph generation problem.

Particularly, the paper make a extensive exploration of how to make DFM work perfectly on it. It

(1) Explore on the design choice of marginal distribution and training timestep schedule.

(2) Explore on sampling design, including sampling time step schedule, stochaticity rate matrix(similar to the predictor-corrector ) as well as guided diffusion.

(3) Explore on model architecture design.

The exploration is extensive and provides valuable experience for the research community

**Weaknesses:**

The only weakness is in the innovation - DFM is a well-established framework in recent years. The paper does not make key methodology innovation rather than make exploration. Moreover, it's not clear what's the major limitation of the previous diffusion-based graph model, the paper should highlight how the paper address them.

**Questions:**

See weakness

---

> ### Author Response · Authors · 2024-11-20
>
> We appreciate the reviewer's feedback and the acknowledgement of our extensive exploration of DeFoG's design space. We address the raised concerns below:
>
> **W1**: To address concerns regarding innovation, we kindly refer the reviewer to our general comments on **Novelty**, where we provide a detailed explanation of how DeFoG addresses critical gaps in the graph generation research community.
>
> Regarding the comparison with previous diffusion-based graph models, we believe this has been addressed in multiple sections of the paper. Below, we highlight key points from our work. Nevertheless, to make this comparison more compact and detailed, we added a new section to the appendix highlighting the advantages of DeFoG *vs* previous continuous diffusion models (see Appendix A.2).
>
> We first remark in the Introduction:
>
>     "However, the sampling processes of these methods [referring to previous graph diffusion models] remain highly constrained due to a strong interdependence with training: design choices made during training mostly determine the options available during sampling. Therefore, optimizing parameters such as noise schedules requires re-training, which incurs significant computational costs." and "Specifically, its [referring to DFM formulation] noising process involves a linear interpolation in the probability space between noisy and clean data, displaying well-established advantages for continuous state spaces in terms of performance (Esser et al., 2024; Lipman et al.) and theoretical properties (Liu et al., 2023). Most notably, the denoising process in DFM uses an adaptable sampling step scheme (both in size and number) and CTMC rate matrices that are independent of the training setup. Such a decoupling allows for the design of sampling algorithms that are disentangled from training, enabling efficient performance optimization at the sampling stage without extensive retraining. However, despite its promising results, the applicability of DFM to graph generation remains unclear, and the proper exploitation of its additional training-sampling flexibility is still an open question in graph settings".
>
> In the following paragraphs of the Introduction, we ellaborate on how to successfully apply DFM to graph generation, yielding DeFoG. Then, in the "Graph Diffusion" paragraph of the Related Work section:
>
>     "One of the initial research directions in graph diffusion sought to adapt continuous diffusion frameworks for graph-structured data (Niu et al., 2020; Jo et al., 2022; 2024), which introduced challenges in preserving the inherent discreteness of graphs. In response, discrete diffusion models (Austin et al., 2021) were effectively extended to the graph domain (Vignac et al., 2022; Haefeli et al.), utilizing Discrete-Time Markov Chains to model the stochastic diffusion process. However, this method restricts sampling to the discrete time points used during training. To address this limitation, continuous-time discrete diffusion models incorporating CTMCs have emerged (Campbell et al., 2022), and have been recently applied to graph generation (Siraudin et al., 2024; Xu et al., 2024). Despite employing a continuous-time framework, their sampling optimization space remains limited by training dependent design choices, such as fixed rate matrices, which hinders further performance gains."
>
> In the next paragraph of the same section, "Discrete flow Matching", we provide more details in the comparison to continuous-time diffusion models:
>
>     "This DFM approach theoretically streamlines its diffusion counterpart by employing linear interpolation for the mapping from data to the prior distribution. Moreover, its CTMC-based denoising process accommodates a broader range of rate matrices, which need not be fixed during training. In practice, DFM consistently outperforms traditional discrete diffusion models. In this paper, we build on the foundations of DFM with the aim of further advancing graph generative models."
>
> Later, in section "Exploring DeFoG's Design Space" (4.2), we reiterate:
>
>     "While discrete diffusion models offer a broader design space compared to one-shot models, including noise schedules and diffusion trajectories (Austin et al., 2021), its exploration is costly. In contrast, flow models enable greater flexibility and efficiency due to their decoupled noising and sampling procedures. For instance, the number of sampling steps and their sizes are not fixed as in discrete-time diffusion (Vignac et al., 2022), and the rate matrix design can be adjusted independently, unlike continuous-time diffusion models (Siraudin et al., 2024; Xu et al., 2024). In this section, we exploit this rich design space and propose key algorithmic improvements to optimize both the training and sampling stages of DeFoG."

---

> > ### Author Response · Authors · 2024-11-20
> >
> > Considering this, we believe we have made a meaningful effort to highlight the limitations of previous diffusion-based graph models and demonstrate how DeFoG addresses them. Nonetheless, we are open to any further suggestions on how to improve this comparison.
> >
> > We hope this clarifies the innovation and contribution of our work. We remain available for any further questions or clarifications.

---

> > > ### Comment · Reviewer_Jrtv · 2024-11-26
> > > **Question about Guacamol validity**
> > >
> > > I am checking Your figure 7(c), where you perform ablation study on the sampling steps. When using #steps=500, DeFoG gives ~90% validity at best on guacamol. While your reported result in table 2 is 99. Is there an inconsistency?

---

> > > > ### Author Response · Authors · 2024-11-27
> > > >
> > > > We thank the reviewer for the careful evaluation and for bringing up this observation.
> > > >
> > > > In Figure 7, we conduct an extensive hyperparameter search. For efficiency, for MOSES and Guacamol, we sample 2,000 molecules per configuration and we use locally implemented metrics for validity and novelty evaluation, as noted in the figure caption. These metrics are significantly faster than the original MOSES and Guacamol benchmarks but differ slightly in their definitions. Specifically, for consistency across the three datasets, we report a *stricter* validity, which invalidates charged molecules, as our primary goal in this figure is to compare the relative performance of DiGress and the different DeFoG configurations.
> > > >
> > > > The discrepancy arises because Table 2 reports results obtained using the GuacaMol benchmark, which allows charged molecules, whereas Figure 7 uses our local strict validity metric. This distinction accounts for the lower validity percentages noted by the reviewer.
> > > > Importantly, this discrepancy does not occur for MOSES, which does not contain molecules with charges at all, or QM9, which contains only a few molecules with charges. To address this, we have updated the figure caption to clarify this distinction.
> > > >
> > > > Additionally, we have a local implementation of a relaxed validity metric that accommodates charged molecules.
> > > > So, for example, for the 500-step Guacamol point, we obtain 92.3% for strict validity and 99.0% for relaxed validity.
> > > > If the reviewer finds it appropriate, we are happy to include or replace the existing results with this relaxed metric to ensure greater consistency with Table 2.

---

> > > > > ### Author Response · Authors · 2024-12-01
> > > > >
> > > > > Dear Reviewer,
> > > > >
> > > > > Thank you again for your feedback. As we approach the end of the rebuttal period, we wanted to check if our responses have addressed your concerns or if there are any remaining points we could clarify.
> > > > >
> > > > > Best regards,
> > > > >
> > > > > The authors

---

### Official Review · Reviewer_wsK5 · 2024-11-04

**Soundness:** 2
**Presentation:** 3
**Contribution:** 2
**Rating:** 5
**Confidence:** 4

**Summary:**

This work proposes a discrete flow matching framework for graph generation. Based on the existing discrete flow matching, the authors propose several techniques regarding the model architecture, the training process, and the sampling process. Specifically,

**Model Architecture**
- To enhance the model expressivity, the authors adopt the Relative Random Walk Probabilities (RRWP) from the previous work [1] to encode the graphs more informatively.

**Training Process**
- The authors show that the marginal distribution is beneficial for discrete flow matching, adopting it from the previous discrete denoising diffusion model for graph generation [2].
- Instead of directly using the time $t$ sampled from a uniform distribution, the authors explore various methods for distorting the sampled time inspired by the previous work for image generation [3].

**Sampling Process**
- This work proposes distortion functions that vary the step size of the sampling process which is effective in yielding performance improvement.
- The authors introduce a guidance term to ensure the robustness when sampling.
- DeFoG introduces a hyperparameter ($\eta$) that can control the stochasticity of the generative process.
- The authors explore the exact computation of the expectation to guarantee the validity of the generated graphs by reducing unintended stochasticity.

Experimentally, the proposed DeFoG is evaluated on the synthetic graph generation, and molecular graph generation, which show comparable performances.


[1] Ma et al., "Graph inductive biases in transformers without message passing.", ICML 2023.

[2] Vignac et al., "DiGress: Discrete Denoising diffusion for graph generation", ICLR 2023.

[3] Esser et al., "Scaling Rectified Flow Transformers for High-Resolution Image Synthesis", ICML 2024.

**Strengths:**

- The authors propose several techiniques that are tailored to the discrete flow matching, and show these technique improve the performance.
- The ablation study on the proposed techniques both for the training and sampling process is well designed.

**Weaknesses:**

- The connection between the graph generation and the proposed method is weak. The authors argue that the proposed method is tailored to the graph generation. However, I am concerned that the proposed method is not specific to the graph-structured data except for the encoding part (i.e. RRWP). Specifically, the proposed methods for training and sampling are not motivated by the characteristics of the graph-structured data but by the discrete flow matching itself.

- It seems to be unfair as the proposed model architecture has a modification compared to DiGress [2]. Please provide the effect of using RRWP.

- Some related works are not experimentally compared. CatFlow [4] which is based on the flow matching for the graph generation and GruM [5] which is based on the diffusion mixture tailored for the graph generation are competitive baselines that shoulde be compared in this paper.

[4] Eijkelboom et al., "Variational Flow Matching for Graph Generation", NeurIPS 2024.

[5] Jo et al., "Graph Generation with Diffusion Mixture", ICML 2024.

**Questions:**

- How much does the exact computation of the expectation cost?

---

> ### Author Response · Authors · 2024-11-20
>
> We thank the reviewer for their insightful feedback. We address each point below:
>
> **W1**: Please refer to the points "Graph-specificity" and "Novelty" of the general comment.
>
> **W2**: We thank the reviewer for the suggestion regarding the RRWP encodings. We included a more comprehensive study about RRWP in Appendix G.3. Please refer to the point **Isolate Effect of RRWP** of the general comment.
>
> **W3**: We thank the reviewer for the pointers. We have evaluated both CatFlow and GruM with our evaluation pipeline and have added them to our related work discussion (see section *Related Work* and *Appendix A*).
>
> - **GruM**: for synthetic datasets, we used directly the generated graphs by GruM (readily available in the official repository) and evaluated them with our pipeline. For the molecular dataset (QM9), the generated smiles provided result from *a posteriori* correction, which would lead to an unfair comparison. Therefore, we re-sampled the graphs using the provided GruM sampling codes and checkpoints, and, again, passed these graphs through our molecular evaluation pipeline. In particular, unlike the original paper, we report strict validity rather than the relaxed version to maintain consistency across our results on the QM9 dataset.
> *Discussion*: DeFoG demonstrates improved performance over GruM on the Planar and SBM datasets, with the exception of the average ratio metric on SBM. On the QM9 dataset, both methods have comparable performance. However, while not in our original tables, we remark that the novelty score of DeFoG is 33.2% compared to GruM's 25.2%. This suggests that GruM is more prone to overfitting to the training dataset, while DeFoG generates more diverse molecular structures. (*Note*: these low values of novelty for highly performant graph generative models in QM9 are not unusual, since this dataset consists of an exhaustive enumeration of the small molecules complying with a given set of constraints [1]).
>
> - **CatFlow**: However, we are not able to compute the average ratio metric, nor to run our molecular evaluation with the information available. Unfortunately, we were unable to find any official implementation of the proposed method. The code submitted as supplementary material in OpenReview seems to be a preliminary version (has no README, we could not reproduce the environment, ...), which does not allow us to provide a fair comparison. We tried to reach out to the authors, but we did not receive any response. If this status changes during the rebuttal period, we will update our results accordingly.
> *Discussion*: For synthetic datasets, we use the V.U.N. values reported in their paper: 80.0 and 85.0 for Planar and SBM, respectively. DeFoG outperforms CatFlow on both datasets.
>
>
> | Method         |               | **Planar**               |               | **SBM**                 |               | **QM9**                   |           |         |
> |----------------|---------------|--------------------------|---------------|-------------------------|---------------|---------------------------|-----------|---------|
> |                |               | V.U.N.                   | Avg. Ratio    | V.U.N.                  | Avg. Ratio    | Val.                      | Uniq.     | FCD     |
> | GruM |    | $90.0$              | $1.8$                         | $85.0$              | $1.1$                        |           $99.2$                          |$96.7$          | $0.11$        |
> | CatFlow |    | $80.0$              | ---                         | $85.0$              | ---                        |           ---                          |---          | ---        |
> | DeFoG (50 steps)   |               | $95.0\pm 3.2$                      | $3.2\pm1.1$          | $86.5\pm5.3$                        | $2.2\pm0.3$              | $98.9\pm0.1$                          | $96.2\pm0.2$          | $0.26\pm0.00$        |
> | DeFoG |               | $99.5\pm 1.0$                      | $1.6\pm0.4$          | $90.0\pm5.1$                        | $4.9\pm1.3$              | $99.3\pm0.0$                          | $96.3\pm0.3$          | $0.12\pm0.00$        |

---

> ### Author Response · Authors · 2024-11-20
>
> **Q1**: The exact computation of the expectation requires evaluating a weighted sum over possible rate matrices:
>
> $E_{p_{{1|t}}^{\theta,d} (z_1^d | z_t^{1:D})} [ R_t^d(z_t^d, z_{t + \Delta t}^d | z_1^d)] = \sum_{z_1^d} p_{1|t}^{\theta,d} (z_1^d | z_t^{1:D}) R^d_t(z_t^d, z_{t + \Delta t}^d | z_1^d)$
>
> This involves a sweep over the possible states of $z_1^d$. For graphs, this corresponds to a full sweep over possible node and edge states per each denoising step.
> However, since one forward pass of the denoising neural network computes $p_{1|t}^{\theta,d}$ for all $z^d_1$, the exact expectation does not require additional network calls.
>
> In our experiments, the cardinalities of the node ($X$) and edge ($E$) state spaces are relatively small:
>
> - Synthetic datasets: $X=1$, $E=2$;
> - Molecular datasets: $X=4$, $E=5$;
> - Digital Pathology datasets: $X=2$, $E=9$.
>
> The table below summarizes the runtimes (mean ± std over 5 runs) for representative datasets:
>
> | Dataset | Approximate   (s)      | Exact (s) | Overhead (%) |
> |-------------|-----------------|-------|----------|
> |Planar (Synthetic) | 117.92 ± 0.74 | 119.83 ± 0.40 | + 1.62 |
> | QM9 (Molecular) | 269.65 ± 0.90   |275.18 ± 0.86| + 2.05|
> | Digital Pathology | 629.00 ± 30.57 | 667.45 ± 32.84 | + 6.11 |
>
> These results show that the computational overhead of the exact expectation in our setting is minimal. Importantly, as anticipated, the overhead increases with the cardinalities of the state spaces.
>
> We hope these clarifications address the raised concerns and provide a more comprehensive overview of our work. We remain at your disposal for any further questions or clarifications.
>
> [1] - *DiGress: Discrete Denoising Diffusion for Graph Generation*, Vignac et al., ICLR 2023

---

> > ### Author Response · Authors · 2024-11-27
> >
> > Dear reviewer,
> >
> > We would like to know if we addressed your concerns or if there is any point that you would like us to further discuss. Your feedback is highly valued.
> >
> > Thank you for your time.
> >
> > Best regards,
> >
> > The authors

---

> ### Author Response · Authors · 2024-12-01
>
> Dear Reviewer,
>
> As we approach the end of the rebuttal period, we wanted to check if our responses have addressed your concerns or if there are any remaining points we could clarify.
>
> Best regards,
>
> The authors

---

### Author Response · Authors · 2024-11-20
**General Comment**

First of all, we thank the reviewers for their time and valuable feedback. Here, we provide a general comment to address shared concerns.

**Novelty**: Our work, as the first discrete flow matching-based graph generative model, introduces substantial advances beyond existing frameworks, offering both theoretical insights and practical innovations in graph generation.

Discrete state-space continuous-time diffusion frameworks have emerged as a powerful approach to graph generation [1,2], since this framework naturally aligns with the discrete nature of graph structures, while leveraging the robustness and flexibility of continuous-time modeling. However, the training and stages of existing graph diffusion-based models remain highly interdependent. This prevents an extensive exploration of the design space of those models, as design choices made during training mostly determine the options available during sampling. Therefore, optimizing parameters such as noise schedules or rate matrices requires re-training, which incurs significant computational costs.

DeFoG borrows the principles of the DFM framework to fill this gap within the graph generation domain. However, this adaptation to graph generation is not straightforward.
First, the DFM framework is not as theoretically well-established as continuous-time discrete diffusion models [3].
Second, the original formulation of DFM does not contemplate the specific challenges of graph generation, such as the preservation of permutation equivariance (see point **Graph-specificity**).
Third, as the sampling time is fully decoupled from training time, the sampling stage design space is significantly larger than the continuous-time discrete diffusion models. This space remains under-explored, especially in the graph setting.

DeFoG covers the three points highlighted above.
For the first point, while previous work [5] mainly focuses on the rate matrix validity to verify the Kolmogorov equation used in CTMC simulations, DeFoG provides new theoretical guarantees to both the training and sampling processes.
In particular, we reinforce the motivation of the training loss function, which originally was related to ELBO, by also directly relating it to the rate matrix estimation error. This result provides a more direct connection between the training loss and the sampling process. We also further motivate the approximate CTMC simulation method used during sampling by providing an upper bound on its error. This bound is crucial for understanding the sampling process and provides a solid foundation for its application to graph data.

For the second point, our work is the first model for graph generation based on DFM principles, representing a meaningful addition to the field. For more details, refer to the point **Graph-specificity**.

For the third point, we note that DeFoG's vanilla sampling configuration is insufficient to achieve state-of-the-art performance, as shown in our experiments. However, the training-sampling decoupling formulation allows us to explore its design space through a series of novel algorithmic contributions that unlock the full potential of the DFM framework for graph generation. While certain components, such as stochasticity during sampling via $R^\text{DB}$ [5], draw from prior work, our key innovations — most notably target guidance and sample distortion — address limitations of the vanilla framework. We consider this extensive exploration of DeFoG's optimization space a contribution in itself, as it offers rich empirical insights into graph generative models. Such explorations have proven pivotal in other domains, as exemplified by Karras et al. [4] for continuous diffusion models.

Finally, we observe that our novel optimization pipeline with the multiple contributions described above is essential to achieve state-of-the-art performance, as demonstrated in Figures 2, 6, and 7. We emphasize our contribution through strong performance with as few as 50 steps, significantly improving efficiency across diverse datasets. This efficiency does not only highlight the practicality of our approach but also establishes a benchmark for future  generative modeling research in the community.

---

> ### Author Response · Authors · 2024-11-20
> **General Comment (cont.)**
>
> **Graph-specificity**:
> DeFoG is specifically tailored to the graph generation domain. First, we address key challenges in graph-specific modeling, ensuring that the full framework achieves permutation equivariance — a scenario radically different from the one originally considered in DFM [5]. We manage to overcome GNN architectural expressivity limitations by incorporating graph transformers and RRWP encodings. In particular, we provide new results evaluating the time complexity and efficacy of these encodings in graph generation performance and sampling efficiency (see **Isolate Effect of RRWP**). Additionally, we adopt a differentially weighted loss function for nodes and edges, which enforces a differential bias during training justified by their distinct roles in the graph structure.
>
> Furthermore, we explore a class of initial distributions of the noising process, $p_\epsilon$, that maintain DeFoG's permutation equivariance. Our aim is to identify which distributions provide better inductive biases for graph generation tasks, connecting our findings to existing results in the graph generation literature. We also offer insights specific to graph generation, such as identifying the types of time distortions that enhance generation quality for different datasets. These findings reveal significant differences from other data modalities, e.g.:
>
>     "In contrast to prior findings in image generation, which suggest that focusing on intermediate time regions is preferable (Esser et al., 2024), we observe that for most graph generation tasks, the best-performing distortion functions particularly emphasize $t$ approaching 1. Our key insight is that, as $t$ approaches 1, discrete structures undergo abrupt transitions between states — from 0 to 1 in a one-hot encoding — rather than the smooth, continuous refinements seen in continuous domains. Therefore, later time steps are critical for detecting errors, such as edges breaking planarity or atoms violating molecular valency rules."
>
>
> Finally, DeFoG's decoupling of training and sampling enables flexible sampling procedures that previous diffusion-based graph generative models did not offer. This flexibility allows us, for the first time to the best of our knowledge, to explore varying stochasticity configurations ($R^\mathrm{DB}$) at sampling time specifically for graph settings. Additionally, we propose a target guidance mechanism ($R^\omega$), which is crucial to DeFoG's success in graph generation tasks. While some building blocks (e.g., target guidance) may not be exclusively applicable to graphs, their adaptability to other domains highlights the distinct strength of our method. Nevertheless, all our contributions are designed for, developed, and validated on graph data.

---

> > ### Author Response · Authors · 2024-11-20
> > **General Comment (cont.)**
> >
> > **Isolate Effect of RRWP**:
> > We present an ablation study to isolate the effect of utilizing the RRWP encoding *vs* the previously employed additional features within the DiGress framework. In particular, we add the results of running DiGress with the RRWP encodings and compare it with the original DiGress and DeFog, across three datasets: QM9, Planar, and SBM. The results demonstrate that RRWP achieves comparable or superior performance within the DiGress framework, validating its effectiveness as a graph encoding method. Despite these improvements, DiGress's performance remains significantly below that of DeFoG on the Planar and SBM datasets, while achieving similar validity and uniqueness on the QM9 dataset. A full ablative study over the impact of RRWP to the sampling efficiency is further included in the Appendix G.3 of the revised version of our PDF.
> >
> >
> > | Method         |               | **Planar**               |               | **SBM**                 |               | **QM9**                   |           |         |
> > |----------------|---------------|--------------------------|---------------|-------------------------|---------------|---------------------------|-----------|---------|
> > |                |               | V.U.N.                   | Avg. Ratio    | V.U.N.                  | Avg. Ratio    | Val.                      | Uniq.     | FCD     |
> > | DiGress |               | $77.5$                     | $5.1$           | $60.0$          | $1.7$        | $99.0\pm0.0$                        | $96.2\pm0.1$              | -                          |
> > | DiGress (RRWP) |    | $90.0$              | $4.0$                         | $70.0$              | $1.7$                        |           $99.0$                          | $96.2$          | -        |
> > | DeFoG (RRWP, 50 steps)   |               | $95.0\pm 3.2$                      | $3.2\pm1.1$          | $86.5\pm5.3$                        | $2.2\pm0.3$              | $98.9\pm0.1$                          | $96.2\pm0.2$          | $0.26\pm0.00$        |
> > | DeFoG (RRWP)   |               | $99.5\pm 1.0$                      | $1.6\pm0.4$          | $90.0\pm5.1$                        | $4.9\pm1.3$              | $99.3\pm0.0$                          | $96.3\pm0.3$          | $0.12\pm0.00$        |
> > |
> >
> > We hope these clarifications underscore the innovation and impact of our work within the graph generation landscape. We have also integrated all the new experimental results requested by the reviewers in the new version of our PDF. Again, we thank the reviewers for their feedback and we hope that our responses address their concerns. We remain at your disposal for any further questions or clarifications.
> >
> > [1] - *Discrete-state Continuous-time Diffusion for Graph Generation*, Xu et al., NeurIPS 2024
> >
> > [2] - *Cometh: A continuous-time discrete-state graph diffusion model*, Siraudin et al., ArXiv 2024
> >
> > [3] - *A Continuous Time Framework for Discrete Denoising Models*, Campbell et al., NeurIPS 2022
> >
> > [4] - *Elucidating the Design Space of Diffusion-Based Generative Models*, Karras et al., NeurIPS 2022
> >
> > [5] - *Generative Flows on Discrete State-Spaces: Enabling Multimodal Flows with Applications to Protein Co-Design*, Campbell et al., ICML 2024

---

### Meta-Review · Area_Chair_dpU2 · 2024-12-20

**Metareview:**

This paper tackles graph generation problem, utilising approaches based on discrete flow matching. Despite the interesting idea using flow matching on graphs, multiple reviewers pointed out the methodology has built primarily on previous works including DiGress and discrete flow matching, which largely limit its novelty and methodology contribution. It would be suggested for another round of revision and polish before appear in the conference as ICLR.

**Additional Comments On Reviewer Discussion:**

The reviewers' concerns are partially resolved.

---

### Decision · Program_Chairs · 2025-01-22

Reject